# Wdr4 regulates ribosome biogenesis and intestinal homeostasis via *let-7*

Kreeti Kajal [1,2,3], Elham Rastegari[3,4], Wen-Der Wang [5], Jian-Chiuan Li[6], Chun-Hong Chen[6], Wan Hsuan Chou [7], Wei Chiao Chang[7], Tzu-Yang Lin [3], Kevin Tsai [8], Tsai Ming Lu [3], Kartik Venkatachalam [4,9] & Hwei-Jan Hsu [1,2,3 ✉]

## Abstract

Proper regulation of ribosome biogenesis is essential for stem cell function and tissue homeostasis, yet its upstream control in adult intestinal stem cells (ISCs) remains unclear. Here, we identify the WD repeat protein Wdr4 as a key regulator of ISC homeostasis in the *Drosophila* midgut. Wdr4 cooperates with the methyltransferase Mettl1 to catalyze $N^7$-methylguanosine (m$^7$G) modification of *let-7* miRNA. Wdr4 or Mettl1 depletion disrupts this modification, reducing *let-7* levels and aberrantly activating TOR-JNK-dMyc signaling. This drives elevated ribosome biogenesis, ISC overproliferation, misdifferentiation, and intestinal dysplasia. Overexpression of *let-7*, inhibition of TOR, or suppression of JNK rescues these defects. Importantly, expression of human WDR4 and METTL1, but not catalytic-dead METTL1 mutant, restores ISC homeostasis in Wdr4- and Mettl1-depleted flies, establishing a conserved Wdr4/Mettl1-*let-7*-TOR-JNK axis that links miRNA modification to translational control and tissue integrity. Together, our findings uncover a previously unrecognized function of miRNA m$^7$G methylation in regulating ribosome biogenesis and maintaining intestinal homeostasis.

**Keywords** TOR; JNK; Myc; ISC; m$^7$G
**Subject Categories** RNA Biology; Signal Transduction; Stem Cells & Regenerative Medicine

## Introduction

Gut homeostasis is vital for the health and longevity of all metazoans, relying on the maintenance and regeneration of gastrointestinal tissues by intestinal stem cells (ISCs). Disruption of ISC function—through aberrant proliferation or differentiation —can lead to disorders such as leaky gut syndrome and tumorigenesis. Recent studies emphasize that ribosome biogenesis and protein translation are tightly regulated processes essential for stem cell function and aging (Martínez Corrales et al, 2020; Tiku et al, 2017). However, how ribosome biogenesis is controlled in ISCs remains unclear.

The *Drosophila* intestine provides an ideal model to investigate these questions due to its well-characterized ISC lineage and powerful genetic tools. The adult fly gut consists of three main regions: the foregut, midgut, and hindgut (see Fig. 1D). The foregut contains the crop for food storage (Stoffolano and Haselton, 2013), while the midgut functions like the small intestine and partly like the stomach, performing digestion and absorption (Buchon et al, 2013). This tissue is grossly subdivided into the anterior midgut (AMG), the middle midgut (MMG), and the posterior midgut (PMG). The anterior midgut encompasses the proventriculus. This region opens into the acidic midgut, which contains acid-secreting copper cells that aid in food digestion (Dubreuil, 2004). The AMG includes the entry region of the gut, which connects to the acidic midgut. The acidic midgut contains specialized copper cells that secrete acid to help digest food (Buchon et al, 2013; Marianes and Spradling, 2013). Within the PMG, ISCs rapidly regenerate the epithelium through well-characterized mechanisms (Jiang et al, 2016). The PMG consists of five major cell types: ISCs, two immediate daughter cells [enteroblasts (EBs), and EE progenitors (pEEs)], and two differentiated cells [enterocytes (ECs), and enteroendocrine cells (EEs)]. With high Notch signaling, EBs differentiate into ECs, which undergo extensive growth and DNA endoreplication, constituting the majority of the midgut cell population (Micchelli and Perrimon, 2006); pEEs, with low Notch signaling, differentiate into EEs (Zeng and Hou, 2015). ECs secrete digestive enzymes and absorb nutrients, while EEs secrete gut hormones that regulate gut mobility and function in response to external stimuli and bacteria.

WD repeat domain 4 (Wdr4) encodes a conserved protein with five WD40 domains involved in protein-protein interactions (Schapira et al, 2017; Wu et al, 2006). Wdr4 exerts a wide range

[1]Molecular and Biological Agricultural Sciences Program, Taiwan International Graduate Program, National Chung Hsing University and Academia Sinica, Taipei 11529, Taiwan. [2]Graduate Institute of Biotechnology, National Chung Hsing University, Taichung 40227, Taiwan. [3]Institute of Cellular and Organismic Biology, Academia Sinica, Taipei 11529, Taiwan. [4]Department of Integrative Biology and Pharmacology, McGovern Medical School at the University of Texas Health Sciences Center, Houston, TX, USA. [5]Department of Bio Agricultural Science, Chiayi University, Chiayi City 6004, Taiwan. [6]National Institute of Infectious Diseases and Vaccinology, National Health Research Institutes, Zhunan 35053, Taiwan. [7]Department of Clinical Pharmacy, School of Pharmacy, Taipei Medical University, Taipei 11031, Taiwan. [8]Institute of Biomedical Sciences, Academia Sinica, Taipei 11529, Taiwan. [9]Graduate Programs in Neuroscience and Molecular and Translational Biology, MD Anderson Cancer Center and UTHealth Graduate School of Biomedical Sciences, Houston, TX, USA. ✉E-mail: cohsu@gate.sinica.edu.tw

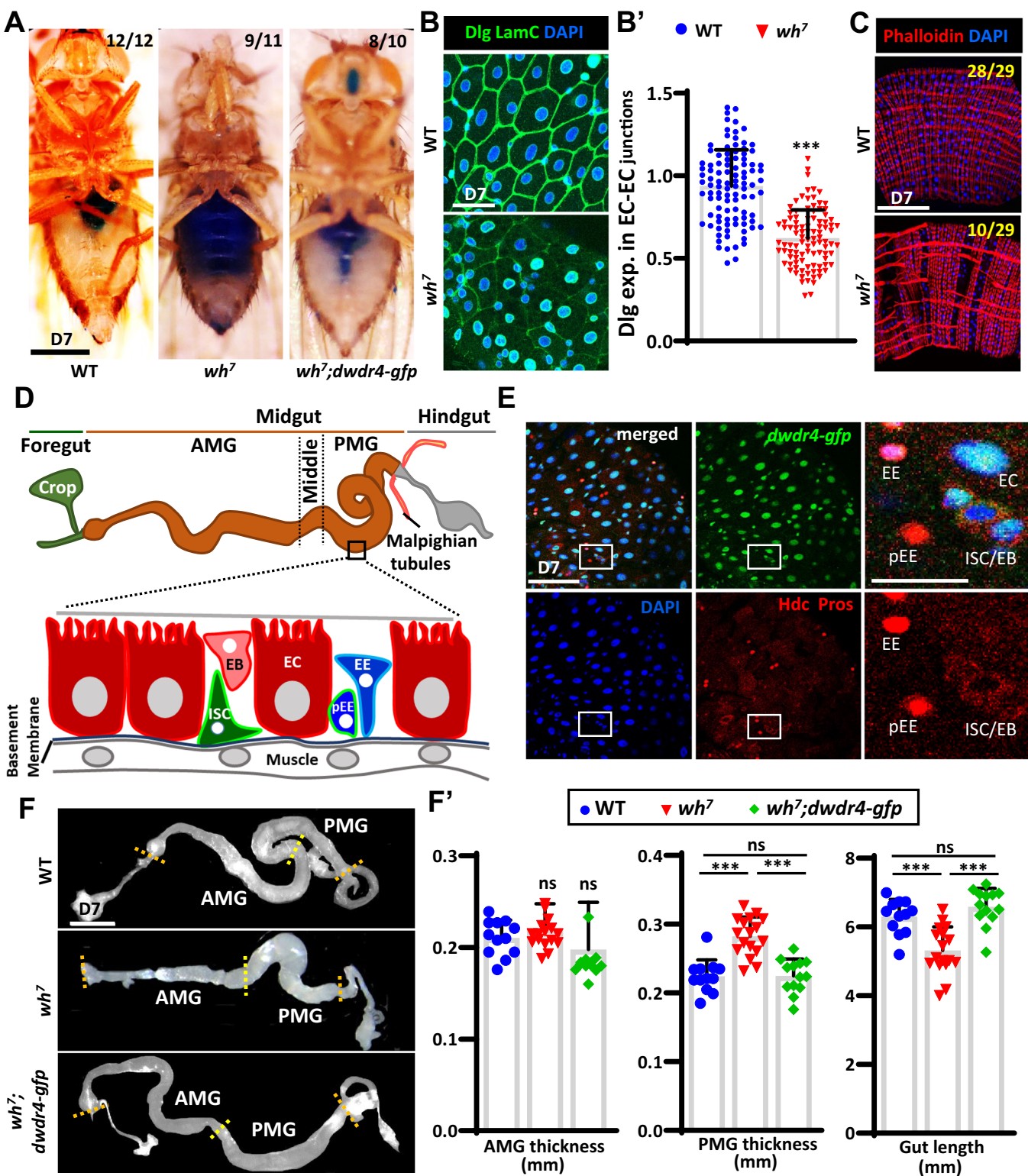

of cellular functions, including DNA replication (via Flap endonuclease 1) (Cheng et al, 2016), protein degradation (via the CUL4 E3 ligase) (Wang et al, 2017), and m⁷G tRNA modification (with methyltransferase Mettl1) (Alexandrov et al, 2002). This modification is known to maintain the pluripotency of mouse

embryonic stem cells and their differentiation into neural cells (Lin et al, 2018), as well as *Drosophila* spermatogenesis (Kaneko et al, 2024). Our prior work showed that *Drosophila* Wdr4 (dWdr4), previously known as Wuho (Wh), regulates germline stem cell proliferation with Mei-p26 (Rastegari et al, 2020). Interestingly,

**Figure 1. dWdr4 is expressed in the ISC lineage and its loss disrupts epithelial integrity and gut morphogenesis.**

(A) Smurf assay of wild-type (WT), *dwdr4* mutant ($wh^7$) mutant flies with or without a genomic *dwdr4-gfp* transgene. The fraction of flies exhibiting the depicted smurf pattern is provided in each panel. (B) Dlg expression (green) at EC-EC junctions in control and $wh^7$ mutant posterior midguts (PMGs). LamC (green) marks the nuclear envelope; DAPI (blue) labels nuclei. (B') Quantification of the Dlg-to-LamC intensity ratio in WT ($N = 24$) and $wh^7$ mutant ($N = 20$) ($n = 100$ cells) PMGs. Each dot represents a single cell. (C) F-actin organization in the PMG of the WT ($N = 11$) and $wh^7$ mutant ($N = 14$) PMGs visualized by Phalloidin (red) and DAPI (blue). Percentages of gut sections displaying the shown pattern are indicated. (D) Schematic of the adult *Drosophila* gut, including foregut, midgut, and hindgut. The foregut includes the crop, which stores food ingested by the fly. The Malpighian tubules (analogous to the mammalian kidney) connect at the midgut-hindgut junction. The midgut is the main site of digestion and is divided into the anterior midgut (AMG), middle midgut (MMG), and posterior midgut (PMG). The adult midgut contains intestinal stem cells (ISC), which reside above the basement membrane (blue line), and give rise to EBs (absorptive EC precursors), pEEs (secretory EE precursors), ECs, and EEs. Muscle lies beneath the epithelium. (E) Flies carrying *dwdr4-gfp* (green) ($N = 10$) are widely expressed in the ISC lineage. ISCs/EBs/pEEs, marked by cytoplasmic Hdc (red); pEEs and EEs, marked by nuclear Prospero (Pros, red). DAPI, blue. Enlarged views of the boxed region are shown to the right. (F, F') Brightfield images of dissected adult intestines showing the AMG and PMG in flies of the WT ($N = 12$), $wh^7$ mutant ($N = 17$), and $wh^7$ mutant with a genomic *dwdr4-gfp* transgene ($N = 13$). Each dot represents a single gut. The scale bar is 500 μm in (A, F), 50 μm in (B, C), and 50 μm for the low-magnification images and 25 μm for the enlarged views in (E). Dashed lines roughly define the region of AMG and PMG. Quantification (F') shows the width of the AMG, PMG, and total gut length in the indicated genotypes. Error bars, mean ± SD. Statistical significance in (B') was identified by the non-parametric Mann–Whitney test, and in (F') by the unpaired Student's *t* test. Statistically significant differences in (B') were observed between WT and $wh^7$ (***$P = < 0.0001$). (F') AMG thickness did not differ between WT and $wh^7$ (ns, not significant, $P = 0.254$) or WT and $wh^7$;*dwdr4-gfp* (ns, $P = 0.415$). In contrast, PMG thickness differed significantly between WT and $wh^7$ (***$P = < 0.0001$) and $wh^7$ and $wh^7$;*dwdr4-gfp* (***$P = < 0.0001$); with no difference between WT and $wh^7$;*dwdr4-gfp* (ns, $P = 0.957$). Gut length showed significant reduction between WT and $wh^7$ (***$P = < 0.0001$) and $wh^7$ and $wh^7$;*dwdr4-gfp* (***$P = < 0.0001$); with no difference between WT and $wh^7$;*dwdr4-gfp* (ns, $P = 0.183$). N, number of midguts; n, number of ISCs or ISC/EBs analyzed, unless otherwise stated. Source data are available online for this figure.

WDR4 has opposing roles in cancer; it promotes lung tumorigenesis (Regan et al, 2016), but correlates with improved survival in gastrointestinal cancers (Zeng et al, 2021), highlighting its critical role in the ISC and its lineage.

In this study, we show that dWdr4 cooperates with *Drosophila* Mettl1 (dMettl1) to maintain ribosome biogenesis in ISCs by mediating m7G modification of let-7 miRNA. Loss of this modification lowers *let-7* levels, activating TOR-JNK-dMyc signaling, which in turn elevates ribosome biogenesis, ISC overproliferation, and misdifferentiation, ultimately leading to gut dysplasia and a reduced lifespan. These findings reveal a novel mechanism by which Wdr4 safeguards ISC homeostasis and suggest a possible association with gastrointestinal cancer outcomes.

## Results

### dWdr4 maintains lifespan, locomotor activity, and intestinal function

Wdr4 is evolutionarily conserved, implicated in DNA replication, RNA modification, and protein degradation (Cheng et al, 2016; Lin et al, 2018; Wang et al, 2017), but its physiological roles remain unclear. We found that both male and female *dwdr4* mutant ($wh^7$) flies exhibited shortened lifespans and impaired climbing activity, which were rescued by reintroducing a genomic *dwdr4-gfp* transgene (Fig. EV1A–D) (Wu et al, 2006), indicating that dWdr4 is essential for maintaining organismal health. Given the tight link between gut function and lifespan (Biteau et al, 2010; Hodge et al, 2022), and the observation that higher WDR4 levels are associated with better survival in gastrointestinal cancers (Appendix Fig. S1) (Zeng et al, 2021), we examined the role of dWdr4 in intestinal health.

To assess gut barrier integrity, we performed the Smurf assay, in which flies were fed a non-absorbable blue dye, and the distribution of the dye was assessed (Rera et al, 2011). In one-week-old control female flies, the dye remained confined to the gut, whereas ~80% of *dwdr4* mutant females, but not males (Appendix Fig. S2), showed

systemic dye leakage. This defect was corrected by reintroducing a genomic *dwdr4-gfp* transgene. Consistently, levels of Discs large (Dlg), a septate junction protein essential for epithelial integrity (Tepass et al, 2001), were markedly reduced at EC-EC junctions in *dwdr4* mutant female intestines (Fig. 1B,B'). Additionally, phalloidin staining revealed disorganized, branched and discontinuous actin filaments in the visceral muscle layer surrounding the *dwdr4* mutant female intestine (*dwdr4* mutant: 34% of image sections: 0.6% of image sections (Fig. 1C), which likely impairs gut contraction (Min et al, 2017).

As the gut is essential for lipid uptake and transfer (Wit et al, 2022), we used Bodipy staining to assess lipid metabolism (Qiu and Simon, 2016). dwdr4 mutant intestines showed excessive lipid droplet accumulation (Fig. EV1E), similar to phenotypes from loss of the endocrine peptides Tachykinin (Tk) or *Drosophila* insulin-like peptide (Dilp3) in ECs, both regulated by the innate immune deficiency (IMD) pathway (Bolukbasi et al, 2017; Song et al, 2014). Consistently, RNA-seq revealed strong downregulation of *tk*, *TKR99D* (encoding TK receptor), *dilp3*, and multiple IMD pathway genes, along with many digestion-related genes, in one-week-old *dwdr4* mutant female intestines (Fig. EV1F; Appendix Table S1). These findings demonstrate that dWdr4 is essential for intestinal barrier integrity, visceral muscle organization, and gut endocrine signaling in females.

### dWdr4 is expressed in all intestinal cells and controls gut morphogenesis

We next examined dWdr4 expression in the ISC lineage using a genomic dwdr4-GFP transgene (Fig. 1D,E). Co-labeling with Headcase (Hdc, ISC/EB marker) (Resende et al, 2017), Prospero (Pros, pEE/EE marker) (Zeng and Hou, 2015), and nuclear size to identify ECs showed that dWdr4-GFP was broadly expressed in all intestinal cell types, including ISCs and EBs (Hdc+), pEEs (Pros+/ Hdc+), EEs (Pros+), and ECs (Pros−/Hdc− with large nuclei).

We proceeded to investigate the requirement of dWdr4 in the intestine. Strikingly, *dwdr4* mutant females, but not males (Appendix Fig. S3), displayed shorter intestines with thickened

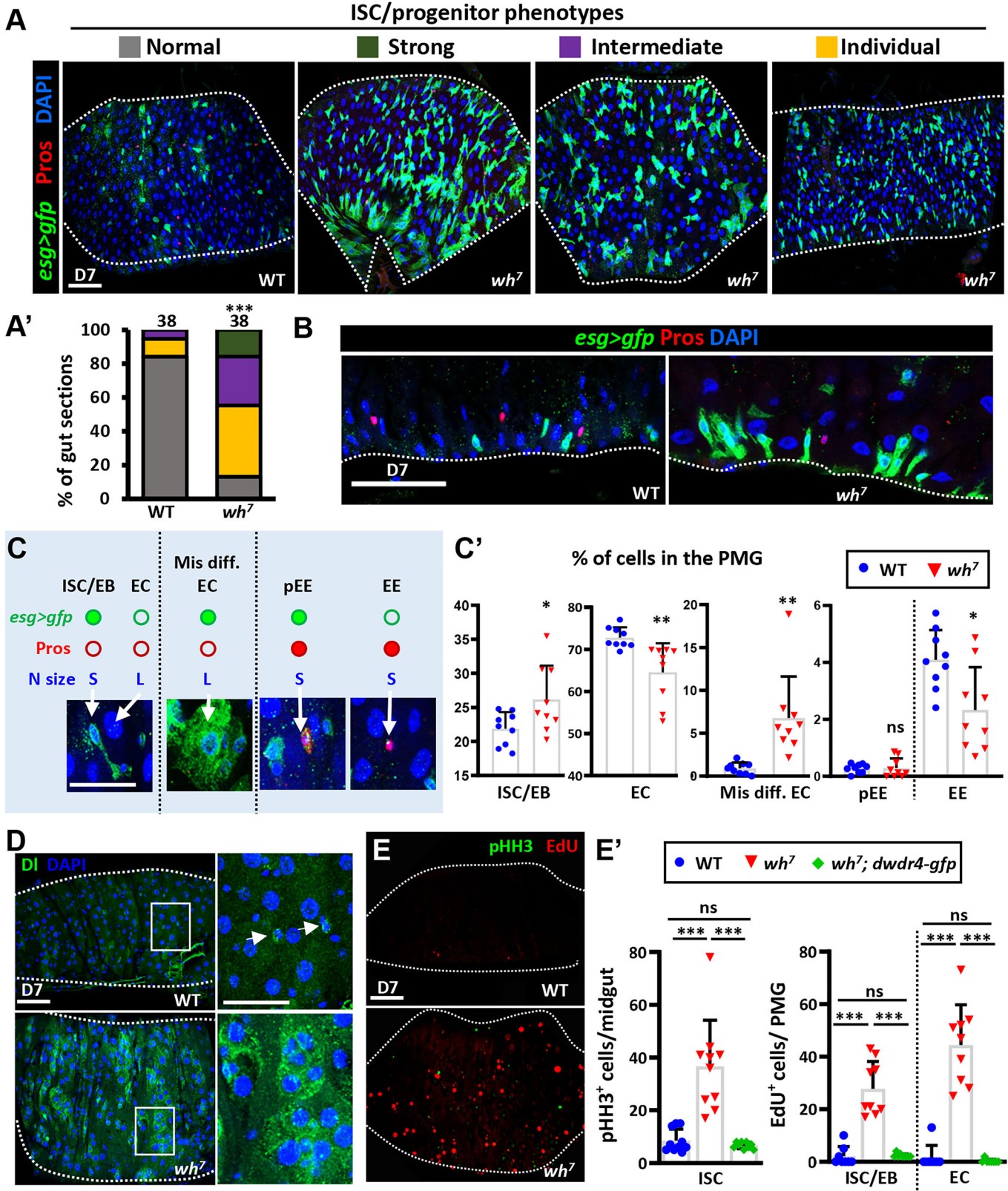

**Figure 2. dWdr4 loss leads to gut dysplasia.**

(A) Posterior midguts (PMGs) from wild-type (WT) and *dwdr4* mutant (*wh⁷*) females expressing *esg>gfp* (green, ISCs, EBs, and pEEs), stained for Pros (red, pEEs and EEs) and DAPI (blue, nuclei). *esg>gfp⁺* cell occupancy in WT and *wh⁷* mutant (*N* = 9) PMGs is classified as "normal," "strong," "intermediate," or "individual" based on the extent of GFP⁺ cell expansion. (A') Quantification of PMG phenotypes, color-coded to match the images. The number of gut sections analyzed is shown above each bar. (B) Apical-basal optical sections of a "strong" *wh⁷* mutant PMG show clustered ISCs/EBs (green) in contrast to the scattered distribution of ISCs/EBs in WT. (C) Summary of *esg>gfp* and Pros expression, along with nuclear size in the ISC lineage. Filled and open green or red circles indicate positive and negative expression of *esg > GFP* and Pros, respectively. Nuclear size is categorized as small (S) or large (L). Arrows indicate representative cell types. Mis diff. ECs, misdifferentiated ECs. DAPI, blue. (C') Quantification of indicated cell types in WT and *wh⁷* mutant PMGs. Each dot represents one PMG. (D) Dl expression (green, ISCs) in WT (*N* = 10) and *wh⁷* mutant (*N* = 11) PMGs. DAPI, blue. Boxed regions are magnified at right. Arrows in the WT inset indicate representative Dl-positive ISCs. (E) WT and *wh⁷* mutant PMGs labeled with PHH3 (green, mitosis marker) and EdU (red, DNA replication marker). (E') Quantification of PHH3⁺ cells per midgut (left) and EdU⁺ ISCs/EBs or ECs per PMG (right) in the WT (*N* = 10), *wh⁷* mutant (*N* = 10), and *wh⁷* mutant with a genomic *dwdr4-gfp* transgene (*N* = 9). Each dot in (C', E') represents one midgut. The scale bar in (A, D, E) is 50 μm; enlarged views of (C, D) are shown at 25 μm. (B) The scale bar is 100 μm. White dashed lines in (A, B, D, E) outline the edge of the gut. Bars represent the mean; error bars indicate standard deviation (SD). Statistically significant differences in (A') were analyzed by Chi-square between WT and *wh⁷* (***P = < 0.001), and in (C') were identified by Welch's *t* test, and (E') by the non-parametric Mann–Whitney test. (C') Significant differences were shown between WT and *wh⁷* mutant in ISC/EB (***P = 0.039), EC (**P = 0.007), misdifferentiated EC (**P = 0.007), and EE (*P = 0.01) cell populations, whereas pEE showed no significant difference (ns, P = 0.913). (E') The number of PHH3⁺ cells was significantly increased in *wh⁷* compared with WT (***P = < 0.0001) and reduced in *wh⁷;dwdr4-gfp* compared with *wh⁷* (***P = < 0.0001); with no significant difference between WT and *wh⁷;dwdr4-gfp* (ns, P = 0.8504). Similarly, EdU⁺ ISC/EB and EC cells showed significant differences between WT vs *wh⁷* and between *wh⁷* and *wh⁷;dwdr4-gfp* (***P = 0.0001); while no difference between WT and *wh⁷;dwdr4-gfp* (ns, P = 0.1765 for ISC/EB and P = 0.9999 for EC cells). N, number of midguts; n, number of ISCs or ISC/EBs analyzed, unless otherwise stated. Source data are available online for this figure.

PMGs (Fig. 1F,F'), the region containing the most active ISCs and functionally analogous to the mammalian small intestine (Marianes and Spradling, 2013). In contrast, the AMG appeared unaffected. These morphological abnormalities were fully rescued by reintroducing a genomic *dwdr4-gfp* transgene (Fig. 1F,F'). The thickness of the MMG was not measured because of its small size and limited visibility in gross dissections. Together, these results demonstrate that loss of *dwdr4* specifically disrupts female gut morphogenesis in a region-dependent manner.

## dWdr4 regulates ISC proliferation and differentiation to preserve intestinal tissue integrity

To elucidate the role of dWdr4 in the ISC lineage, we analyzed ISCs and progenitor populations (EBs and pEEs) in the PMG using *esg-GAL4*-driven GFP (*esg>gfp*). In control PMGs, ISCs/progenitors were sparsely and evenly distributed (classified as "normal"). In contrast, *dwdr4* mutant PMGs exhibited varying degrees of ISC/progenitor expansion, categorized as "individual," "intermediate," or "strong" based on the extent of GFP⁺ cell accumulation (Fig. 2A,A'), with severe phenotypes often localized near the MMG region. Moreover, in mutants exhibiting "strong" or "intermediate" phenotypes, ISC/progenitor cells often formed clusters (Fig. 2B).

To assess cell-type composition, we combined nuclear size with *esg>gfp* and Pros immunolabeling (Fig. 2C). We defined ISCs/EBs as GFP⁺ Pros⁻ with small nuclei; pEEs as GFP⁺ Pros⁺ with small nuclei; EEs as GFP⁻ Pros⁺ with small nuclei; and ECs as GFP⁻ Pros⁻ with large nuclei. *dwdr4* mutant PMGs showed a significant increase in ISCs/EBs and a modest decrease in ECs (Fig. 2C'). The proportion of pEEs was unchanged, and EEs, a minor cell population, were slightly reduced but not statistically significant (Fig. 2C'). Because pEEs represent a small and unaffected subset of the lineage, they were not further analyzed in this study.

We also identified ~7% of cells in *dwdr4* mutant PMGs resembling misdifferentiated ECs; these cells were located closer to the MMG, retained GFP expression, but exhibited irregular morphology and large nuclei characteristic of ECs (Fig. 2C,C'). These misdifferentiated ECs aberrantly expressed ISC markers,

including Notch ligand Delta (Dl) (Ohlstein and Spradling, 2007) and Hdc (Figs. 2D and EV2A). Thus, the presence of Dl and Hdc in ECs reflects true misdifferentiation rather than GFP perdurance.

To determine whether the dysplastic phenotype in *dwdr4* mutant PMGs was due to increased ISC proliferation, we labeled intestines with phosphorylated Histone H3 (pHH3, a mitosis marker) and EdU (a DNA replication marker) (Fig. 2E,E'). Compared to controls, *dwdr4* mutant PMGs showed a ~fourfold increase in pHH3⁺ mitotic cells (*dwdr4* mutants: 36.6 ± 17 vs. control: 8.5 ± 5), consistent with increased ISC proliferation, as ISCs are the primary cell type known to undergo division. EdU incorporation further revealed a ~14-fold increase in replicating ISCs/EBs (*dwdr4* mutants: 36.6 ± 17, *n* = 10; control: 1.7 ± 4) and a ~40-fold increase in replicating ECs (*dwdr4* mutants: 40 ± 15; control: 1.3 ± 5), indicating both enhanced ISC activity and EC endoreplication. These phenotypes were fully rescued by introducing a genomic *dwdr4-gfp* transgene (Fig. 2E'). Together, these results demonstrate that dWdr4 is required to restrain ISC proliferation and ensure proper differentiation.

## dWdr4 cell autonomously controls ISC division and proper differentiation to support intestinal homeostasis and lifespan

To test whether dWdr4 directly regulates ISC division, we generated *dwdr4* mutant ISCs in the gut using the mosaic analysis with a repressible cell marker (MARCM) system (Wu and Luo, 2006). In this system, heat shock-induced Flippase (FLP) triggers recombination at FRT sites in a dividing ISC, producing one GFP⁺ daughter cell that is homozygous for the *dwdr4* mutation and one unlabeled homozygous wild-type twin. All non-recombined cells remain heterozygous (*dwdr4/+*), forming the background tissue. This setup allows direct comparison of homozygous mutant clones with heterozygous tissue within the same gut (Fig. 3A).

We quantified clone numbers per gut and cells per clone at 3, 7, and 14 days after clone induction (aci). At 3 days aci, clone numbers were similar between control and *dwdr4* heterozygous intestines (Fig. EV2B), indicating that loss of a single *dwdr4* copy does not impair ISC division. Likewise, clone numbers in

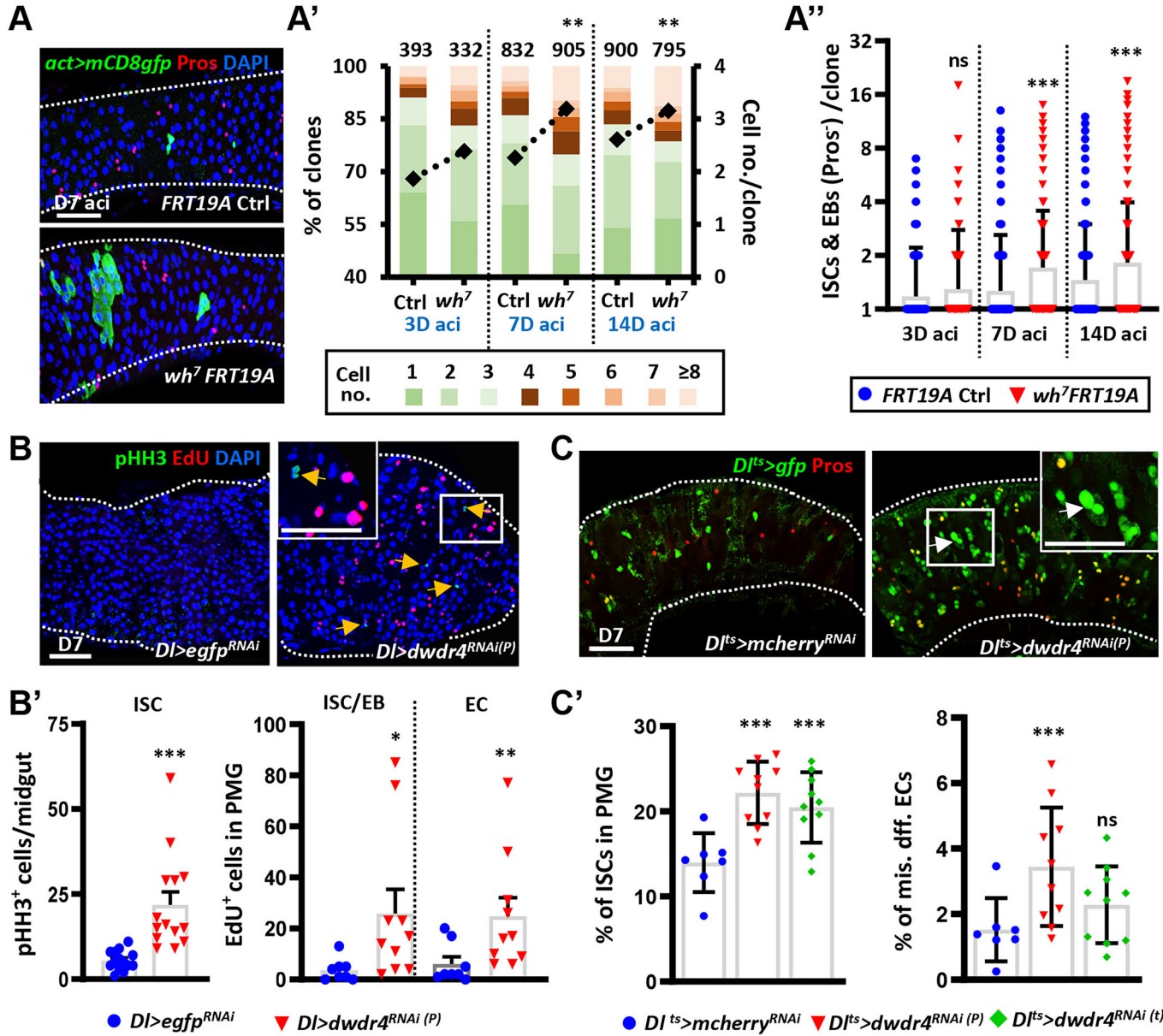

**Figure 3. dWdr4 cell-autonomously controls ISC proliferation and differentiation.**

(A) Mosaic control (*FRT19A* ctrl) and *dwdr4* (*wh[7] FRT19A*) heterozygous mutant posterior midguts (PMGs) at 7 days after clone induction (aci), labeled with GFP (green, mutant cells), Pros (red, pEEs or EEs), and DAPI (blue, DNA). (A') Percentage of clones in *FRT19A* control (3D: N = 27, 7D: N = 30, 14D: N = 34) and *wh[7]FRT19A* heterozygous mutant posterior PMGs (3D: N = 29, 7D: N = 28, 14D: N = 28) bearing the indicated cell number (no.) at 3, 7, and 14 days aci. The right Y axis shows the average cell number per clone. (A") Number of ISCs/EBs (Pros-negative cells with small nucleus) per clone in the indicated genotypes. (B) *Dl>dwdr4[RNAi(P)]* PMG (N = 14) displays more PHH3+ (green, mitotic marker, indicated by arrows) and EdU+ cells (red, DNA replication marker), as compared to control (*Dl>egfp[RNAi]*) (N = 14). DAPI, Blue. (B') Number of PHH3+ cells per midgut, and number of ISC/EB or ECs positive for EdU per PMG with indicated genotypes. (C) *Dl[ts]>dwdr4[RNAi(P)]* (N = 10) PMG shows more ISCs (*Dl[ts]>gfp*, green) compared to *Dl[ts]>mcherry[RNAi]* (N = 7) Pros, red, pEEs or EEs. (C') Percentage of ISCs and misdifferentiated (mis. diff.) ECs (*Dl>gfp*-positive cells with big nuclei, indicated by white arrows) in PMGs with indicated genotypes. Insets show enlarged views of the boxed regions in (B, C). Each dot in (A") represents one clone, and in (B', C') represents one midgut. The scale bar in (A, B, C) is 50 μm; enlarged views of (B, C) are shown at 25 μm. White dashed lines in (A, B, C) outline the edge of the gut. Bars represent the mean; error bars indicate standard deviation (SD). Statistical significance in (A') was analyzed by Chi-square, and in (A", B', C') by the non-parametric Mann–Whitney test. (A') showed significant differences between *FRT19A* ctrl and *wh[7] FRT19A* in ISC/EB cells after 7D aci (**P = < 0.0001), and 14D aci (***P = 0.006), whereas 3D aci showed no significant difference (ns, P = 0.12). (A") Significant differences were shown between *FRT19A* ctrl and *wh[7] FRT19A* in ISC/EB cells after 7D aci (***P = < 0.0001), and 14D aci (***P = 0.0006), whereas 3D aci showed no significant difference (ns, P = 0.6135). (B') PHH3+ cells were significantly increased in *Dl>dwdr4[RNAi(P)]* compared with *Dl>egfp[RNAi]* (**P = < 0.0001; EdU+ ISC/EB (*P = 0.010) and EC cells (**P = 0.009) showed significant increase. (C') % of ISCs in PMGs cells were significantly increased in *Dl[ts]>dwdr4[RNAi(P)]* (***P = 0.0007) and *Dl[ts]>dwdr4[RNAi(t)]* (***P = 0.0097) compared with *Dl[ts]>mcherry[RNAi]*, while the misdifferentiated ECs differed in *Dl[ts]>mcherry[RNAi]* and *Dl[ts]>dwdr4[RNAi(P)]* (**P = 0.0097) without any difference in *Dl[ts]>dwdr4[RNAi(t)]* (ns, P = 0.474). N, number of midguts; n, number of ISCs or ISCs/EBs analyzed, unless otherwise stated. Source data are available online for this figure.

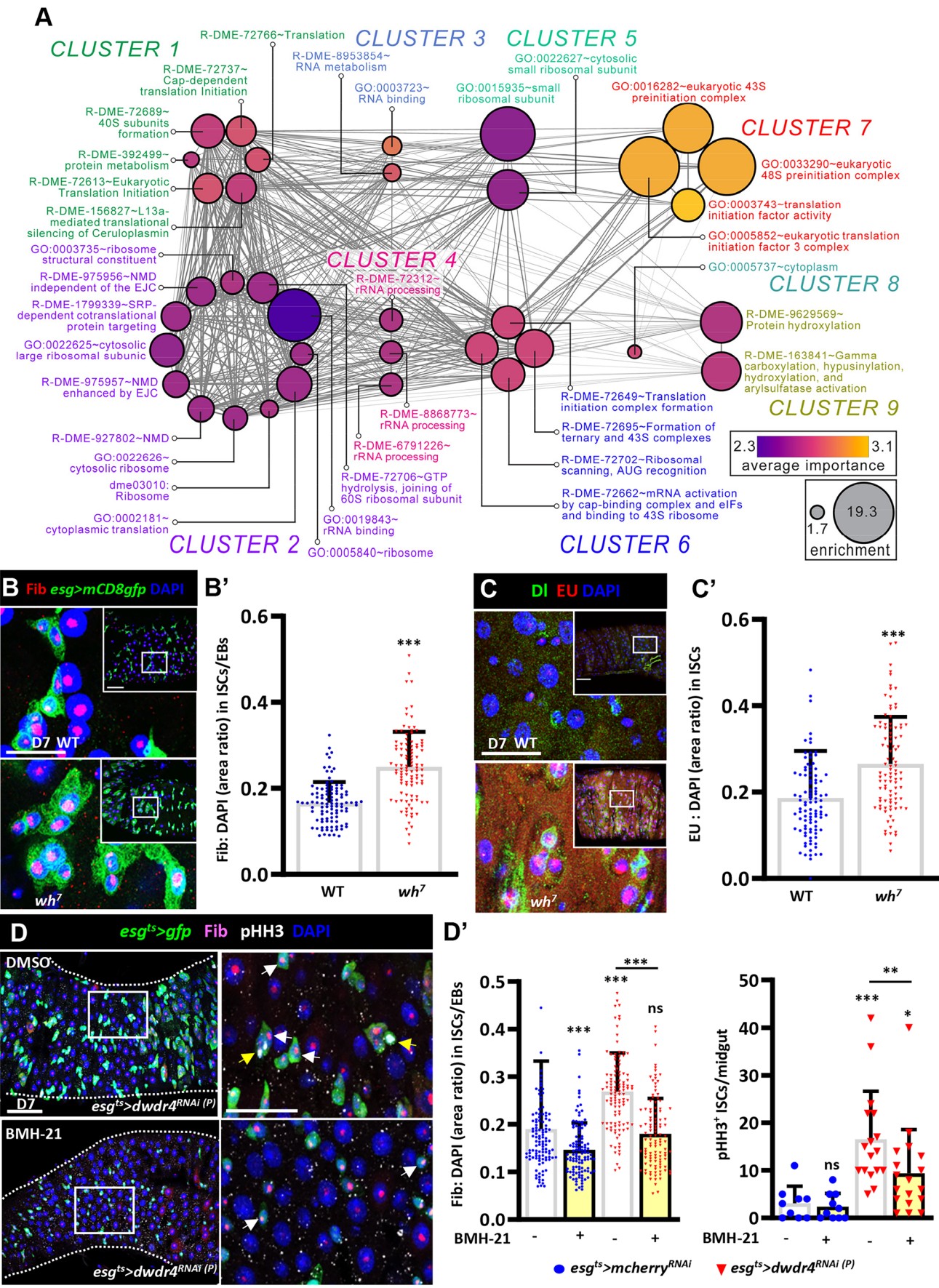

**Figure 4. dWdr4 depletion increases ribosome biogenesis in the intestine.**

(A) A total of 166 genes that were predictive of *dwdr4* expression belonged to functional categories represented by nodes of the network. Each node represents a GO term or pathway, with node size indicating enrichment significance (larger circles = higher enrichment) and color representing average importance (purple to orange scale; see legend). Edges represent shared genes or functional overlap between nodes. Clusters are labeled and color-coded by dominant biological themes, including translation initiation (Clusters 1 and 7), RNA processing and metabolism (Clusters 3 and 4), ribosomal subunit assembly (Cluster 5), and mRNA surveillance or decay (Clusters 2 and 6). (B) Wild-type (WT) ($N = 16$) and *dwdr4* mutant (*wh$^7$*) mutant ($N = 19$) ($n = 102$ ISC/EB cells) posterior midguts (PMGs) carrying *esg>mCD8gfp* (green, ISCs/EBs), stained for Fibrillarin (Fib) (red, nucleoli) and DAPI (blue, nuclei). (B') Quantification of nucleolar (Fib) to nuclear (DAPI) area ratio in GFP-positive cells with small nuclei in the indicated genotypes. (C) *wh$^7$* mutant PMG labeled with ethylene uridine (EU) (red, nascent RNA transcripts), Dl (green, ISCs), and DAPI (blue). The image shown in this fig. was obtained from the same experimental sample as Fig. 2D. (C') Quantification of the EU to nuclear (DAPI) area ratio in GFP-positive cells with small nuclei from WT ($N = 11$) and *wh$^7$* mutant ($N = 10$) ($n = 100$ ISC cells) PMGs. (D) *esg$^{ts}$>dwdr4$^{RNAi(P)}$* PMGs bearing *esg$^{ts}$>gfp* (green, ISCs/EBs) with DMSO or BMH-21 treatment (ribosome biogenesis inhibitor), and stained for PHH3 (gray, mitosis marker), and Fibrillarin (Fib) (magenta, nucleoli) and DAPI (blue). Yellow arrows indicate representative PHH3$^+$ cells, and white arrows indicate representative ISCs with Fib labels. (D') Quantification of nucleolar (Fib) to nuclear (DAPI) area ratio in GFP-positive cells with small nuclei and of PHH3$^+$ ISCs per midgut and in the *esg$^{ts}$>dwdr4$^{RNAi(P)}$* (DMSO: $N = 17$, BMH-21: $N = 18$) and *esg$^{ts}$>mcherry$^{RNAi}$* flies (DMSO: $N = 9$, BMH-21: $N = 10$) ($n = 111$ ISC/EB cells) with and without BMH-21 treatment. Yellow arrows in D indicate representative PHH3$^+$ ISCs, and white arrows mark representative ISC/EB nucleoli. Enlarged views of the boxed regions are shown in (B, C, insets) and to the right in (D). Each dot in (B', C', D', left graph) represents a single ISC or ISC/EB, and each dot in (D', right graph) represents one midgut. The low magnification and enlarged views of (B, C) are shown at 50 and 25 µm, and in (D) are 50 µm. White dashed lines in (D) outline the edge of the gut. Bars represent the mean; error bars indicate standard deviation (SD). Statistically significant differences in (B', C', D') were identified by a non-parametric Mann–Whitney *U* test. The nucleolar size was significantly increased in *wh$^7$* compared with WT in (B', C') (***$P = 0.0001$). (D', left graph) Nucleolar size differed with and without BMH-21 in *esg$^{ts}$>mcherry$^{RNAi}$* (***$P = 0.0003$), and in *esg$^{ts}$>dwdr4$^{RNAi(P)}$* with and without BMH-21 (***$P = <0.0001$). No difference was observed between *esg$^{ts}$>mcherry$^{RNAi}$* and *esg$^{ts}$>dwdr4$^{RNAi(P)}$* upon BMH-21 treatment (ns, $P = 0.8117$), whereas a significant difference was detected without BMH-21 (***$P = <0.0001$). (D', right graph) pHH3 cells were unchanged by BMH-21 in *esg$^{ts}$>mcherry$^{RNAi}$* (ns, $P = 0.7601$), but differed significantly between *esg$^{ts}$>mcherry$^{RNAi}$* and *esg$^{ts}$>dwdr4$^{RNAi(P)}$* with (*$P = 0.013$) and without BMH-21 (***$P = <0.0001$); a difference was seen in *esg$^{ts}$>dwdr4$^{RNAi(P)}$* between with and without BMH-21 treatment (**$P = 0.0075$). *N*, number of midguts; *n*, number of ISCs or ISC/EBs analyzed, unless otherwise stated. Source data are available online for this figure.

heterozygous guts remained stable from 3 to 14 days aci, suggesting that *dWdr4* is dispensable for ISC maintenance (Fig. EV2B). In contrast, homozygous *dwdr4* mutant clones expanded significantly over time (Fig. 3A'), containing more ISCs/EBs, identified by their small nuclei and absence of Pros (a pEE/EE marker) (Fig. 3A"). Notably, approximately 54-60% of clones in both control and *dwdr4* heterozygous guts consisted of a single cell, consistent with a previous report (Neophytou and Pitsouli, 2022). This likely reflects GFP labeling of an EB or pEE, the immediate daughter of a dividing ISC, after recombination. Together, these results demonstrate that dWdr4 controls ISC division in a cell-autonomous manner.

To further validate the function of dWdr4 in different intestinal cell types, we performed targeted knockdown using cell-type-specific drivers: Dl-GAL4 (ISCs), esg-GAL4 (ISCs/EBs), Su(H)GBE-GAL4 (EBs), and NP1-GAL4 (ECs). Knockdown with either Dl-GAL4 or esg-GAL4, or with the addition of temperature control using GAL80$^{ts}$ (Dl$^{ts}$ or esg$^{ts}$) to minimize developmental effects, produced similar phenotypes, including ISC overproliferation (Fig. 3B,B'), increased ISC numbers, and misdifferentiated ECs (Figs. 3C,C' and EV2C,D'). These misdifferentiated ECs expressed Dl (see Fig. 2D) and Su(H)GBE-lacZ (Fig. EV2B), an EB marker (Ohlstein and Spradling, 2007).

In contrast, *dwdr4* knockdown in EBs or ECs caused no obvious defects, except that EC-specific knockdown led to increased EC endoreplication (Fig. EV2E,F'). Given the similar severity of cellular defects observed with Dl-GAL4 and esg-GAL4 drivers, we used either driver, with or without GAL80$^{ts}$-mediated temporal control, for subsequent analyses of dWdr4 function in ISC.

Notably, knocking down *dwdr4* in ISCs, or in visceral muscles, using How-GAL4 (Jiang and Edgar, 2009), did not induce intestinal leakage (Fig. EV2G), suggesting that these phenotypes arise only when dWdr4 is lost throughout the entire intestinal epithelium. Furthermore, *esg$^{ts}$>dwdr4$^{RNAi (P)}$* flies showed a modest reduction in lifespan compared to controls (Fig. EV2H), indicating that dWdr4 function in ISCs/EBs is essential for intestinal homeostasis and contributes to organismal longevity.

## dWdr4 regulates ribosome biogenesis in ISCs

Genes associated with a particular pathway are often coregulated/coexpressed due to the activity of shared upstream transcription factors. We used this notion of "guilt by association" [reference: PMID 16162296] to uncover the molecular pathways associated with dWdr4 function. We analyzed publicly available RNA-seq datasets in FlyBase and identified 3,886 genes whose expression correlated with that of *dwdr4* (Dataset EV1). To refine this list, we applied the Boruta R package to ensure predictive accuracy (Kursa and Rudnicki, 2010), which identified 166 genes (Dataset EV1). These genes were subsequently grouped into nine functional clusters (Fig. 4A; Dataset EV1). Notably, Cluster 7, containing components of the eukaryotic 43S and 48S preinitiation complexes and translation initiation factors, displayed the highest classifier importance, suggesting a strong link between dWdr4 and early steps of translational regulation. Additional Clusters (1, 2, 4, 5, and 6) were enriched for genes involved in rRNA processing, ribosome assembly, and translation initiation. In contrast, Cluster 9, related to protein hydroxylation and post-translational modification, may represent an indirect role of dWdr4. Notably, RNA-seq analysis revealed that several ribosomal genes were upregulated upon dWdr4 depletion (Appendix Table S2). Together, these results highlight a central role for dWdr4 in ribosome biogenesis and translational control.

We next examined ribosome biogenesis by labeling Fibrillarin to mark nucleoli (Fomproix et al, 1998). Since ribosome biogenesis occurs in the nucleolus, nucleolar size is widely used as a proxy for ribosome biogenesis activity (Ma et al, 2016). In *dwdr4* mutant PMGs, the nucleolar-to-nuclear area ratio in ISCs increased by ~1.5-fold compared to controls (mutant: $0.25 \pm 0.08$ vs control: $0.17 \pm 0.05$) (Fig. 4B,B'). Consistently, *dwdr4* mutant ISCs and their derived EBs, generated by FLP-mediated recombination and marked by GFP, also exhibited larger nucleoli relative to controls (Fig. EV3A,A'). To directly assess ribosome RNA (rRNA) synthesis (Dvořáčková and Fajkus, 2023; Mikhaleva et al, 2019), we

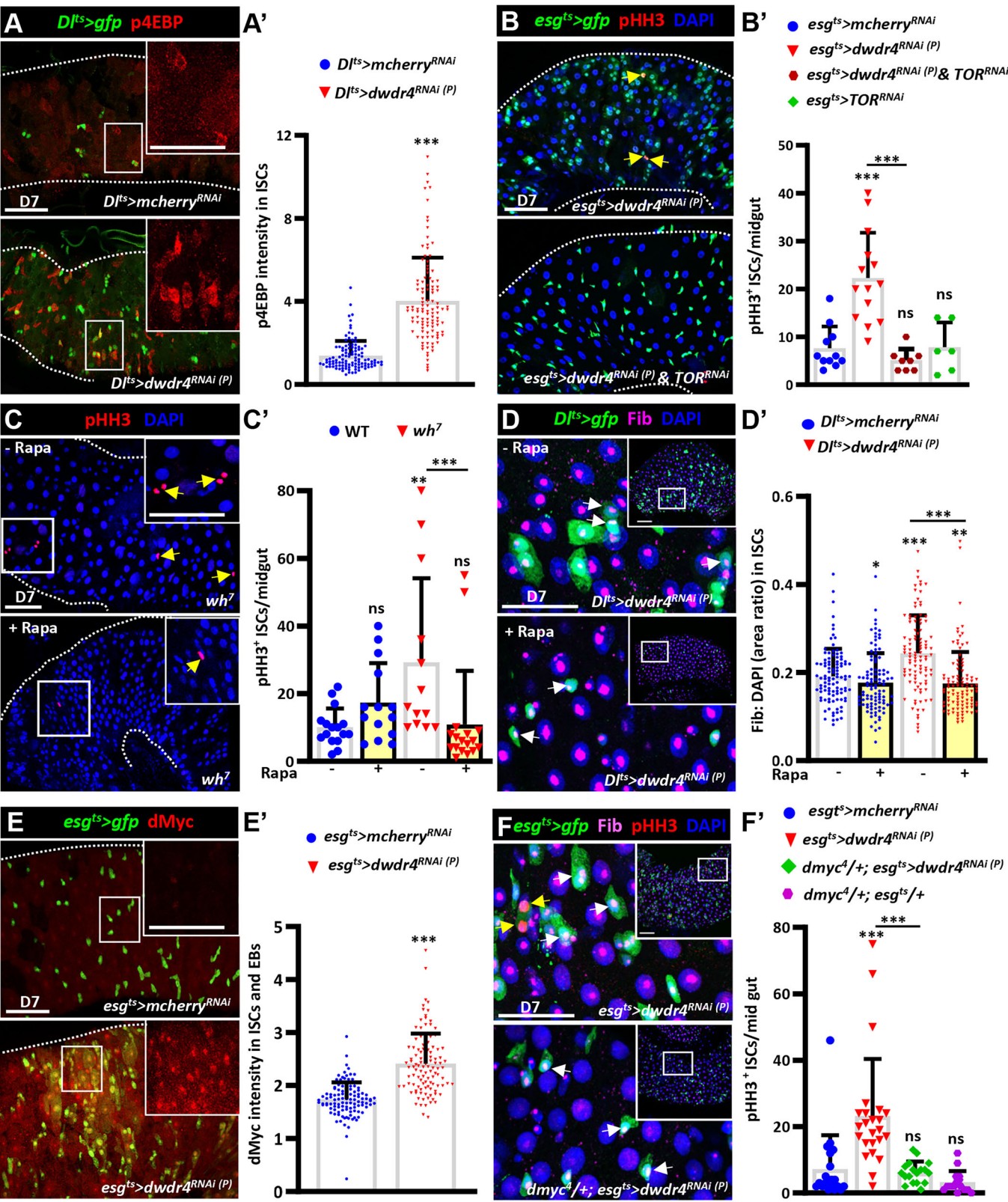

◄    **Figure 5.  dWdr4 regulates ribosome biogenesis in ISCs via the TOR and dMyc.**

(A) Posterior midguts (PMGs) from control ($Dl^{ts}$>$mcherry^{RNAi}$) and dWdr4-depleted females bearing $Dl^{ts}$>$gfp$ (green, ISCs), stained for p4EBP (red, TOR signaling reporter) and DAPI (blue, nuclei). (A') Quantification of average p4EBP intensity in ISCs in the $Dl^{ts}$>$mcherry^{RNAi}$ and $Dl^{ts}$>$dwdr4^{RNAi(p)}$ ($N = 10$, $n = 110$ ISC cells). (B) PMGs from the indicated genotypes bearing $esg^{ts}$>$gfp$ (green, ISCs/EBs), stained for PHH3 (red, mitotic maker) and DAPI (blue). (B') Quantification of PHH3+ ISCs per midgut in the $esg^{ts}$>$mcherry^{RNAi(P)}$ ($N = 11$), $esg^{ts}$>$dwdr4^{RNAi(P)}$ ($N = 14$), $esg^{ts}$>$dwdr4^{RNAi(P)}$ with $TOR^{RNAi}$ ($N = 8$) and $TOR^{RNAi}$ alone ($N = 6$) flies. (C) $dwdr4$ mutant ($wh^7$) PMGs with or without rapamycin treatment, stained for PHH3 (red, mitotic marker) and DAPI (blue). (C') Quantification of PHH3+ ISCs per midgut in the $wh^7$ mutant (dH$_2$O: $N = 13$; Rapa: $N = 17$) and WT flies (dH$_2$O: $N = 16$; Rapa: $N = 14$). (D) $Dl^{ts}$>$dwdr4^{RNAi(P)}$ PMGs bearing $Dl^{ts}$>$gfp$ (green, ISCs) with or without rapa treatment, stained for Fibrillarin (Fib; magenta, nucleoli) and DAPI (blue). (D') Quantification of nucleolar (Fib) to nuclear (DAPI) area ratio in ISCs from $esg^{ts}$>$dwdr4^{RNAi(P)}$ (dH$_2$O: $N = 12$; Rapa: $N = 16$) and $esg^{ts}$>$mcherry^{RNAi}$ flies (dH$_2$O: $N = 16$, Rapa: $N = 14$) ($n = 101$ ISC/EB cells). (E) Control ($N = 18$) and $esg^{ts}$>$dwdr4^{RNAi(P)}$ ($N = 17$) ($n = 110$ ISC/EB cells) PMGs bearing $esg^{ts}$>$gfp$ (green, ISCs/EBs), stained for dMyc (red, ribosome biogenesis regulator). (E') Quantification of average dMyc intensity in ISCs/EBs in the indicated genotypes. (F) $esg^{ts}$>$dwdr4^{RNAi(P)}$ PMGs with or without a copy of the $dmyc^4$ mutant allele bearing $esg^{ts}$>$gfp$ (green), Fib (magenta), PHH3 (red), and DAPI (blue). (F') Quantification of PHH3+ ISCs per midgut in the $esg^{ts}$>$mcherry^{RNAi(P)}$ ($N = 20$), $esg^{ts}$>$dwdr4^{RNAi(P)}$ ($N = 24$), $esg^{ts}$>$dwdr4^{RNAi(P)}$ with $dmyc^4$ ($N = 18$) and $dmyc^4$ mutant alone ($N = 15$). Yellow arrows in (B, C, F) indicate representative PHH3+ ISCs, and white arrows in (D, F) indicate representative ISC nucleoli. Insets in (A, C, E) show enlargements of boxed regions; (D, F) show enlargements of boxed regions within insets. Each dot in (A', D', E') represents a single ISC or ISC/EB, and in (B', C', F') represents one midgut. The scale bars of (A, C–F) are shown at 50 μm (overview) and 25 μm (enlarged views); in (B), it is 50 μm. White dashed lines in (A, B, C, E) outline the edge of the gut. Bars represent the mean; error bars indicate standard deviation (SD). Statistical analysis (A', B', C', D', E', F') was performed using the non-parametric Mann–Whitney $U$ test. (A') The p4EBP levels were increased in $Dl^{ts}$>$dwdr4^{RNAi(P)}$ compared to $Dl^{ts}$>$mcherry^{RNAi}$ (***$P = $ < 0.0001). (B') PHH3+ cells, differed between $esg^{ts}$>$dwdr4^{RNAi(P)}$ compared to $esg^{ts}$>$mcherry^{RNAi}$ without $TOR^{RNAi}$ (***$P = $ < 0.0001) but not with $TOR^{RNAi}$ (ns, $P = 0.220$); whereas $TOR^{RNAi}$ alone had no effec$^{ts}$ (ns, $P = 0.903$). (C') PHH3+ cells were unchanged by rapamycin in WT (ns, $P = 0.08$), but differed between WT and $wh^7$ with (ns, $P = 0.06$) and without rapamycin (**$P = 0.0019$); in $wh^7$, rapamycin significantly reduced PHH3+ cells (***$P = $ < 0.0001). In D' nucleolar size differed in $Dl^{ts}$>$mcherry^{RNAi}$ and $Dl^{ts}$>$dwdr4^{RNAi(P)}$ with and without rapamycin (*$P = 0.04$ and ***$P = $ < 0.0001), with a modest difference between the genotypes upon rapamycin (**$P = 0.005$), whereas a significant difference was observed without rapamycin (***$P = $ < 0.0001). (E') The dMyc levels were increased in $esg^{ts}$>$dwdr4^{RNAi(P)}$ compared to $esg^{ts}$>$mcherry^{RNAi}$ (***$P = $ < 0.0001). (F') PHH3+ cells, there were differences in $esg^{ts}$>$dwdr4^{RNAi(P)}$ compared to $esg^{ts}$>$mcherry^{RNAi(P)}$ without $dmyc^4$ (***$P = $ < 0.0001) but not with $dmyc^4$ (ns, $P = 0.228$); whereas $dmyc^4$ alone had no effects (ns, $P = 0.127$). $N$, number of midguts; $n$, number of ISCs or ISC/EBs analyzed, unless otherwise stated. Source data are available online for this figure.

performed a 5-ethynyl uridine (EU) incorporation assay. EU signals were significantly stronger and EU-labeled nucleoli appeared larger in ISCs from $dwdr4$ mutant PMGs than in controls (Fig. 4C,C').

To test whether increased ribosome biogenesis contributes to the gut hyperplasia observed upon dWdr4 depletion, we treated flies with BMH-21, a small molecule widely used in cultured cells, flies, and mice to suppress rRNA transcription by inhibiting RNA polymerase I (Cho et al, 2025; Gui et al, 2023). Both control and dWdr4-depleted flies treated with 5 μM BMH-21 for 3 days showed a reduced nucleolar-to-nuclear area ratio in ISCs/EBs (Fig. 4D,D'), indicating suppression of ribosome biogenesis. Furthermore, ISC overexpression and ISC/EB expansion were markedly reduced in dWdr4-depleted flies upon BMH-21 treatment (Fig. 4D,D''). These results identify dWdr4 as a key regulator of ribosome biogenesis in ISCs, with its loss triggering elevated ribosome biogenesis, ISC overproliferation, and ISC/EB expansion.

## dWdr4 regulates ribosome biogenesis through TOR signaling and dMyc

Both Target of Rapamycin (TOR) and c-Myc are key master regulators of ribosome biogenesis (Chaillou et al, 2014); TOR regulates the translation of ribosomal proteins, while c-Myc activates the transcription of multiple ribosomal protein genes. We first examined TOR signaling in the PMGs bearing $dwdr4$-knockdown ISCs by tracking the expression of phosphorylated Eukaryotic initiation factor 4E-binding protein (p4EBP), a downstream effector of TOR signaling (Miron et al, 2003). p4EBP expression was dramatically increased in $dwdr4$-knockdown ISCs compared to controls (Fig. 5A,A'). Depletion of TOR in $dwdr4$-knockdown ISCs suppressed ISC overproliferation, whereas TOR depletion alone did not affect ISC proliferation (Fig. 5B,B'). Consistently, treating $dwdr4$ mutant flies or flies bearing $dwdr4$ knockdown ISCs with rapamycin, a TOR inhibitor (Lamming,

2016), suppressed ISC proliferation, reduced ISC/EB clustering, and decreased nucleolar enlargement in ISCs/EBs (Figs. 5C,D' and EV3B,B'). These results indicate that TOR activation contributes to elevated ribosome biogenesis and ISC overproliferation upon dWdr4 depletion.

We next asked whether dWdr4 depletion also promotes ribosome biogenesis through c-Myc. *Drosophila* c-Myc (dMyc) expression was strongly elevated throughout the ISC lineage (Fig. 5E,E'). Furthermore, decreasing dMyc expression in $esg^{ts}$>$dwdr4^{RNAi}$ flies by removing a copy of $dmyc$ suppressed both ISC overexpression and nucleolar enlargement in ISCs/EBs, whereas reducing dMyc alone had no effect (Figs. 5F,F' and EV3C,C'). These results indicate that elevated ribosome biogenesis upon dWdr4 depletion requires dMyc. Together, our data demonstrate that loss of dWdr4 promotes ribosome biogenesis via both TOR signaling and dMyc activation.

## Elevated ribosome biogenesis caused by dWdr4 depletion drives gut aging via JNK signaling activation

dWdr4-deficient guts exhibit features resembling aged or damaged intestines, including ISC hyperproliferation, ISC/EB expansion, and EC misdifferentiation (Biteau et al, 2008). These features are commonly linked to age-dependent oxidative stress and Jun N-terminal kinase (JNK) signaling (Herrera and Bach, 2021). In $dwdr4$-deficient ISCs, both TOR signaling and dMyc are upregulated, pathways known to elevate cellular ROS levels (Bonawitz et al, 2007; Vafa et al, 2002). Consistently, RNA-seq analysis revealed significant upregulation of genes involved in redox activity and oxidative stress response in $dwdr4$ mutant guts (Fig. EV4A; Appendix Table S3). Direct detection of ROS, by dihydroethidium (DHE) staining (Owusu-Ansah et al, 2008), confirmed markedly elevated ROS levels in $dwdr4$ mutant PMGs (Fig. EV4B,B'). As expected, JNK signaling activity, monitored by phosphorylated JNK (pJNK) (Cordero et al, 2012) and the $puc$-$lacZ$ reporter (Martín-

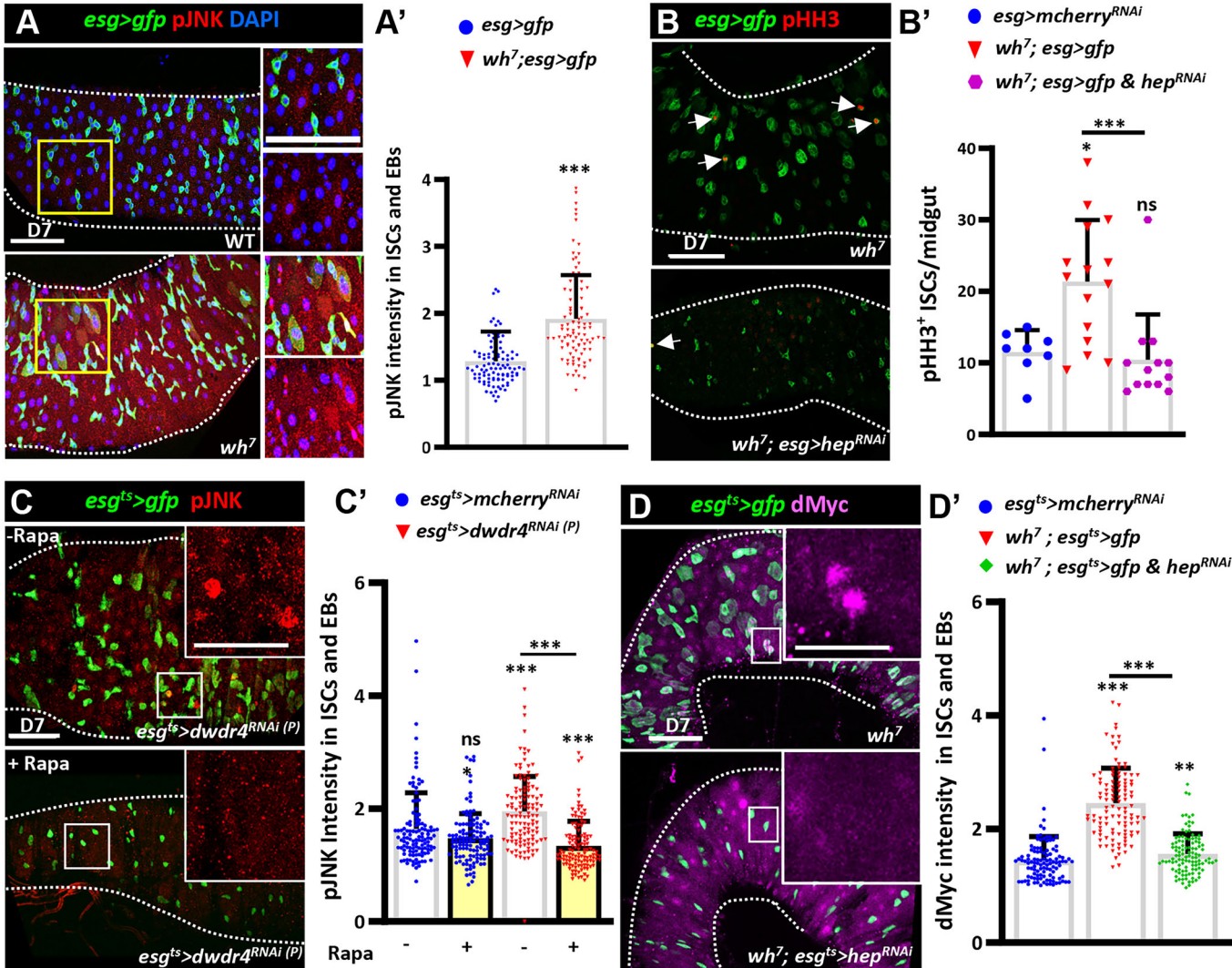

**Figure 6. dWdr4 loss causes ISC overproliferation and misdifferentiation via the TOR-JNK-dMyc regulatory axis.**

(A) Posterior midguts (PMGs) from wild-type (WT) (*N* = 8) and *dwdr4* (*wh⁷*) mutant (*N* = 12) (*n* = 100 ISC/EB cells) females bearing *esg^ts^>gfp* (green, ISCs/EBs), stained for pJNK (red, JNK signaling reporter) and DAPI (blue, nuclei). (A') Quantification of average pJNK intensity in ISCs/EBs of the indicated genotypes. (B) PMGs from *wh⁷* mutant bearing *esg>gfp* (green, ISCs/EBs) with or without Hep depletion, stained for PHH3 (red, mitotic marker). Arrows indicate representative PHH3⁺ ISCs. (B') Quantification of PHH3⁺ ISCs per midgut in *esg>mcherry^RNAi(P)^* (*N* = 8), *wh⁷* mutant with *esg>gfp* (*N* = 15), and *wh⁷* mutant with *esg>gfp* and *hep^RNAi^* (*N* = 13). (C) *esg^ts^>dwdr4^RNAi(P)^* PMGs bearing *esg^ts^>gfp* (green, ISCs/EBs) with or without rapamycin (rapa) treatment, stained for pJNK (red). (C') Quantification of average pJNK intensity in ISCs/EBs from *esg^ts^>dwdr4^RNAi(P)^* (dH₂O: *N* = 25, Rapa: *N* = 30) and *esg^ts^>mcherry^RNAi^* flies (dH₂O: *N* = 22, Rapa: *N* = 15) (*n* = 120 ISC/EB cells). (D) PMGs from *wh⁷* mutant bearing *esg^ts^>gfp* (green, ISCs/EBs) with or without Hep depletion, stained for dMyc (magenta, ribosome biogenesis regulator) and DAPI (blue). (D') Quantification of average dMyc intensity in the *esg^ts^>mcherry^RNAi(P)^* (*N* = 11), *wh⁷* mutant with *esg^ts^>gfp* (*N* = 14), and *wh⁷* mutant with *esg^ts^>gfp* and *hep^RNAi^* (*N* = 28) (*n* = 110 ISC/EB cells) flies. Insets in (A, C, D) show enlarged views of the boxed regions with the indicated channels in each panel. Each dot in (A', C', D') represents a single ISC or EB, and each dot in (B') represents one midgut. Scale bars in (A–D) are 50 µm (overview) and enlarged views in (A, C, D) are 25 µm. White dashed lines in (A–D) outline the edge of the gut. Bars represent the mean; error bars indicate standard deviation (SD). Statistically significant differences were identified in (A'–D') by a non-parametric Mann–Whitney *U* test. (A') The pJNK intensity was increased in *wh⁷* compared to WT (***P = 0.001). (B') PHH3⁺ cells, numbers were increased in *wh⁷* compared to WT without *hep^RNAi^* (*P = 0.01) but not with *hep^RNAi^* (ns, P = 0.12); *hep* depletion significantly reduced PHH3⁺ cells in *wh⁷* (***P = 0.0004). (C') pJNK intensity was unchanged by rapamycin in *esg^ts^>mcherry^RNAi(P)^* (ns, P = 0.07), but differed between *esg^ts^>mcherry^RNAi(P)^* and *esg^ts^>dwdr4^RNAi(P)^* with and without rapamycin (***P = < 0.0001); in *esg^ts^>dwdr4^RNAi(P)^*, rapamycin significantly reduced pJNK intensity (***P = 0.001). (D') dmyc levels were increased in *wh⁷* compared to *esg^ts^>mcherry^RNAi(P)^* without *hep^RNAi^* (***P = < 0.0001) and with *hep^RNAi^* (**P = 0.002); dmyc levels differed significantly between *wh⁷* with and without *hep^RNAi^* (***P = 0.0001). *N*, number of midguts; *n*, number of ISCs or ISC/EBs analyzed, unless otherwise stated. Source data are available online for this figure.

Blanco et al, 1998), was significantly increased in *dwdr4* mutant PMGs (Figs. 6A,A' and EV4C).

Overexpression of Catalase, an antioxidant enzyme (Ighodaro and Akinloye, 2018), in ISCs/EBs suppressed both elevated pJNK expression, ISC proliferation, and ISC/EB expansion caused by

dWdr4 depletion (Fig. EV4D,E'). To test whether JNK activation contributes to those phenotypes, we knocked down *hemipterous* [*hep*, encoding the JNK kinase (Herrera and Bach, 2021)] in ISCs/ EBs of *dwdr4* mutant guts. Hep knockdown suppressed ISC expansion, overproliferation, and EC misdifferentiation in *dwdr4*

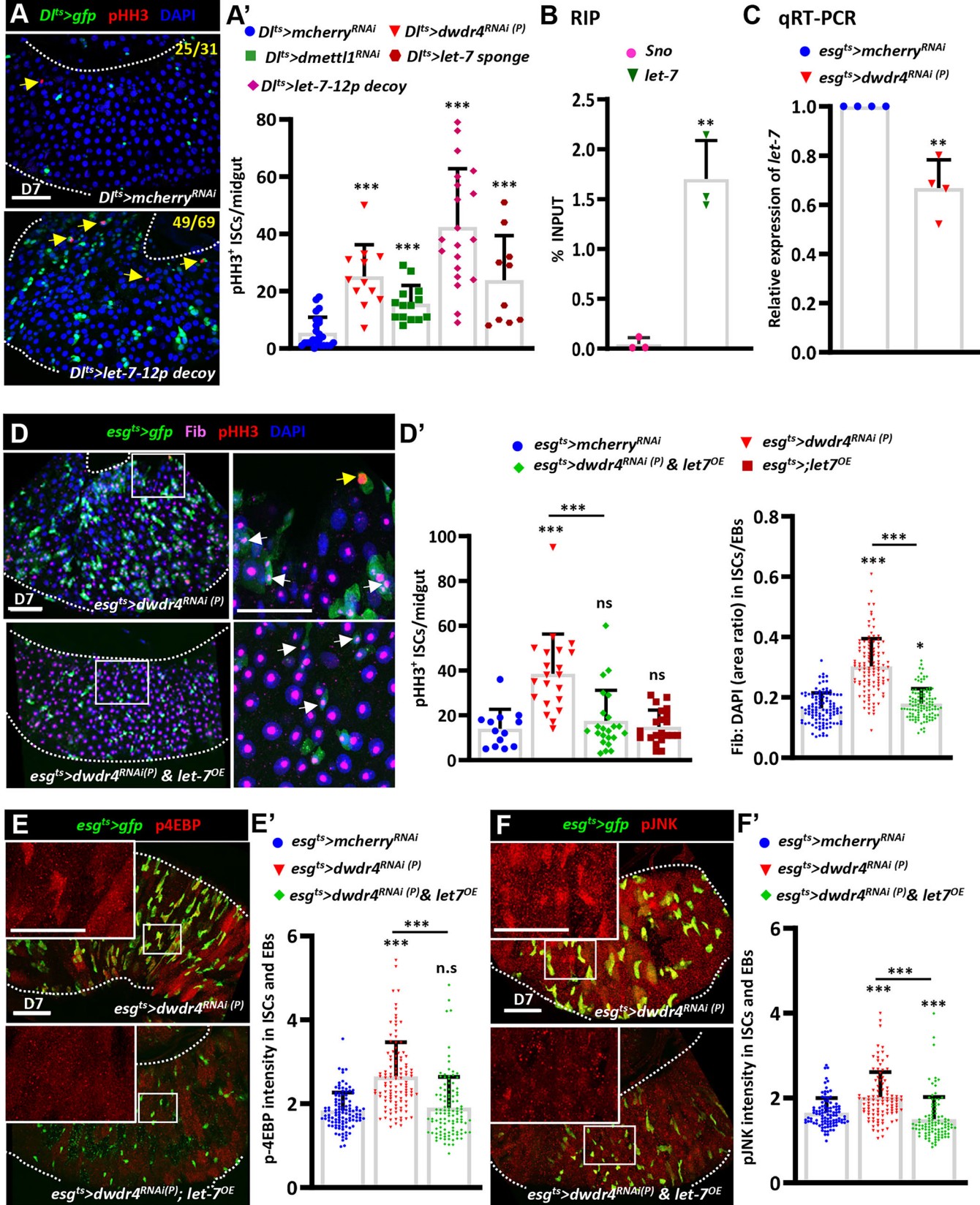

◀  **Figure 7.  dWdr4 maintains *let-7*, which can be m7G modified, to regulate the TOR-JNK-dMyc regulatory axis and maintain ISC homeostasis.**

(A) Posterior midguts (PMGs) from control (*Dl^ts^>mcherry^RNAi^*) and *Dl^ts^>let-7-12p decoy* females bearing *Dl^ts^>gfp* (green, ISCs), stained for PHH3 (red, mitosis marker) and DAPI (blue, nuclei). Fractions of gut sections displaying the indicated pattern are shown. (A') Quantification of PHH3⁺ ISCs per midgut in the *Dl^ts^>mcherry^RNAi(P)^* (N = 25), *Dl^ts^>dwdr4^RNAi(P)^* (N = 12), *Dl^ts^>dmettl1^RNAi^* (N = 14), *Dl^ts^>let-7-sponge* (N = 10) and *let-7-12p decoy* (N = 20) flies. (B) RNA immunoprecipitation (RIP) using an anti-m7G antibody followed by RT-qPCR analysis showed that *let-7* is m7G methylated in control female guts (*esg^ts^>mCherry^RNAi^*) (N = 3 biological replicates). *Sno*, a lncRNA, was used as a control for m7G-negative RNA. (C) qRT-PCR result shows that *let-7* expression is reduced in *esg>dwdr4^RNAi(P)^* guts compared to control (*esg^ts^>mcherry^RNAi^*) (N = 4 biological replicates). (C, C') Overexpression (OE) of *let-7* suppresses dWdr4-depleted phenotypes. (C) PMGs from *esg^ts^>dwdr4^RNAi^* and *esg^ts^>dwdr4^RNAi^ & let-7^OE^* females bearing *esg^ts^>gfp* (green, ISCs/EBs), stained for PHH3 (red), Fibrillarin (Fib) (magenta, nucleoli), and DAPI (blue). (C') Quantification of PHH3⁺ ISCs per midgut (left graph) and nucleolar (Fib) to nuclear (DAPI) area ratio in GFP-positive cells with small nuclei (right graph) in the *esg^ts^>mcherry^RNAi^* flies (N = 13), *esg^ts^>dwdr4^RNAi(P)^* (N = 21), *esg^ts^>dwdr4^RNAi^ & let-7^OE^* (N = 23) (n = 111 ISC/EB cells) and *let-7^OE^* alone (N = 21) flies. (D, D') Overexpression of *let-7* suppresses elevated TOR signaling induced by dWdr4 depletion. (D) PMGs from *esg^ts^>dwdr4^RNAi^* and *esg^ts^>dwdr4^RNAi^ & let-7^OE^* females bearing *esg^ts^>gfp* (green), stained for p4EBP (red, TOR signaling reporter). (D') Quantification of average p4EBP intensity in ISCs/EBs in the *esg^ts^>mcherry^RNAi^* (N = 11), *esg^ts^>dwdr4^RNAi^* (N = 11), and *esg^ts^>dwdr4^RNAi^ & let-7^OE^* (N = 13) (n = 110 ISC/EB cells) female flies. (E, E') Overexpression of *let-7* suppresses elevated JNK signaling induced by dWdr4 depletion. (E) PMGs from *esg^ts^>dwdr4^RNAi^* and *esg^ts^>dwdr4^RNAi^ & let-7^OE^* females bearing *esg^ts^>gfp* (green), stained for JNK (red, JNK signaling reporter). (E') Quantification of average JNK intensity in ISCs/EBs in the *esg^ts^>mcherry^RNAi^* (N = 28), *esg^ts^>dwdr4^RNAi^* (N = 38), and *esg^ts^>dwdr4^RNAi^ & let-7^OE^* (N = 20) (n = 100 ISC/EB cells) flies. Yellow arrows in (A, C) indicate the representative mitotic ISCs, and white arrows in (C) mark the representative ISC or EB nucleoli. Insets in (C, D, E) are enlarged views of the white square in each panel. Each dot in (C', right graph, D', E') represents a single ISCs/EBs, and each dot in (A', C') represents one midgut. Scale bars in (A, B, C, D) are 50 μm, and enlarged views in (D, E, F) are 25 μm. White dashed lines in (A, D, E, F) outline the edge of the gut. Bars represent the mean; error bars indicate standard deviation (SD). Statistically significant differences in (A', D', E', F') were identified by the non-parametric Mann–Whitney *U* test, and in (B, C) by the unpaired Student's *t* test. (A') PHH3⁺ cells were increased in *Dl^ts^>dwdr4^RNAi(P)^*, *Dl^ts^>dmettl1^RNAi^*, *Dl^ts^>let-7-12-decoy*, *Dl^ts^> let-7 sponge* compared with *Dl^ts^>mcherry^RNAi^* (***P = < 0.0001). (B) The statistical difference is **P = 0.0012. (C) *let-7* levels were reduced in *esg^ts^>dwdr4^RNAi(P)^* and *esg^ts^>mcherry^RNAi^* (**P = 0.0012). (D') PHH3⁺ cells (left graph), differed between *esg^ts^>dwdr4^RNAi(P)^* compared to *esg^ts^>mcherry^RNAi(P)^* without *let7^OE^* (***P = 0.0001) but not with *let7^OE^* (ns, P = 0.72); whereas *let7^OE^* alone had no effects (ns, P = 0.70). (D', right graph) Nucleolar size was increased in *esg^ts^>dwdr4^RNAi(P)^* compared to *esg^ts^>mcherry^RNAi(P)^* without *let7^OE^* (***P = 0.0001) and with *let7^OE^* (*P = 0.03); nucleolar size differed significantly between *esg>dwdr4^RNAi(P)^* with and without *let7^OE^* (***P = 0.001). (E') p4EBP levels, differed between *esg^ts^>dwdr4^RNAi(P)^* compared to *esg^ts^>mcherry^RNAi(P)^* without *let7^OE^* (***P = < 0.0001) but not with *let7^OE^* (ns, P = 0.67); p4EBP levels differed significantly between *esg^ts^>dwdr4^RNAi^* with and without *let7^OE^* (***P = 0.0001). (F') pJNK signals was differed in *esg^ts^>dwdr4^RNAi(P)^* compared to *esg^ts^>mcherry^RNAi^* without *let7^OE^* (***P = 0.001) and with *let7^OE^* (***P = 0.001); pJNK differed significantly between *esg^ts^>dwdr4^RNAi(P)^* with and without *let7^OE^* (***P = 0.001). N, number of midguts; n, number of ISCs or ISC/EBs analyzed, unless otherwise stated. Source data are available online for this figure.

---

mutant guts (Fig. 6B,B'), indicating that JNK signaling is a key downstream effector of dWdr4 loss. Together, these findings demonstrate that *dWdr4* loss elevates ROS levels and activates JNK signaling, which in turn drives ISC hyperproliferation and expansion resembling features of intestinal aging.

## The elevated TOR-JNK-dMyc signaling cascade drives gut hyperplasia upon dWdr4 loss

To further map the signaling cascade, we examined the relationship between TOR, JNK, and dMyc. Rapamycin treatment reduced elevated pJNK levels in *dwdr4*-depleted PMGs, but not in controls (Fig. 6C,C'), indicating that TOR acts upstream of JNK. In contrast, removing one copy of *dmyc* did not reduce p4EBP or pJNK levels (Fig. EV4F,G'), indicating that dMyc is not upstream of TOR or JNK. Notably, *hep* knockdown suppressed the elevated dMyc expression caused by dWdr4 depletion (Fig. 6D,D'), placing JNK upstream of dMyc. Together, these findings support a model in which dWdr4 loss activates TOR signaling in ISCs, leading to increased ribosome biogenesis and ROS accumulation that triggers JNK signaling. In turn, JNK promotes dMyc expression, further amplifying ribosome biogenesis. This positive feedback loop elevates biosynthetic activity and drives gut hyperplasia and aging.

## The dWdr4-dMettl1 complex maintains m7G levels to maintain ISC homeostasis in a conserved manner

To investigate how dWdr4 depletion activates TOR expression in ISCs, we first depleted known dWdr4-interacting proteins, Mei-p26 [involved in germline homeostasis (Rastegari et al, 2020)] and Fen1 [involved in DNA replication (Cheng et al, 2016)], in ISCs. However, neither depletion produced obvious intestinal

phenotypes. In human cells, WDR4 is known to form a complex with METTL1 (a tRNA methyltransferase) to mediate 7-methylguanosine (m7G) modification (Alexandrov et al, 2002). Recently, dMettl1 was shown to mediate m7G modification of tRNA for spermatogenesis (Kaneko et al, 2024). To test if dWdr4 cooperates with dMettl1 to maintain ISC homeostasis, we knocked down *dmettl1* in ISCs and found increased ISC proliferation (Fig. 7A,A'), reminiscent of *dwdr4* depletion phenotypes. Both *dwdr4*-and *dmettl1*-knockdown ISCs/EBs exhibited decreased m7G levels (Fig. EV5A,B'), and a Proximity Ligation Assay (PLA) assay revealed the interaction between dWdr4 and dMettl1 (Fig. EV5C,C'). Furthermore, ISC overproliferation and ISC/EB expansion induced by dWdr4 depletion were dramatically rescued by human WDR4 (Fig. EV5D,E'), while those caused by dMettl1 depletion were rescued by human METTL1 but not by the catalytic domain mutant (METTL1^DK^) (Fig. EV5F,F'). These results indicate that the Wdr4-Mettl1 complex regulates ISC homeostasis through m7G modification, and that this regulatory mechanism is conserved from flies to humans.

## dWdr4-dMettl1 maintains let-7 levels, likely through m7G modification, to control TOR signaling

In addition to modifying tRNAs, the WDR4-METTL1 complex has been shown to mediate the m7G modification of miRNAs in cancer cell lines, including *let-7* miRNA (Pandolfini et al, 2019). The *let-7* miRNA is downregulated in intestinal adenocarcinomas and colorectal cancer (Madison et al, 2015; Mizuno et al, 2018), and targets several components in the amino acid sensing pathway to suppress TOR activation (Dubinsky et al, 2014). Interestingly, *let-7* is also decreased in *dwdr4* mutant ovaries (Fig. EV6A) (Rastegari et al, 2020), raising the possibility that the dWdr4-dMettl1

complex regulates TOR signaling in ISCs through m$^7$G modification of *let-7*.

The *let-7* miRNA is co-transcribed with *miR-100* and *miR-125* from the *let-7* complex (let-7C) locus (Sokol et al, 2008). To test whether *dWdr4* affects let-7 expression, we examined PMGs using a validated transcriptional reporter, *let7-Cp12.5Kb::lacZ*, which contains a 12.5 kb intronic fragment of the *let-7C* locus upstream of *LacZ* (Chawla and Sokol, 2012). Reporter expression was detected in ISCs and remained unchanged upon *dWdr4* depletion (Fig. EV6B). We next disrupted *let-7* function in ISCs by overexpressing either a *let-7* sponge (mCherry with a 3'UTR containing 20 complementary *let-7* binding sites) (Fulga et al, 2015), and *let7-12p* decoy (containing 12 complementary *let-7* binding elements generated in this study). Both manipulations phenocopied *dwdr4* or *dmettl1* depletion, leading to ISC overproliferation (Fig. 7A,A') and enhanced ribosome biogenesis (Fig. EV6C,C'). We further showed that *let-7* can be immunoprecipitated by anti-m$^7$G antibody (Fig. 7B) and *let-7* expression was reduced in dWdr4-depleted guts (Fig. 7C), as revealed by RT-PCR. This is consistent with a requirement for m$^7$G methylation in *let-7* processing (Pandolfini et al, 2019). Moreover, overexpression of *let-7* in ISCs/EBs significantly suppressed ISC overproliferation, ISC/EB expansion, and nucleolar enlargement, demonstrating that elevated ribosome biogenesis in dWdr4-depleted intestines is mediated by *let-7* loss (Fig. 7D,D'). In contrast, *let-7* overexpression alone did not produce noticeable differences compared with controls (Fig. 7D,D'). Consistently, *let-7* overexpression in ISCs/EBs suppressed TOR (Fig. 7E,E') and JNK signaling (Fig. 7F,F') in dWdr4-depleted intestines. Together, these findings imply that the dWdr4-dMettl1 complex mediates m$^7$G modification of *let-7* to restrain the TOR-JNK-dMyc signaling cascade and maintain steady-state ribosome biogenesis for ISC homeostasis.

# Discussion

## Wdr4-Mettl1-mediated m$^7$G modification of *let-7* restrains TOR–JNK–dMyc signaling to maintain ribosome biogenesis and intestinal integrity

Tight regulation of ISCs is essential for maintaining intestinal epithelial homeostasis. While ribosome biogenesis has emerged as a key determinant of stem cell properties (Gabut et al, 2020), the mechanisms that regulate ribosome biogenesis in ISCs and their impact on intestinal homeostasis remain poorly understood. In parallel, the *let-7* miRNA is known to act as a tumor suppressor (Chirshev et al, 2019) and to prevent the development of intestinal adenocarcinomas (Madison et al, 2015), yet how *let-7* is regulated in the ISC lineage remains unclear.

Our study identifies a regulatory pathway in which the Wdr4-Mettl1 complex promotes *let-7* expression in ISCs through m$^7$G modification. This represents the first in vivo evidence that Wdr4-Mettl1 modifies miRNAs with m$^7$G to support proper ribosome biogenesis at the tissue level. Loss of dWdr4 in ISCs relieves *let-7*-mediated suppression of TOR signaling, resulting in hyperactive TOR signaling. Elevated TOR, in turn, activates JNK-dMyc signaling, driving ISC overproliferation and ISC/EB expansion. In addition, Dl and Notch are misexpressed, leading to defective ISC-EC differentiation. Importantly, this mechanism involving miRNA

modification, TOR signaling, and ribosome biogenesis may be evolutionarily conserved, as human WDR4 and METTL1 can substitute for their fly counterparts to restore ISC regulation.

TOR-induced accumulation of ROS has been reported in yeast (Zeng et al, 2021), fly larval imaginal discs (Lee et al, 2010), and hematopoietic stem cells (Chen et al, 2008), however, the underlying mechanism remains unclear. One possibility is that increased TOR signaling enhances mitochondrial oxidative phosphorylation, thereby generating more ROS (Hwang et al, 2012). In dWdr4-depleted ISCs/EBs, ROS generation activates JNK–dMyc signaling cascades not only within ISCs/EBs but also in neighboring differentiated cells located several diameters away, consistent with the role of JNK as a stress-signal amplifier that promotes secondary ROS or cytokine release to propagate JNK–dMyc activation (Santabárbara-Ruiz et al, 2015; Zhang et al, 2024). Furthermore, dWdr4 depletion in ECs promotes EC endoreplication, which may also induce ROS-JNK-dMyc signaling cascades, as ROS, pJNK, and dMyc expression are elevated in the entire PMGs upon dWdr4 loss. Together, these findings demonstrate how dWdr4-mediated miRNA modification safeguards ISC function and intestinal homeostasis, and they may help explain why patients with gastrointestinal cancers exhibiting low WDR4 expression have poorer survival outcomes, suggesting potential therapeutic avenues.

## WDR4 as a suppressor of intestinal cancers through Let-7

Recent studies have been implicated in multiple cancers (Wang et al, 2023; Zeng et al, 2021). Notably, our analysis and prior work (Zeng et al, 2021) indicate that patients with stomach or rectal cancer exhibiting higher WDR4 expression have better overall survival than those with lower expression. Consistent with this association, loss of dWdr4 in Drosophila causes *let-7*–dependent ISC hyperproliferation, intestinal dysplasia, and shortened lifespan, suggesting a conserved WDR4–Let-7 pathway that may influence intestinal tumorigenesis.

miRNAs are short, single-stranded RNAs that post-transcriptionally repress gene expression, typically by binding 3'UTRs to promote mRNA degradation or inhibit translation (Ranganathan and Sivasankar, 2014). Among them, the *Let-7* is abundant and well established as a tumor suppressor (Büssing et al, 2008). Mice lacking *let-7* develop intestinal adenocarcinomas and *Let-7* is significantly reduced in human colorectal cancer, correlating with elevated stem cell markers (Madison et al, 2015). *let-7* processing can be blocked by Lin28 binding (Piskounova et al, 2008; Viswanathan et al, 2008), which is upregulated in several cancers, including colorectal cancer (Madison et al, 2013; Viswanathan et al, 2009). In our study, functional elimination of *let-7* phenocopied dWdr4 depletion, and *let-7* overexpression rescues these phenotypes. Although a previous study reported no overt midgut phenotype upon *let-7* elimination (Chen et al, 2015), their *let-7* sponge contained only six complementary binding sites, whereas our sponges harbor 12 and 20 sites, respectively, likely increasing sequestration efficiency and phenotypic detectability, which may account for the discrepant outcomes.

Mechanistically, in mammalian cells, m$^7$G installed within a RAGGU motif on *let-7e-5p* disrupts a nearby inhibitory G-quadruplex, thereby facilitating *let-7* processing (Pandolfini et al, 2019). By contrast, in a recent *Drosophila* study reported that fly

*let-7* lacks the conserved RAGGU context for m⁷G based on sequence analysis, and *dmettl1* mutant testes show no reduced *let-7* abundance (Kaneko et al, 2024). In our intestinal model, however, loss of dWdr4 decreases *let-7*, *let-7* elimination phenocopies dWdr4 deficiency, and *let-7* is immunoprecipitated by anti-m⁷G antibody, together suggesting a tissue-specific regulation of *let-7* in *Drosophila*. Together with human evidence, these finding support a model in which WDR4 regulates *let-7* and may function as a suppressor of intestinal tumorigenesis.

## Sex differences in dWdr4 regulation of ISC lineage

Sexual dimorphism in physiology and metabolism is increasingly recognized in *Drosophila* intestinal biology. Male intestines are shorter and thinner than female intestines, and male ISCs proliferate more slowly, with the difference further enhanced by mating (Regan et al, 2016). Aging also impacts the sexes differently: female intestines exhibit epithelial deterioration and barrier dysfunction (Biteau et al, 2008), while male intestines maintain epithelial integrity even at advanced ages (Regan et al, 2016). Elevated ROS and JNK signaling drive female intestinal aging, whereas male ECs are protected by higher basal autophagy, which degrades damaged proteins and organelles independently of TOR signaling (Regan et al, 2022). Disruption of autophagy in males induces female-like intestinal aging, whereas activating autophagy in females reduces ROS sensitivity by suppressing p62 and Hippo signaling, protecting barrier integrity (Nagai et al, 2021). Males and females also differ in carbohydrate metabolism (Hudry et al, 2019) and gut microbiota composition (Han et al, 2017), with similar sex differences reported in humans (Houdeau, 2016; Kim et al, 2020). In our study, *dwdr4* mutant females displayed shortened intestines, thickened PMGs, and gut leakage, whereas males did not show these phenotypes. These sex-specific differences could reflect the gene's X-linked dosage (two copies in females vs. one in males) (Regan et al, 2016), differences in ISC populations, or influences of sex-specific hormones or microbiota. Further studies will be required to clarify these mechanisms.

# Methods

### Reagents and tools table

| Reagent/ resource | Reference- or source | Identifier/- catalog no. |
|---|---|---|
| **Experimental - models** | | |
| *Drosophila melanogaster* Strains | Bloomington Drosophila Stock Center (BDSC); Vienna Drosophila Resource Center (VDRC) | N/A |
| *yw* | N/A | N/A |
| *wh7* | Generated in the previous study | N/A |
| *UASp-hWDR4* | Generated in the previous study | N/A |

| Reagent/ resource | Reference- or source | Identifier/- catalog no. |
|---|---|---|
| *UASp-dwdr4* | Generated in the previous study | N/A |
| *dwdr4-gfp* | Generated in the previous study | N/A |
| *esg-GAL4* | A gift from G-C Chen | N/A |
| *Dl-GAL4* | A gift from G-C Chen | N/A |
| *NP1-GAL4* | A gift from G-C Chen | N/A |
| *esg-GAL4, UAS-GFP, tub-GAL80ts (esgts)* | A gift from Lucy Erin O'Brien | N/A |
| *Dl-GAL4,UAS-GFP,tub-GAL80ts (Dlts)* | A gift from Dr. Bruno Lemaitre | N/A |
| *Su(H)GBE-GAL4,UAS-mCD8gfp;tub-GAL80ts(EBts)* | A gift from Dr. Bruno Lemaitre | N/A |
| *UASp-mCD8gfp* | A gift from C-Yuan Tseng | N/A |
| *Uasp-Catalase* | Huang et al, 2024 | N/A |
| *how-GAL4* | BDSC | 1767 |
| *UAS-mcherryRNAi* | BDSC | 35785 |
| *UAS-egfpRNAi* | BDSC | 41552 |
| *UAS-dwdr4RNAi(P)* | BDSC | 61281 |
| *UAS-dwdr4RNAi(t)* | VDRC | 41618 |
| *UAS-hepRNAi* | BDSC | 35210 |
| *UAS-dmettl1RNAi* | VDRC | 106055 |
| *UAS-TORRNAi* | BDSC | 33951 |
| *dmyc4 mutant* | BDSC | 64769 |
| *let7-Cp12.5Kb::lacZ* | BDSC | 92822 |
| *UAS-let-7-sponge* | A gift from Dr. Tsai-Feng Fu | N/A |
| *UASz-Mettl1-V5* | This study | N/A |
| *UASz-FLAG-hMETTL1* | This study | N/A |
| *UASz-FLAG-hMETTL1 catalytic-dead mutant* | This study | N/A |
| *pUAST-attB_decoy-let-7-12p* | This study | N/A |
| *UAS-LucRNAi* | BDSC | 61306 |
| **Antibodies** | | |
| Mouse anti-β-gal | Promega | Z3781 |
| Mouse anti-Dlg | DSHB | 4F3 |
| Mouse anti-Pros | DSHB | MR1A |

| Reagent/ resource | Reference- or source | Identifier/- catalog no. |
|---|---|---|
| Chicken anti-GFP | Abcam | ab13970 |
| Rabbit anti-GFP | GeneTex | GTX113617 |
| Rabbit anti-Phospho-Histone H3 (PHH3) (Ser10) | Merck Millipore | 06-570 |
| Rabbit anti-phospho (p)-JNK (Thr183/Tyr185) | Cell signaling | #9251 |
| Mouse anti-Dl | DSHB | AB_C594.9B |
| Mouse anti-Lamin (Lam) C | DSHB | LC28.26 |
| Mouse anti-Fibrillian 38F3 | Abcam | AB_4566 |
| Mouse anti-dMyc | DSHB | P4C4-B10 |
| Rabbit anti-(p)-4E-BP (Thr37/46) | Cell Signaling | #236B4 |
| Mouse anti-headcase | DSHB | HDC U33 |
| Mouse anti-m7G | MBL | RN017M |
| Rabbit anti-GFP | Torrey Pines Biolabs | #TP401 |
| Mouse anti-V5 | Invitrogen | R960-25 |
| Alexa Fluor 488-conjugated goat anti-chicken | Jackson ImmunoResearch Lab | 125986 |
| Alexa Fluor 568 conjugated goat anti-mouse | Invitrogen | A21094 |
| Alexa Fluor 633 conjugated goat anti-mouse | Invitrogen | A11077 |
| Alexa Fluor 488-conjugated goat anti-rabbit | Invitrogen | A1008 |
| **Chemicals, enzymes, and other reagents** | | |
| Phalloidin | Invitrogen | R415 |
| Bodipy (493/503) | Thermo Fisher | D3922 |
| Rapamycin | Sigma-Aldrich | R0935 |
| DAPI | Sigma | N/A |
| Hoechst | Invitrogen | 2098863 |
| DMSO | Sigma | D8418 |
| Grace's insect medium | Lonza | 04-457F0 |
| PBS | Uni Region Biotech | N/A |
| Formaldehyde | Alfa Aesar | N/A |
| Glycerol containing 20 µg/ml N-propyl gallate | Sigma | N/A |

| Reagent/ resource | Reference- or source | Identifier/- catalog no. |
|---|---|---|
| BMH-21 | AdooQ Bioscience | A14335 |
| DHE | Invitrogen | 2140299 |
| Blue Food Dye | Sigma | MKBS2204V |
| EU Incorporation Click-iT RNA Alexa Fluor 594 Imaging kit | Invitrogen | C10330 |
| Click-iT EdU imaging kit | Invitrogen | C10338 |
| TaqMan® 2X Universal PCR Master Mix | Applied Biosystems | N/A |
| TaqManTM MicroRNA Assay Kit | Applied Biosystems | 4427975 |
| Phusion™ High-Fidelity DNA Polymerase | New England Biolabs | M0530S |
| NEBuilder HiFi DNA Assembly Master Mix | New England Biolabs | E2621 |
| 1-bromo-3-chloropropane | Sigma-Aldrich | N/A |
| yDcpS | NEB | M0463S |
| RNAsin Plus | Promega | N2611 |
| RNA Clean and Concentrator-5 kit | Zymo Research | R1017 |
| RNaseOUT | Thermo Fisher | 10777019 |
| Dynabeads Protein G | Thermo Fisher | 10004D |
| Duolink In Situ PLA kit | Sigma-Aldrich | DUO 9200 |
| Trizol | Novel-Gene | N/A |
| anti-rabbit PLUS PLA probe | Sigma-Aldrich | DUO92002 |
| anti-mouse MINUS PLA probe | Sigma-Aldrich | DUO92004 |
| Duolink in situ mounting medium with DAPI | Sigma-Aldrich | DUO82040 |
| **Primers** | | |
| Primers used for constructing decoy-let7-12P | | |
| decoy let7 function | F | GACGGCGCTAGGAT CATCAACACTATACAA CCATCTTACTACCTC ACAAGTATTCTGGTCA CAGAATACAACACTA |
| | R | GACGGCGCTAGGATCATCTTGT GAGGTAGTAAGATGGTTGT ATAGTGTTGTATTCTGT GACCAGAATACTTGTGAG |
| decoy-miR-EcoR1/Nhel | F | TCTGAATAGGGAATTCCGCCG CTAGCGACGGCGCTAGG ATCATCAAC |

| Reagent/ resource | Reference- or source | Identifier/- catalog no. |
|---|---|---|
| decoy-miR-Not1/SpeI | R | TCTGAATAGGGAATTCC GCCGCTAGCGACGGC GCTAGGATCATCAAC |
| Mettl1-V5n | F | AGGATCCCTCGAGGGTAC CAATGGTGG CCACCGGTGGACA |
| | R | TCTGAATAGGGAATT CCGCCGCTAGCGACGG CGCTAGGATCATCAAC |
| V5c-pA | F | CCGAATCCGCTCTTGGGCTT GGATTCGACATAGGGGTACCA CTAGTTCTAGAATGAATC |
| | R | TCACGAGGCCCTTTCGTCTT |
| **TaqManTM microRNA assay miRbase ID and assay ID used in this study** | | |
| Target | miRbase ID | Assay ID |
| let-7 | dme-let-7-5p | OOO332 |
| snoRNA227 | dmesnoRNA-227 | CTXGPYT |
| **Software** | | |
| GraphPad Prism 9.0 | GraphPad Software Inc. | https://www.graphpad.com/ |
| ImageJ | National Institute of Health, USA | https://imagej.nih.gov/ij/ |
| Adobe Photoshop 2025 | Adobe Systems, San Jose, CA | https://www.adobe.com/ |
| ZEN 3.1 (blue edition) | ZEISS, Germany | https://www.zeiss.com/-microscopy/en/products/-software/zeiss-zen.html |
| Cytoscape | National Human Genome Research Institute | https://cytoscape.org/ |
| Microsoft Excel 2021 | Microsoft | N/A |
| **Other** | | |
| BDSC | Bloomington Drosophila Stock Center | https://bdsc.indiana.edu/ |
| VDRC | Vienna Drosophila Resource Center | https://shop.vbc.ac.at/vdrc_store/ |
| FlyBase | | https://flybase.org/ |
| Illumina NovaSeq 6000 | Illumina | N/A |

## Drosophila strains and culture

*Drosophila* stocks were maintained at 22–25 °C on a standard cornmeal (7.1% w/v), agar (0.8% w/v), and yeast (2.4% w/v) diet, unless otherwise indicated. The *yw* strain was used as a wild-type control. The following fly strains were used in this study: *wh⁷*, *UASp-hWDR4, UASp-dwdr4* and *dwdr4-gfp* were generated in previous studies (Rastegari et al, 2020; Wu et al, 2006); *esg-GAL4, Dl-GAL4,* and *NP1-GAL4* (gift from Dr. Guang-Chao Chen, Institute of Biological Chemistry, Academia Sinica, Taiwan); *esg-GAL4,UAS-GFP,tub-GAL80ᵗˢ* (*esgᵗˢ*) (gift from Dr. Lucy Erin

O'Brien, Department of Molecular and Cellular Physiology, Stanford University, California, USA), *Dl-GAL4,UAS-GFP,tub-GAL80ᵗˢ* (*Dlᵗˢ*), *Su(H)GBE-GAL4,UAS-mCD8gfp;tub-GAL80ᵗˢ* (*EBᵗˢ*) (gift from Dr. Bruno Lemaitre, Global Health Institute, Ecole Polytechnique Fédérale de Lausanne, Switzerland), *UASp-mCD8gfp* (Tseng et al, 2014), *UASp-Cat* (Huang et al, 2024), *how-GAL4* (BDSC 1767), *UAS-mcherryᴿᴺᴬⁱ* (BDSC 35785), *UAS-egfpᴿᴺᴬⁱ* (BDSC 41552), *UAS-dwdr4ᴿᴺᴬⁱ⁽ᴾ⁾* (BDSC 61281), *UAS-dwdr4ᴿᴺᴬⁱ⁽ᵗ⁾* (VDRC 41618), *UAS-hepᴿᴺᴬⁱ* (BDSC 35210), *UAS-dmettl1ᴿᴺᴬⁱ* (VDRC 106055), *UAS-TORᴿᴺᴬⁱ* (BDSC 33951), *dmyc⁴* mutant (BDSC 64769), *let7-Cp¹²·⁵ᴷᵇ::lacZ* (BDSC 92822) and *UAS-let-7-sponge* (gift from Dr. Tsai-Feng Fu, Department of Applied Chemistry, National Chi Nan University, Taiwan), were obtained from the Bloomington *Drosophila* Stock Center (BDSC). Other genetic elements are described in FlyBase (http://flybase.bio.indiana.edu). Flies expressing *RNAi* were cultured at 18 °C with the indicated GAL4 with or without GAL80ᵗˢ and shifted to 29 °C for 7 days before dissection. For earlier experiments, we used *Dl-GAL4* and *esg-GAL4*, while in later experiments we employed *Dlᵗˢ* and *esgᵗˢ* to achieve better temporal control of *UAS* transgene expression. Fly genetics to obtain *wh⁷* homozygous female flies with or without other genetic elements is shown in the Appendix information. Genotypes of flies used in each figure and figure EV are provided in Appendix Table S4.

## Fly genetics for generating *wh⁷* mutants with and without other genetic elements

To obtain *wh7* homozygous mutant female flies, *wh⁷/FM7c* females were crossed with *dwdr4-gfp /CyO* males to produce *wh⁷/Y; dwdr4-gfp /+*, used for fertility assays since *wh⁷* mutant males are sterile (Wu et al, 2006). These males were then crossed with *wh⁷/FM7c* females to obtain *wh⁷/wh⁷; +/+* females for experiments.

To generate *wh⁷* homozygous mutant females expressing *esg > hWDR4*, males of the genotype *esg-GAL4/CyO; UAS-mCD8gfp/TM3* were crossed with *wh⁷/FM7c; dwdr4-gfp/CyO* females to produce *wh⁷/Y; esg-GAL4/dwdr4-gfp; UAS-mCD8gfp/+* male progeny. These males were then crossed with *wh⁷/FM7c; hWDR4/TM3* females to obtain *wh⁷/wh⁷; esg-GAL4/+; hWDR4/UAS-mCD8gfp* females. The desired genotypes were confirmed by the absence of dwdr4-gfp expression (present in nuclei) through blinded immunostaining.

To generate *wh⁷* homozygous mutant females expressing *esgᵗˢ>hepᴿᴺᴬⁱ*, males of the genotype *esg-GAL4 UAS-gfp/CyO; Su(H)lacZ tub-GAL80ᵗˢ/TM6B* (esgts-GAL4) were crossed with *wh7/FM7c; dwdr4-gfp/CyO* females to produce *wh⁷/Y; esg-GAL4 UAS-gfp/ dwdr4-gfp; Su(H)lacZ tub-GAL80ts/+* male progeny. These males were then crossed with *wh⁷/FM7c; UAS-hepᴿᴺᴬⁱ* females to obtain *wh⁷/wh⁷; esg-GAL4 UAS-gfp/+; Su(H)lacZ tub-GAL80ts/UAS-hepᴿᴺᴬⁱ* females. The desired genotypes were confirmed by the absence of *dwdr4-gfp* (present in nuclei) through blinded immunostaining.

## Rapamycin treatment

Rapamycin (Sigma-Aldrich, R0935) was dissolved in DMSO at 50 mM and stored at −20 °C as a stock solution. For use, the stock was diluted to 50 μM with ddH₂O, and 100 μl of the rapamycin solution (or ddH₂O for controls) was evenly applied to the food

surface across grooves made with a scalpel. The food was air-dried in a hood and stored at 4 °C. Newly eclosed female flies were transferred to rapamycin-containing food [warmed to room temperature (RT)] and maintained with daily food changes until dissection.

## BMH-21 treatment

BMH-21 (AdooQ Bioscience, A14335) (gift from Dr. Jim-Jinn-Chyuan Sheu, Institute of Biomedical Sciences, National Sun Yat-sen University, Taiwan) was dissolved in DMSO at 5 mM and stored at −20 °C as a stock solution. For use, the stock was diluted to 5 µM with ddH$_2$O, and 100 µl of the freshly prepared BMH-21 solution (or 0.1% DMSO for controls) was applied evenly to the surface of the food across grooves made with a scalpel. Three to 4-day-old female flies were transferred to BMH-21-containing food (warmed to RT) for 3 days, with food changed daily until dissection.

## Genetic mosaic analysis

Genetic mosaic clones were generated using FLP/FRT-mediated MARCM analysis. Flies with genotypes *FRT19A/hs-flp GAL80 FRT19A; act-GAL4/UAS-mCD8gfp* and *wh$^7$FRT19A/hs-flp GAL80 FRT19A; act-GAL4/UAS-mCD8gfp* were generated from the standard crosses. Three-to-four-day-old mated female flies were subjected to heat shock for 45 min at 37 °C, a condition commonly used in intestinal studies. After heat shock, flies were cultured at 25 °C until dissection; food was changed daily. Homozygous mutant cells were identified by the presence of GFP.

## Transgenic fly line generation

Fragments of *FLAG-METTL1-WT* and *FLAG-METTL1* catalytic-dead (KD) mutant [harboring L160A and D163A mutations (Orellana et al, 2021) were amplified from the *pBAbe-METTL1-WT* and *pBAbe-METTL1-KD* mutant plasmids (gifts from Dr. Richard Gregory, Department of Hematology Oncology, Boston Children's Hospital, USA), respectively, using PCR. The resulting DNA fragments were subcloned into the *pUASz1.0* vector (*Drosophila* Genomics Resource Center, DGRC #1431).

The *UASz-Mettl1-V5* construct was generated by inserting the coding sequence of Mettl1 with a C-terminal V5 tag into the *pUASz* vector. The Mettl1-V5 insert was prepared through PCR amplification. PCR was performed using Phusion™ High-Fidelity DNA Polymerase (New England Biolabs). Specifically, the Mettl1-V5 coding sequence was amplified from fly genomic DNA using custom-designed forward and reverse primers to generate overlapping fragments (Mettl1-V5n and V5c-pA) [V5n, V5 n-terminal; V5c, v5 c-terminal; and PolyA, pA] with a minimum of 20 bp overlap. The resulting PCR fragments and the EcoRI/SpeI linearized *pUASz* vector were then assembled using the NEBuilder HiFi DNA Assembly Master Mix (New England Biolabs, # E2621) following the manufacturer's protocol. The assembled plasmid was confirmed by Sanger sequencing. The primer sequences used for PCR amplification are detailed in Reagents and Tools Table. Final plasmid constructs were injected into embryos harboring an *attP2* landing site on the 3rd chromosome (WellGenetics, Taiwan).

Decoy RNA targeting the mature miRNA let-7 was designed according to the tough decoy (TuD) strategy described by

Haraguchi et al (Haraguchi et al, 2009), with modifications for improved applicability. The complementary sequence of let-7 was incorporated into a pair of synthetic oligonucleotides ("decoy_let-7_functional primers") and was amplified by PCR to generate a DNA fragment encoding a basic *decoy-let-7* unit predicted to fold into a trap-like RNA structure upon transcription. The amplified fragment was re-amplified using primers decoy-miR-EcoRI/NheI-F and decoy-miR-NotI/SpeI-R, through which EcoRI and NotI restriction sites were introduced at the 5' and 3' ends, respectively, to enable directional cloning. To increase sequestration efficiency, twelve tandem copies of the decoy-let-7 unit were assembled through NheI and SpeI sites, and the resulting decoy-let-7-12p cassette was obtained. The multimerized fragment was digested with EcoRI and NotI and was subsequently ligated into the corresponding sites of the pUAST-attB vector, by which the final construct, pUAST-attB_decoy-let-7-12p, was generated. This construct retained the fundamental TuD RNA architecture while its functionality was extended through multimerization, thereby enhancing the efficiency of *let-7* sequestration. The primer sequences used for PCR amplification are detailed in Reagents and Tools Table.

## Smurf assay

Intestinal integrity was evaluated according to the body distribution of non-absorbable blue food dye (Sigma, MKBS2204V). A total of 10–15 female flies (7-day-old) in each group were transferred from normal food to blue dye mixed with wet yeast (2.5% w/v) for 5–6 h. The gut integrity was determined by monitoring the blue dye coloration inside the body cavity.

## Immunostaining and fluorescence microscopy

For imaging of fly guts, immunostaining was performed as previously described (Micchelli, 2014), with modifications. In brief, ten guts in a group were dissected within 30 min in ice-cold 1X PBS (Uni Region Biotech) and fixed in 4% formaldehyde (FA; Alfa Aesar) for 2 h at RT without agitation. Samples were washed with cold 1× PBS for 2 h four times. Then, the samples were incubated with primary antibodies in 0.1% PBST (1× PBS with 0.1% Triton-X 100) at 4 °C O/N. After washing, samples were incubated with secondary antibodies at 4 °C O/N, followed by washing and counterstaining. Guts were dissected from 7-day-old females unless otherwise indicated. The following primary antibodies were used at the indicated dilutions: mouse anti-β-gal (Promega, Z3781, 1:200), mouse anti-Dlg (DSHB, 4F3, 1:100), anti-Pros (DSHB, MR1A, 1:100), chicken anti-GFP (Abcam, ab13970, 1:1000), rabbit anti-GFP (GeneTex, GTX113617, 1:500), rabbit anti-Phospho-Histone H3 (PHH3)(Ser10) (Merck Millipore, 06-570, 1:250), rabbit anti-phospho (p)-JNK (Thr183/Tyr185) (Cell signaling, #9251, 1:200; a gift kindly provided by Dr. Hsueh-Cheng Chiang, Department of Pharmacology, National Cheng Kung University, Tainan, Taiwan), mouse anti-Dl (DSHB, AB_C594.9B, 1:10), mouse anti-Lamin (Lam) C (DSHB, LC28.26, 1:500), mouse anti-Fibrillarin 38F3 (Abcam, AB_4566, 1:500), mouse anti-dMyc (DSHB, P4C4-B10, 1:5) rabbit anti-(p)-4E-BP (Thr37/46) (Cell Signaling, #236B4, 1:500), mouse anti-Hdc (DSHB, HDC-U33, 1:10), and mouse anti-m$^7$G antibody (MBL, RN017M, 1:1000). Secondary antibodies were used as follows: Alexa Fluor 488-conjugated goat anti-chicken

(Jackson ImmunoResearch Lab, 125986, 1:500), Alexa Fluor 568- and 633-conjugated goat anti-mouse (Invitrogen, A21094 and A11077, 1:500), and Alexa Flour 488-conjugated goat anti-rabbit (Invitrogen, A1008, 1:500). F-actin was stained with phalloidin (Invitrogen, R415, 1:100), and neutral lipids were stained with 50 μM Bodipy (493/503) (Thermo Fisher, D3922) in 0.1% PBST, at RT for 30 min. DNA was stained with 0.5 μg/ml DAPI (Sigma) for 10 min at RT. Finally, samples were mounted in 80% glycerol containing 20 μg/ml N-propyl gallate (Sigma) and analyzed using a Zeiss LSM 700 or 900 confocal microscope. EdU incorporation was performed with the Click-iT EdU imaging kit (Invitrogen, C10338), following the instruction manual. In brief, dissected guts were incubated with 10 μM EdU in pre-warmed Grace's insect medium (GIM, Lonza, 04- 457F0) for 1 or 2 h at RT before fixation and immunostaining as described above.

EU incorporation was performed with the Click-iT RNA Alexa Fluor 594 Imaging kit (Invitrogen, C10330), following the instruction manual. In brief, dissected guts were incubated with 10 mM EU at RT for 2 h. Guts were then fixed and stained with the primary antibodies and washed as mentioned above.

## DHE staining

DHE staining was performed as previously described (Owusu-Ansah et al, 2008). Briefly, guts were dissected in GIM and incubated with 30 μM DHE (Invitrogen, 2140299) for 7 min at RT, followed by two washes of 5 min each with GIM. The guts were stained with Hoechst (Invitrogen, 2098863, 1:1000) for 2 min at RT to visualize the cell nuclei, and then post-fixed in 4% FA in 1× PBS for 7 min. After washing, the guts were mounted on slides as described above, and images were captured.

## Lifespan assay

Lifespan assays were performed as previously described (Chen et al, 2012). To standardize genetic backgrounds, $wh^7$ mutant flies were outcrossed for 10 generations to the $yw$ strain that had been inbred for 10 generations. Newly eclosed flies of each genotype were collected and placed in vials at 10 pairs per vial, with 8 replicates. Flies were maintained at 25 °C, transferred to fresh food every 2 days, and mortality was recorded daily until all flies had died.

For RNAi-mediated assays, flies expressing $UAS$-$dwdr4^{RNAi(P)}$ (a TRIP $RNAi$ line) were compared to control flies expressing $UAS$-$LucRNAi$ (a TRIP control $RNAi$ line; BDSC #61306), both driven by $esg$-$GAL4$. Newly eclosed flies of each genotype were collected and placed in vials at ten pairs per vial, with ten replicates. Flies were maintained at 29 °C, transferred to fresh food every 2 days, and mortality was recorded daily until all flies had died.

Similar results were observed in $wh^7$ mutant flies without outcrossing or when $UAS$-$mcherry^{RNAi}$ (non-TRIP $RNAi$) was driven by $esg$-$GAL4$. The statistical significance was calculated by the log-rank test.

## Climbing assay

Climbing assays were performed as previously described (Manjila and Hasan, 2018). Ten well-fed 1-week-old female or male non-anesthetized flies were transferred to an empty plastic vial for 5 min to allow the flies to grow accustomed to the new environment.

Another tube was inverted and taped to the vial containing flies; both vials were marked at a height of 8 cm from the bottom. Then the flies were gently tapped to the bottom of one of the vials; the numbers of flies that crossed over the 8 cm mark within 12 s were recorded. The experiments were repeated three times with the same sets of flies after rest intervals of 5 min. Each genotype was tested in at least three independent replicates.

## Analysis of genes coexpressed with dWdr4

RNA-seq counts ("RNA-seq RPKM values", March 2023 release) were obtained from FlyBase (https://flybase.org/downloads/bulkdata). Using Spearman correlation, 3886 genes were identified as being significantly (Bonferroni adjusted $P$ value < 0.05), monotonically, and positively correlated with $dwdr4$. Next, a matrix was generated for training a random forest classifier that could predict the expression of $dWdr4$ using the RNA-seq counts of the 3886 correlated genes as features. Using the Boruta R package (Kursa and Rudnicki, 2010), the feature importance of the genes was determined (higher feature importance equates to a greater contribution to the accuracy of prediction), and all genes that lowered the prediction accuracy of the classifier were removed. After applying this filter, 166 genes remained and were considered predictive of $dWdr4$ expression.

The DAVID functional annotation tool (Huang da et al, 2009; Sherman et al, 2022) was used to determine ontological categories (GO, KEGG, and Reactome) for each of the 166 predictive genes (Benjamini cutoff <0.05). Using Cytoscape, a graph was generated with nodes representing the functional categories and edges representing the correlation between the nodes. Hierarchical clustering and a standard tree-cutting algorithm were used to sort the functional categories into the nine clusters shown in Fig. 4A.

## RNA extraction and RNA sequencing

For each replicate, 20–30 guts from one-week-old $yw$ and $wh^7$ mutant females were dissected and frozen by dipping in liquid nitrogen, and stored at −80 °C until use. Total RNA was extracted using a standard Trizol (Novel-Gene) extraction protocol. In brief, frozen guts were homogenized in 200 μl of Trizol, followed by a phase separation step with 200 μl 1-bromo-3-chloropropane (Sigma-Aldrich). RNA was precipitated using 500 μl isopropanol. The pellet was washed with 75% ethanol twice and re-suspended in 40 μl RNase-free water. RNA-seq was performed according to the standard Illumina protocol (Welgene, Taiwan). In brief, RNA was quantified. Then, library preparation was carried out by SureSelect XT HS2 mRNA Library Preparation kit (Agilent), followed by AMPure XP beads (Beckman Coulter, USA) for size selection. The sequence was determined using Illumina's sequencing-by-synthesis technology; sequencing data (FASTQ reads) were generated based on Illumina's basecalling program bcl2fastq v2.20. TRimmomatic v0.36 was used for adaptor clipping and sequence quality trimming with a sliding-window approach. HISAT2 was used for transcriptome alignment. Differential expression analysis was performed using StringTie (StringTie v2.1.4) and DEseq (DEseqv1.39.0) or DEseq2 (DEseqv1.28.1) via an in-house pipeline from Welgene Biotech. ClusterProfiler v3.6 was used to perform functional enrichment of differentially expressed genes. The gene expression levels were calculated as fragments per Kb of transcript per million

mapped reads (FPKM). Genes with low expression levels (Transcripts per million, TPM < 0.3) were discarded.

## RNA immunoprecipitation (RIP)

For each genotype, 200–300 guts from 1-week-old females were dissected per replicate, flash-frozen in liquid nitrogen, and stored at –80 °C until use. Total RNA was extracted using the standardized protocol described above. m$^7$G-modified RNAs were immunoprecipitated using a mouse anti-m$^7$G antibody (MBL, RN017M; 1:1000), as previously described (Pandolfini et al, 2019). In brief, 20 μg of extracted RNA was subjected to decapping by using yDcpS (NEB, M0463S), 10× reaction buffer, RNAsin Plus (Promega, N2611), and nuclease-free water for 1 h at 37 °C, followed by purification using the RNA Clean and Concentrator-5 kit (Zymo Research, R1017). The decapped RNA was denatured for 2 min at 80 °C and placed immediately on ice. A 1 μg aliquot of RNA was saved as an INPUT control. For RIP, the rest of the decapped RNA was mixed with 10 μg of anti-m$^7$G antibody (MBL Life Sciences, RNO17M), 10 μl RNaseOUT (Thermo Fisher, 10777019), 200 μl of 5× RIP buffer (50 mM Tris-HCl of pH 7.4, 750 mM NaCl, 5% NP-40 (w/v), and the rest ddH$_2$O to 1 ml for 2 h at °C in rotation.

The Dynabeads Protein G (Thermo Fisher, 10004D) were washed and blocked with BSA (20 μg/μl) in 1× RIP buffer for 2 h at 4 °C with rotation. After washing, 100 μl of beads were added to each sample and incubated for 2 h at 4 °C with rotation. Beads were then washed three times with ice-cold 1× RIP buffer, and bound RNA was eluted in elution buffer at 37 °C for 30 min. The INPUT and RIP samples (adjusted to 100 μl in elution buffer) were purified using RNA Clean and Concentrator-5 kit. Then they were finally eluted using 10–15 μl in ddH$_2$O, quantified by Qubit RNA High Sensitivity Kit (Thermo Fisher Scientific, Q32852), and followed by miRNA qRT-PCR.

## miRNA qRT-PCR

miRNA expression was measured using a TaqMan™ MicroRNA Assay Kit (Applied Biosystems, 4427975), according to the manufacturer's instructions. Briefly, 1–10 ng RNA was reverse transcribed using MultiScribe™ reverse transcriptase to generate cDNA for the *let-7* miRNA, and *snoRNA227* (housekeeping lncRNA) was used as a control reaction, with primers designed by Applied Biosystems. The PCR reaction was carried out using TaqMan® 2× Universal PCR Master Mix (No AmpErase® UNG, Applied Biosystems) after mixing cDNA with a small miRNA buffer containing designed primers specific for cDNA miRNA (1:15 dilution from RT reaction). The TaqManTM miRNA assay ID used in this study is provided in Reagents and Tools Table.

The % in input (the relative amount of immunoprecipitated miRNA compared to input RNA after qRT-PCR) was calculated by the formula:

$$\Delta Ct[\text{Normalized RIP}] = [Ct[IP] - \log 2(d_{IP})] - [Ct[In] - \log 2(1/f_{in}) - \log 2(d_{in})].$$

% of input = $100 \times 2^{-\Delta Ct}$Ct[IP], Ct[In]: raw Ct of target in Input and Immunoprecipitated RNA. $f_{in}$: fraction of input saved. $d_{IP}$, $d_{in}$: PCR dilutions for IP and Input.

## Proximity labeling assay (PLA)

PLAs were performed using the Duolink In Situ PLA kit (Sigma-Aldrich, DUO 9200), according to the manufacturer's instructions with modifications. In brief, the guts were dissected and fixed as previously described. After fixation, the guts were blocked in the PLA blocking buffer for 1 h at 37 °C with gentle shaking. The samples were then incubated with the primary antibody in antibody diluent O/N at 4 °C. The guts were washed three times for 40 min with 0.1% PBST and incubated with 1:5 dilutions of PLA probes: anti-rabbit PLUS (Sigma-Aldrich, DUO92002) and anti-mouse MINUS (Sigma-Aldrich, DUO92004) at 37 °C for 1 h. The guts were then washed twice with wash buffer A from the kit for 5 min, followed by 30-min ligation at 37 °C. The guts were further washed twice with buffer A for 5 min and then incubated in an amplification solution for 100 min at 37 °C. After sequential rinsing with buffer B and then 0.01× buffer B from the kit for 5 min each, samples were transferred to Duolink in situ mounting medium with DAPI (DUO82040) and stored at 4 °C.

The following primary antibodies were used: rabbit anti-GFP (Torrey Pines Biolabs, #TP401, 1:200) and mouse anti-V5 (Invitrogen, R960-25, 1:100). Guts were imaged as described above.

## Sampling, quantification, and statistical analyses

To ensure reproducibility, crosses for each genotype were set up using 15 females and 10 males, and 20–25 progeny were collected over a 2-day period at the designated temperature. From these, 10–12 female flies were randomly selected for gut dissection. For all fluorescence intensity and signal analyses, four to five images were acquired per PMG along the anterior-posterior axis, with ~8 z-sections merged to cover approximately half the gut thickness. In each merged image, 5–10 randomly selected cells were analyzed using ZEN software (ZEISS, Germany). For each PMG, 3–5 sections were examined, yielding an average of 20–30 measurements. A total of 9–10 PMGs per replicate were analyzed to obtain average values. All key experiments were independently repeated in at least two biological replicates and validated using either two independent *RNAi* lines or a combination of genetic and pharmacological approaches.

For morphological analysis of fly guts, images were captured after tissue fixation using a Zeiss SV11 Apo, and the images were then analyzed by ImageJ (NIH, USA). A spline line was drawn along the entire gut region to yield gut length, while three straight lines were drawn at random intervals through the anterior midgut and the PMG region to estimate thickness.

Proportion of each cell type in the PMG was calculated from the numbers of each cell type (according to marker labeling, see Fig. 2C) and the total cell number in two random square frames of 100 × 100 μm from each z-section merged image.

To calculate the nucleolar-to-nuclear ratio, the area of Fibrillarin signals was then measured in selected *esg > GFP* or *Dl > GFP* cells (with small nuclei) per gut section and normalized to their respective nuclear area.

To measure Dlg expression at EC-EC junctions, continuous lines were manually drawn around Dlg and LamC signals in ECs using ZEN software. The average fluorescence intensity of Dlg was normalized to the corresponding average LamC intensity within the same cell.

For F-actin fiber morphology analysis in the PMG, actin filaments in each z-section-merged image were classified as normal if they appeared long, continuous, and showed only minor deviations. In contrast, filaments exhibiting multiple lateral projections or mid-fiber breaks were classified as disorganized.

Levels of p-4EBP, dMyc, and pJNK in *Dl>gfp* or *esg>gfp* cells with small nuclei were measured and normalized to the nearby background, as described previously (Haller et al, 2017). For pJNK measurements in $wh^7$ mutant PMGs, where pJNK is broadly elevated, signal intensity in cells was normalized to the neighboring background, as described earlier.

For mitotic cell analysis, PHH3-positive cells were counted from the entire midgut under the microscope.

PLA puncta were manually counted in the small cells that expressed *esg>dMettl1-V5* and *dwdr4-gfp* using z-section-merged images obtained along the PMG.

Data were compiled in Excel (Microsoft) and graphed in either Excel or Prism 9.0 (GraphPad Software). *P* values were calculated using the Chi-square test, two-tailed unpaired Student's *t* test for normally distributed data, or the non-parametric Mann–Whitney *U* test for skewed distributions or outliers, as appropriate, with Excel or GraphPad Prism. $P < 0.05$ was considered a statistically significant difference.

## Data availability

RNA-seq data have been deposited in GEO under accession number GSE298087.

The source data of this paper are collected in the following database record: biostudies:S-SCDT-10_1038-S44319-026-00701-y.

## Peer review information

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

## Acknowledgements

We thank GC Chen, JJC Sheu, R Gregory, B Lemaitre, LE O'Brien, TF Fu, the BDSC, and the DSHB for fly stock reagents and antibodies. We thank the Drosophila Genomics Resource Center (supported by NIH Grant 2P40OD010949) for the *UASz* plasmid. We thank the Taiwan flycore for ordering fly lines and reagents, the core facilities in the Institute of Cellular and Organismic Biology, Academia Sinica, for assistance with image analysis, and Marcus Calkins for English editing. We also thank RH Chen, GC Chen, HD Wang, WW Hwang-Verslues, and B Boumard for the critical review of the manuscript. Ministry of Science and Technology, Taiwan [grant 111-2311-B-001-022 and 112-2311-B-001-026] supported this work. Work in the Venkatachalam lab is supported by National Institutes on Aging (USA) grants RF1AG072176 and RF1AG069076.

## Author contributions

**Kreeti Kajal**: Conceptualization; Formal analysis; Investigation; Methodology; Writing—original draft; Writing—review and editing. **Elham Rastegari**: Conceptualization; Methodology; Writing—review and editing. **Wen-Der Wang**: Conceptualization; Writing—review and editing. **Jian-Chiuan Li**: Investigation. **Chun-Hong Chen**: Investigation. **Wan Hsuan Chou**: Formal analysis. **Wen Chiao Chang**: Investigation. **Tzu-Yang Lin**: Investigation. **Kevin Tsai**: Investigation. **Tsai Ming Lu**: Formal analysis. **Kartik Venkatachalam**: Conceptualization; Methodology; Writing—original draft; Writing—review and editing. **Hwei-Jan Hsu**: Conceptualization; Supervision; Methodology; Writing—original draft; Writing—review and editing.

Source data underlying figure panels in this paper may have individual authorship assigned. Where available, figure panel/source data authorship is listed in the following database record: biostudies:S-SCDT-10_1038-S44319-026-00701-y.

## Disclosure and competing interests statement

The authors declare that the research was conducted in the absence of any commercial or financial relationships that could be construed as a potential conflict of interest.

# Expanded View Figures

**Figure EV1. Loss of dWdr4 shortens lifespan, impairs locomotor activity, and disrupts gut function in flies.**

(A, B) *Drosophila dwdr4* mutant (*wh*[7]) female (A) and male flies (B) show shortened lifespans compared to wild-type (WT) controls. Lifespan defects were rescued by introducing a copy of *dwdr4-gfp* ($N = 80$ flies). (C, D) Climbing assays reveal reduced locomotor activity in *wh*[7] mutant female (C) and male flies (D) compared to WT control. Climbing performance was partially improved in females ($N = 17$ flies) and completely rescued in males ($N = 15$ flies) by *dwdr4-gfp* expression. (E) dWdr4 loss causes the accumulation of lipid droplets. WT ($N = 7$) and *wh7* mutant ($N = 9$) guts with Bodipy (green, oil droplets), Phalloidin (magenta, F-actin), and DAPI (blue, nuclei). The fraction of gut sections with the pattern shown is indicated. (F) Expression differences in genes involved in lipid metabolism and gut digestion between the *wh*[7] and WT female guts. Fold change of $\log_2$ values ($P < 0.05$) is indicated by the color indicator. Each dot in (C, D) represents one fly. Scale bar in (E) is 50 μm. Bars represent the mean; error bars indicate standard deviation (SD). Statistically significant differences in (A, B) were identified by the Log-rank test, and in (C, D) were identified by the unpaired Student's *t* test. Statistical analysis in (F) was performed on two biological replicates. Lifespan analysis in (A, B) showed significant differences between WT and *wh*[7], *wh*[7] and *wh*[7];*dwdr4-gfp*, and between WT and *wh*[7];*dwdr4-gfp* (\*\*\*$P = < 0.0001$). Climbing assay in (C, D) showed significant differences between WT and *wh*[7] (\*\*\*$P = < 0.0001$), *wh*[7] and *wh*[7];*dwdr4-gfp* (\*\*$P = 0.0014$), and between WT and *wh*[7];*dwdr4-gfp* (\*\*\*$P = 0.0004$) in females, but in males, no differences were observed between WT and *wh*[7];*dwdr4-gfp* (ns, $P = 0.621$). *N*, number of midguts, unless otherwise stated. Source data are available online for this figure.

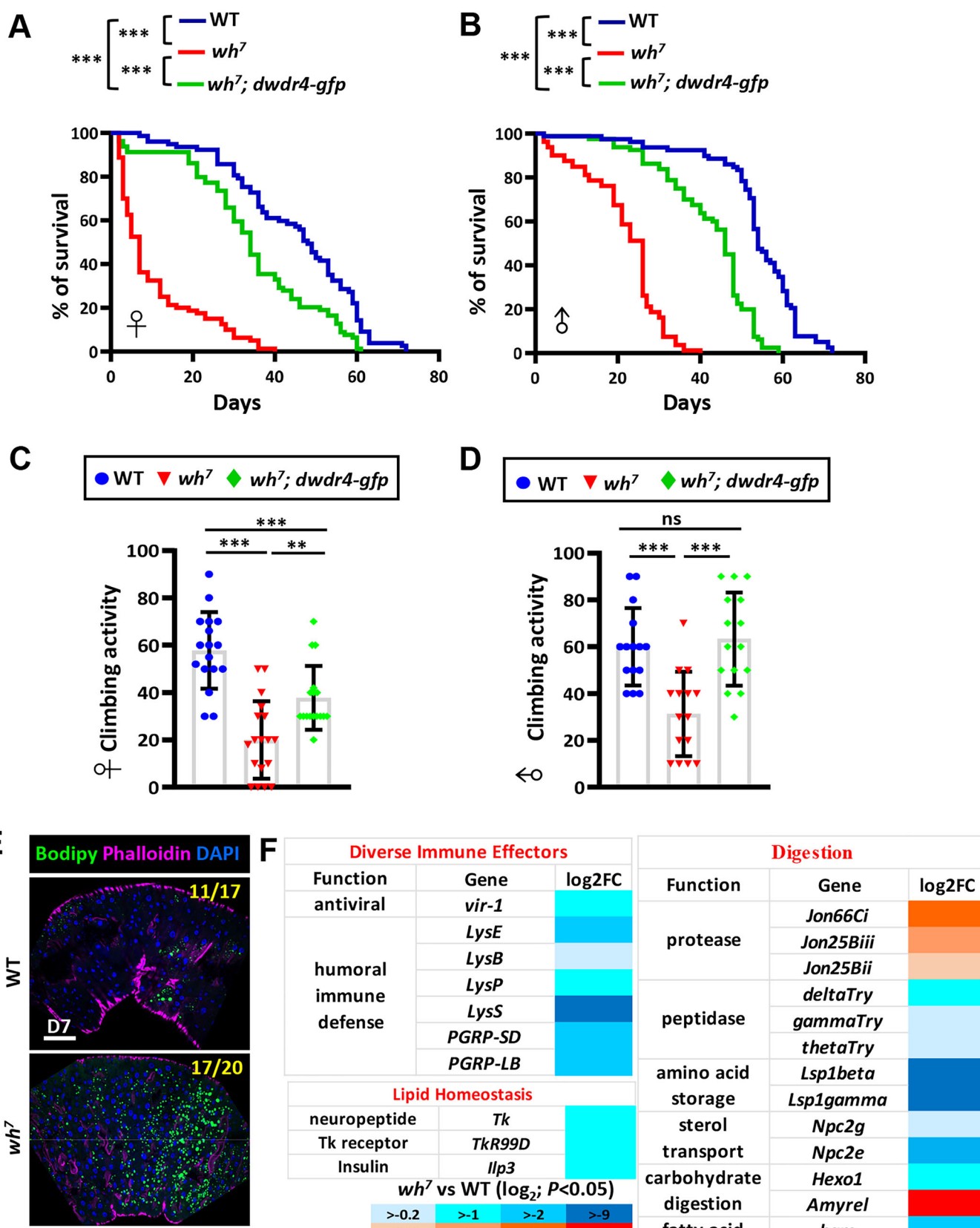

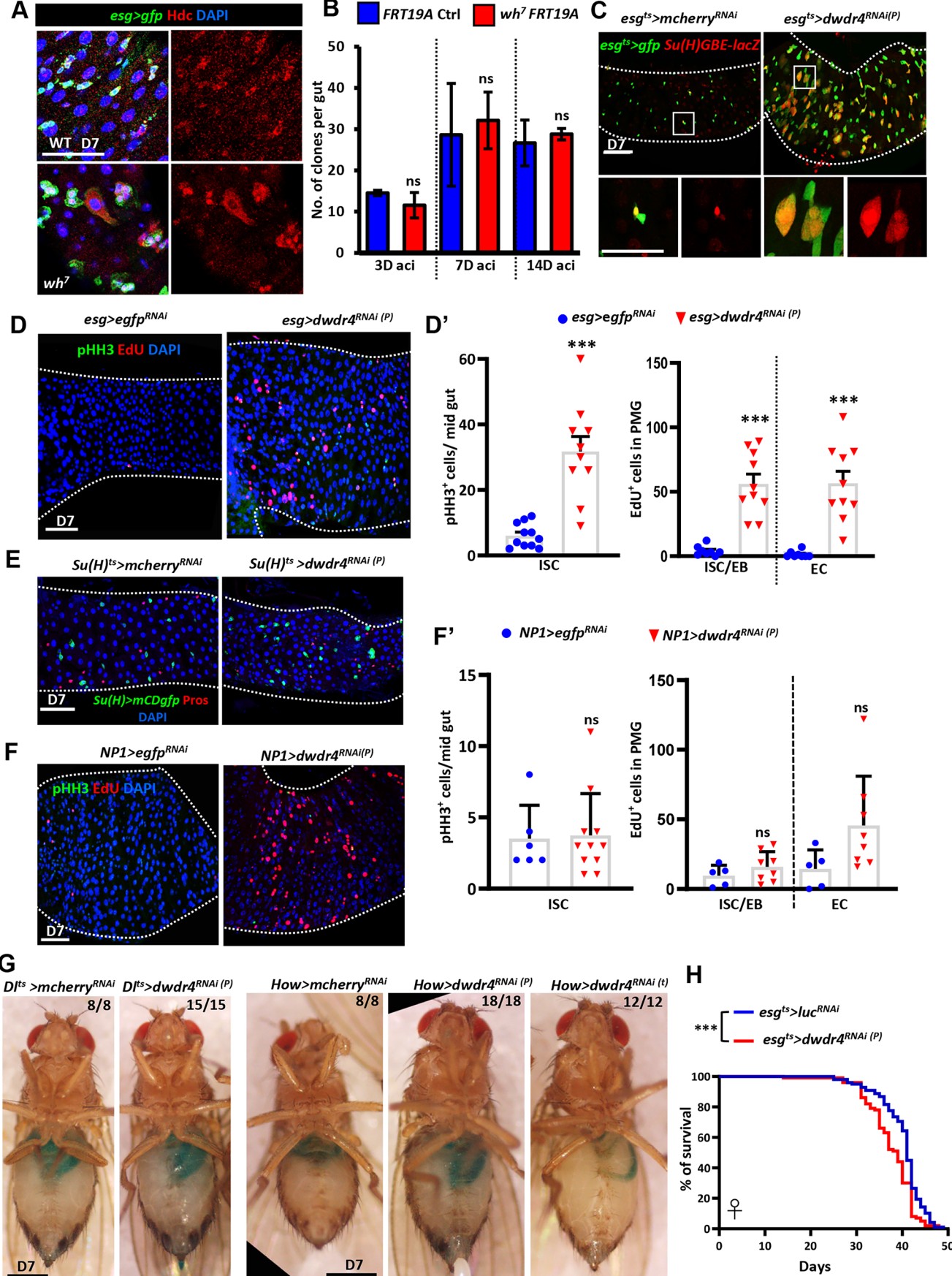

◀ **Figure EV2. Loss of dWdr4 in ISCs promotes their proliferation and shortens lifespan but does not compromise intestinal epithelial integrity.**

(A) Posterior midguts (PMGs) from wild-type (WT) and *dwdr4* mutant (*wh⁷*) females expressing *esg>gfp* (green, ISCs, EBs), stained for Hdc (red, ISCs/EBs) and DAPI (blue, nuclei). In the WT control PMG ($N = 10$), the ISC-EB two-cell cluster has both cells expressing Hdc, whereas in the *dwdr4* mutant (*wh⁷*) PMG ($N = 13$), Hdc expression was seen in all *esg>gfp* cell clusters as well as in misdifferentiated ECs. (B) Number (no.) of clones per midgut in *FRT19A* control (ctrl) (3D: $N = 27$, 7D: $N = 30$, 14D: $N = 34$) and *wh⁷FRT19A* heterozygous mutant guts (3D: $N = 29$, 7D: $N = 28$, 14D: $N = 28$) at 3, 7, and 14 days after clone induction (aci). (C) *esgᵗˢ>dwdr4^{RNAi(P)}* PMG ($N = 7$) shows expansion of ISCs/EBs (green, labeled with *esgᵗˢ>gfp*) with some ISCs/EBs also expressing *Su(H)-lacZ* (red, a Notch signaling reporter used as an EB marker). In the *esgᵗˢ>mcherry^{RNAi}* control PMG ($N = 5$), the ISC-EB two-cell cluster has only one *Su(H)-lacZ*-positive cell. Enlarged views of the white boxed regions are shown below. (D, D') *esg>dwdr4^{RNAi(p)}* PMG ($N = 10$) shows increased mitotic ISCs (PHH3⁺, green), proliferating ISCs/EBs (EdU⁺ with the small nuclei, red), and endoreplicating ECs (EdU⁺ with the large nucleus, red), as compared to the control (*esg>egfp^{RNAi}*) ($N = 8$). (D') Number of PHH3⁺ ISCs per midgut (left) and number of EdU⁺ ISCs/EBs and ECs per PMG (right) for the indicated genotypes. (E) Knockdown of *dwdr4* in EBs (*Su(H)ᵗˢ>dwdr4^{RNAi(P)}*) ($N = 22$) causes no obvious change in the EB population (*Su(H)ᵗˢ>mcD8gfp*, green) compared to *Su(H)ᵗˢ>mcherry^{RNAi}* control PMGs ($N = 20$). Pros, red, pEEs/EEs. (F, F') Knockdown of *dwdr4* in ECs (*NP1>dwdr4^{RNAi(P)}*) ($N = 11$) does not increase mitotic ISCs (PHH3⁺, green) or proliferating ISCs/EBs (EdU⁺) compared to the control (*NP1>egfp^{RNAi}*) ($N = 6$). Still, it slightly increases endoreplicating ECs (EdU⁺ with the large nucleus, red) compared to the control. DAPI, blue. (F') Number of PHH3⁺ ISCs per midgut (left) and number of EdU⁺ ISCs/EBs and ECs per PMG (right) for the indicated genotypes. (G) *Dlᵗˢ>dwdr4^{RNAi(P)}*, *how>dwdr4^{RNAi(P)}*, and *how>dwdr4^{RNAi(t)}* female flies do not show the leaky gut phenotype, as revealed by the smurf assay. Fractions of flies with the pattern shown are indicated. (H) Female flies carrying *esgᵗˢ>dwdr4^{RNAi(P)}* have a shortened lifespan compared to control (*esgᵗˢ>LucRNAi*) ($N = 100$ flies). Each dot in (D', F') represents one midgut. The scale bar in (A) and enlarged views of (C) are shown at 25 µm; (C, D, E, F) are 50 µm, and in (G) is 0.5 mm. White dashed lines in (C, D, E, F) outline the edge of the gut. Bars represent the mean; error bars indicate standard deviation (SD). Statistically significant differences in (B) were identified by the unpaired Student's *t* test, in (D', F') by the non-parametric Mann–Whitney *U* test, and in H by the Log-rank test. (B) No differences were observed between *FRT19A* ctrl and *wh⁷ FRT19A* in clones per gut after 3D aci (ns, $P = 0.23$), 7D aci (ns, $P = 0.69$), and 14D aci (ns, $P = 0.57$). (D') PHH3⁺ cells were significantly increased in *esg>dwdr4^{RNAi(P)}* compared with *esg>egfp^{RNAi}* (***$P = < 0.0001$); EdU⁺ ISC/EB (***$P = < 0.0001$) and EC cells (***$P = < 0.0001$) showed significant increase. (F') no differences in PHH3⁺ cells were observed in *NP1>dwdr4^{RNAi(P)}* compared with *NP1>egfp^{RNAi}* (ns, $P = 0.9$); EdU⁺ ISC/EB (ns, $P = 0.3$) showed no difference; whereas in EC cells (ns, $P = 0.6$) increased slightly. Lifespan analysis in (H) showed a significant reduction in *esgᵗˢ>dwdr4^{RNAi(P)}* compared with *esg>egfp^{RNAi}* (***$P = 0.0003$). *N*, number of midguts; unless otherwise stated. Source data are available online for this figure.

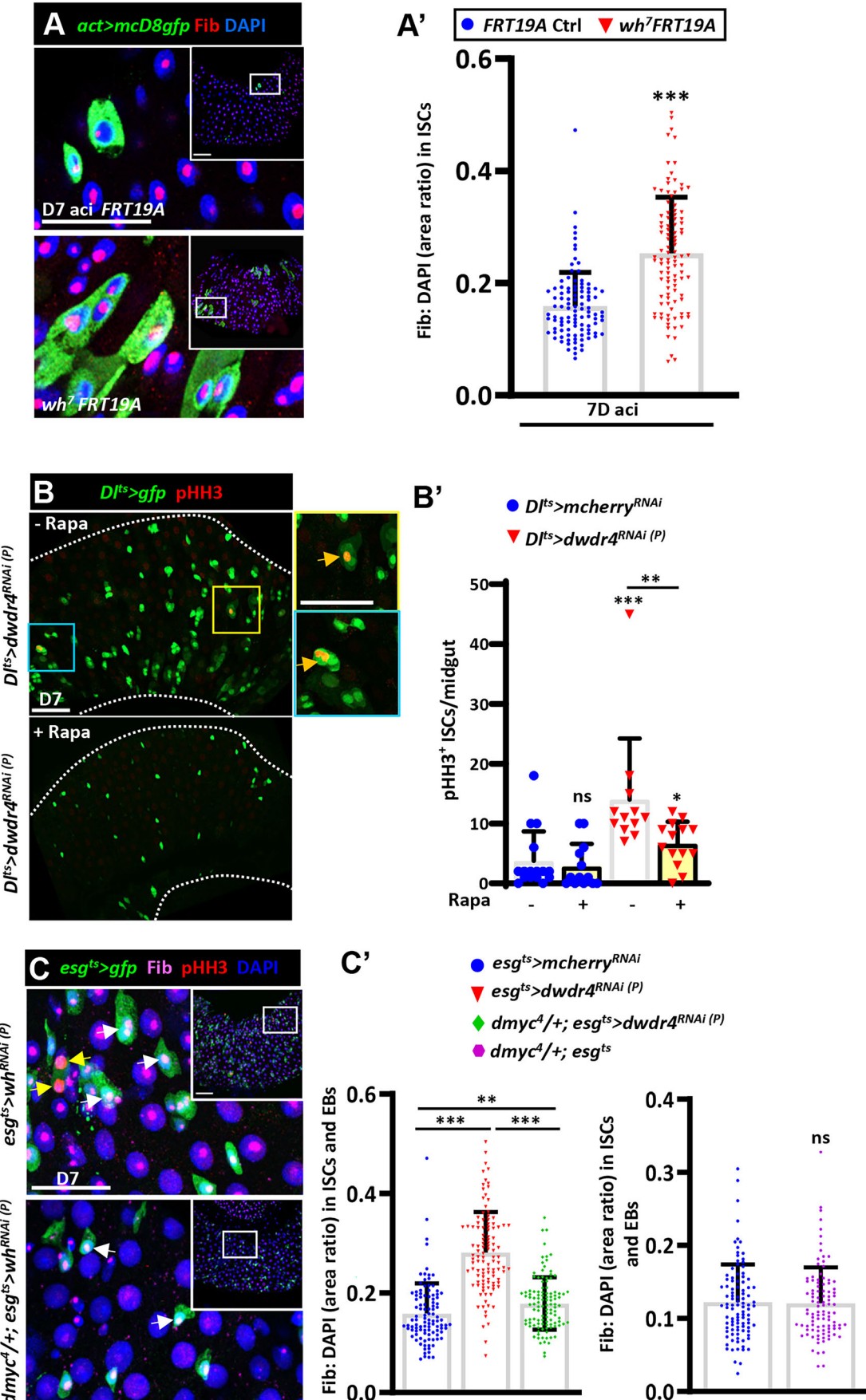

**Figure EV3. Wdr4 regulates ISC proliferation via TOR and dMyc.**

(A, A') dWdr4 cell-autonomously controls ribosome biogenesis in ISCs/EBs. (A) Mosaic *FRT19A* control ($N = 11$) and *wh⁷FRT19A* heterozygous mutant posterior midguts (PMGs) ($N = 10$) ($n = 103$ ISC/EB cells) with *act>mCD8gfp* (green, mutant cells), Fibrillarin (Fib) (red, nucleoli), and DAPI (blue, nuclei) labels. (A') Quantification of nucleolar (Fib) to nuclear (DAPI) area ratio in GFP-positive cells with small nuclei in the indicated genotypes, at 7 days after clone induction (aci). (B, B') Rapamycin treatment suppresses ISC expansion and overproliferation caused by dWdr4 depletion. (B) PMGs from *Dlᵗˢ>dwdr4^{RNAi(P)}* flies expressing *Dlᵗˢ>gfp* (green, ISCs) with or without rapamycin (rapa) treatment, stained for PHH3 (red, mitosis marker). Enlarged views of the boxed regions are shown on the right. (B') Quantification of PHH3⁺ ISCs per midgut for the *Dlᵗˢ>dwdr4^{RNAi(P)}* (dH₂O: $N = 12$ midguts; Rapa: $N = 14$) and *Dlᵗˢ>mcherry^{RNAi}* (dH₂O: $N = 15$; Rapa: $N = 13$). (C, C') Removal of a copy of dMyc suppresses elevated ribosome biogenesis induced by dWdr4 depletion. (C) PMGs from *esgᵗˢ>dwdr4^{RNAi(P)}* and *dmyc⁴/+; esgᵗˢ>dwdr4^{RNAi(P)}* females bearing *esgᵗˢ>gfp* (green, ISCs/EBs), stained for PHH3 (red, mitosis marker) and Fibrillarin (Fib) (magenta, nucleoli) and DAPI (blue, nuclei). These images are also shown in Fig. 5F. (C') Quantification of nucleolar (Fib) to nuclear (DAPI) area ratio in ISCs/EBs in the *esgᵗˢ>mcherry^{RNAi(P)}* ($N = 14$), *esgᵗˢ>dwdr4^{RNAi(P)}* ($N = 8$), *esgᵗˢ>dwdr4^{RNAi(P)}* with *dmyc⁴* mutant ($N = 10$), and *dmyc⁴* mutant alone ($N = 16$) (111 ISC/EB cells) flies. Yellow arrows in (B, C) indicate representative mitotic ISCs, and white arrows mark representative ISC or EB nucleoli. Insets in (B) show enlargements of boxed regions; (A, C) show enlargements of boxed regions within insets. Each dot in (A', C') represents a single ISC/EBs, and each dot in (B') represents one midgut. The low magnification and enlarged views of (A, C) are shown at 50 and 25 μm; (B) is 50 μm. White dashed lines in (B) outline the edge of the gut. Bars represent the mean; error bars indicate standard deviation (SD). Statistically significant differences in (A', B', C') were identified by the non-parametric Mann–Whitney *U* test. (A') The nucleolar size differed in *FRT19A* and *wh⁷FRT19A* (***$P = < 0.0001$). (B') PHH3⁺ cells were unchanged by rapamycin in *Dlᵗˢ>mcherry^{RNAi}* (ns, $P = 0.32$), but differed between *Dlᵗˢ>mcherry^{RNAi}* and *Dlᵗˢ>dwdr4^{RNAi(P)}* with (*$P = 0.03$) and without rapamycin (***$P = < 0.0001$); in *Dlᵗˢ>dwdr4^{RNAi(P)}*, rapamycin significantly reduced PHH3⁺ cells (**$P = 0.002$). (C', left graph) The nucleolar size showed differences in *esgᵗˢ>dwdr4^{RNAi(P)}* compared to *esgᵗˢ>mcherry^{RNAi(P)}* without *dmyc⁴* (***$P = < 0.0001$) and with *dmyc⁴* (**$P = 0.0012$); nucleolar size differed significantly between *esgᵗˢ>dwdr4^{RNAi(P)}* with and without *dmyc⁴* (***$P = < 0.0001$); whereas *dmyc⁴* alone had no effects (ns, $P = 0.75$) (right graph). *N*, number of midguts; *n*, number of ISCs or ISC/EBs analyzed, unless otherwise stated. Source data are available online for this figure.

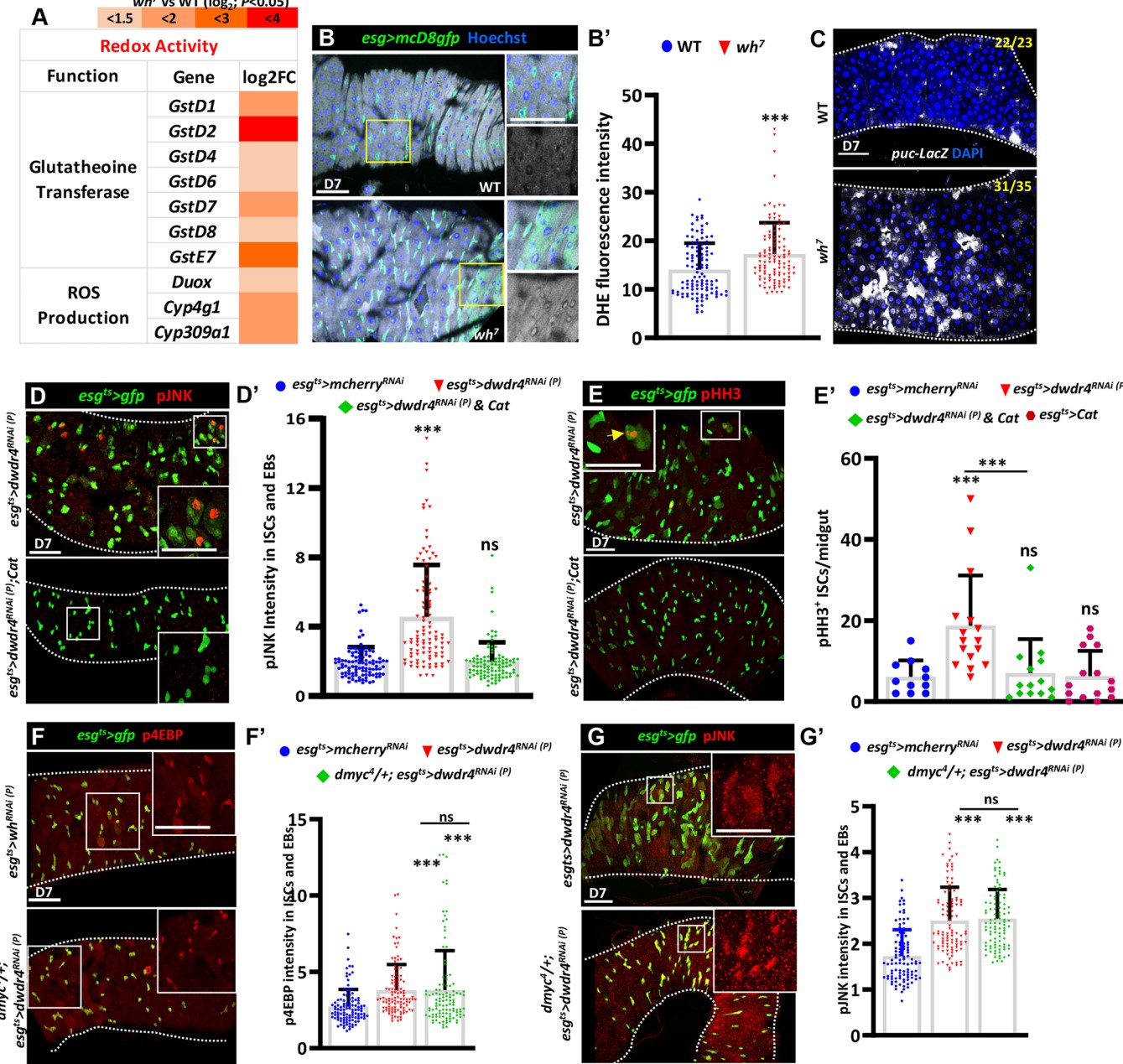

◄ **Figure EV4. Loss of Wdr4 in the gut increases expression of genes involved in stress and redox responses as well as JNK signaling.**

(A) RNA-seq analysis results show that the expression levels of genes involved in redox activity and stress are increased in the guts of *dwdr4* (*wh*[7]) mutant female guts. Fold change of $\log_2$ values (*P* < 0.05) is indicated by color. Statistical analysis was performed on two biological replicates. (B, B') *wh*[7] mutant PMG bearing *esg>mCD8gfp* (*N* = 15) (green, ISCs/EBs) show increased ROS levels compared to WT controls (*N* = 16), (*n* = 110 ISC/EB cells), as detected by DHE staining (gray). (B') Quantification of DHE fluorescence intensity. DAPI (bule) label nuclei. (C) *wh*[7] mutant PMG shows increased JNK signaling compared to controls (*N* = 7), as revealed by *puc-lacZ* (gray, a JNK signaling reporter). DAPI, blue. Fractions of gut sections with the shown pattern are indicated. (D, E') Overexpression of Catalases (Cat) suppresses elevated pJNK and ISC expansion and overproliferation induced by dWdr4 depletion. (D) PMGs from the indicated genotypes bearing *esg*[ts]*>gfp* (green, ISCs/EBs), stained for pJNK (Red, JNK signaling activity). (D') Quantification of pJNK intensity in the *esg*[ts]*>mcherry*[RNAi(P)] (*N* = 11), *esg*[ts]*>dwdr4*[RNAi(P)] (*N* = 16), and *esg*[ts]*>dwdr4*[RNAi(P)] with *cat* overexpression (*N* = 16) (*n* = 100 ISC/EB cells) flies. (E) PMGs from the indicated genotypes bearing *esg*[ts]*>gfp* (green), stained for PHH3 (red, mitotic marker). (E') Quantification of PHH3[+] ISCs per midgut for the *esg*[ts]*>mcherry*[RNAi(P)] (*N* = 11), *esg*[ts]*>dwdr4*[RNAi(P)] (*N* = 16), *esg*[ts]*>dwdr4*[RNAi(P)] with *cat*[OE] (*N* = 14), and *cat*[OE] alone (*N* = 15) flies. (F, G) dMyc acts downstream of TOR and JNK signaling in ISCs for ISC homeostasis. (F, F') Loss of one *dmyc* copy in *esg*[ts]*>dwdr4*[RNAi(P)] posterior midguts (PMGs) fails to suppress ISC/EB expansion (*esg*[ts]*>gfp*, green) or the increased TOR signaling (Red, TOR signaling activity). (F') Quantification of average p4EBP intensity in the *esg*[ts]*>mcherry*[RNAi(P)] (*N* = 11), *esg*[ts]*>dwdr4*[RNAi(P)] (*N* = 12), *esg*[ts]*>dwdr4*[RNAi(P)] with *dmyc*[4] mutant (*N* = 10) (*n* = 110 ISC/EB cells) flies. (G, G') Removal of one copy of *dmyc* in *esg*[ts]*>dwdr4*[RNAi(P)] PMGs does not suppress the increased JNK signaling (Red, JNK signaling activity). (G') Quantification of average pJNK intensity in the *esg*[ts]*>mcherry*[RNAi(P)] (*N* = 9), *esg*[ts]*>dwdr4*[RNAi(P)] (*N* = 16), *esg*[ts]*>dwdr4*[RNAi(P)] with *dmyc*[4] mutant (*N* = 8) (*n* = 110 ISC/EB cells) flies. Insets in (B, D, F, G) show enlarged views of the boxed regions. Each dot in (B', D', F', G') represents a single ISC or EB, and each dot in (E') represents one midgut. The image and insets of (B, D–G) are shown at 50 and 25 μm. White dashed lines in (C–G) outline the edge of the gut. Bars represent the mean; error bars indicate standard deviation (SD). Statistically significant differences in (B'–G') were identified by the non-parametric Mann–Whitney *U* test. (B') The DHE signals increased in *wh*[7] compared to WT (\*\*\**P* = < 0.0001). (D') The pJNK intensity showed differences in *esg*[ts]*>dwdr4*[RNAi(P)] compared to *esg*[ts]*>mcherry*[RNAi(P)] without *cat*[OE] (\*\*\**P* = < 0.0001), but not with *cat*[OE] (ns, *P* = 0.7). (E') The PHH3[+] cells showed differences in *esg*[ts]*>dwdr4*[RNAi(P)] compared to *esg*[ts]*>mcherry*[RNAi(P)] without *cat*[OE] (\*\*\**P* = 0.0002), but not with *cat*[OE] (ns, *P* = 0.67); PHH3[+] cells differed between *esg*[ts]*>dwdr4*[RNAi(P)] with and without *cat*[OE] (\*\*\**P* = 0.0003); whereas *cat*[OE] alone had no effects (ns, *P* = 0.58). (F', G') p4EBP and pJNK levels showed differences in *esg*[ts]*>dwdr4*[RNAi(P)] compared to *esg*[ts]*>mcherry*[RNAi(P)] without *dmyc*[4] and with *dmyc*[4] (\*\*\**P* = < 0.0001); p4EBP and pJNK levels did not differ between *esg*[ts]*>dwdr4*[RNAi(P)] with and without *dmyc*[4] (ns, *P* = 0.52 in F', 0.42 in G'). *N*, number of midguts; *n*, number of ISCs or ISC/EBs analyzed, unless otherwise stated. Source data are available online for this figure.

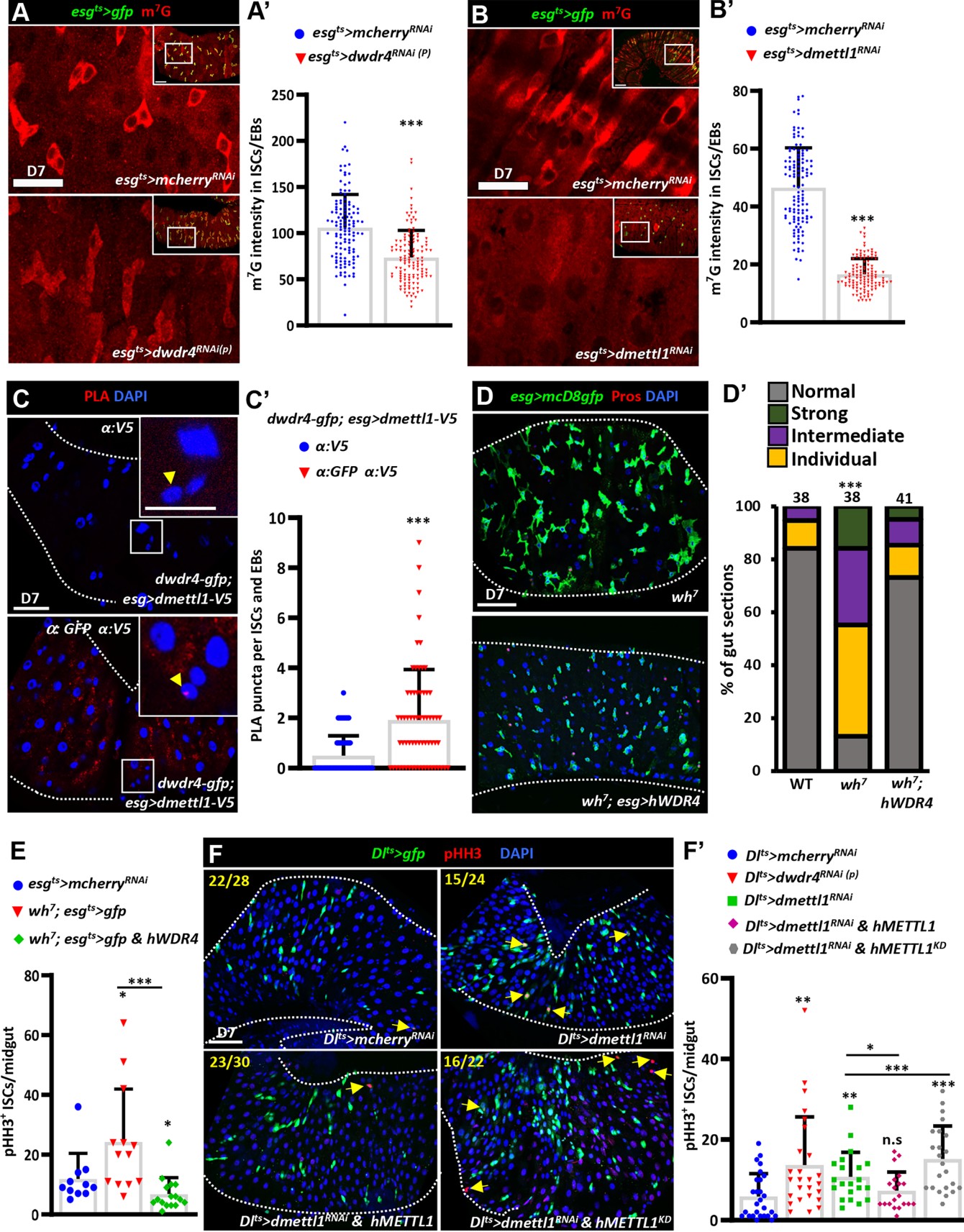

◀ **Figure EV5.** **The *Drosophila* Wdr4-Mettl1 complex mediates m⁷G modification to maintain intestinal homeostasis in a conserved manner.**

(A, B') dWdr4 and dMettl1 control m⁷G levels in ISCs/EBs. (A, B) Posterior midguts (PMGs) from *esg^ts^>dwdr4^RNAi(P)^* (A) and *esg^ts^>dmettl1^RNAi^* (N = 16) (B) females show decreased m⁷G modification (red) in ISCs/EBs (*esg^ts^>gfp*, green) as compared to control (*esg^ts^>mcherry^RNAi^*) (N = 12) (n = 125 ISC/EB cells). (A', B') Quantification of average m⁷G fluorescence intensity in ISCs/EBs in the indicated genotypes. (C) dWdr4 interacts with dMettl1 in ISCs/EBs, as revealed by PLA assay (red punctate) using anti-GFP and anti-V5 antibodies in PMGs from *dwdr4-gfp; esg>dmettl1-V5* females. *dmettl1-V5* is expressed in ISCs/EBs under the control of *esg-GAL4*, while *dwdr4-gfp* is expressed in all intestinal cells. PMGs of the same genotype exposed only to anti-V5 antibody served as a negative control. The arrowhead in C indicates the PLA puncta. DAPI (blue) labels nuclei. (C') Quantification of PLA puncta number in the *dwdr4-gfp; esg>dmettl1-V5* females with and without anti-GFP antibody (N = 8, n = 65 ISC/EB cells). (D) Overexpression of human WDR4 in ISCs/EBs of *dwdr4* mutant (*wh⁷*) ISCs/EBs (labeled by *esg>mCD8gfp*, green) suppresses ISC/EB expansion. Pros, red. DAPI, blue. The *wh⁷* mutant image was obtained together with Fig. 2A and is shown here for comparison with the human gene rescue. (D') Quantification of ISC/EB expansion phenotypes categorized and color-coded as Fig. 2A. The number of gut sections analyzed (N = 9) per genotype is shown above each bar. (E) Overexpression of human WDR4 in *dwdr4* mutant ISCs/EBs (*esg^ts^>gfp*) decreases mitotic ISCs (PHH3⁺) in midguts. Number of PHH3⁺ cells per midgut in the *esg^ts^>mcherry^RNAi(P)^* (N = 11), *wh⁷* mutant with *esg>gfp* (N = 13), *wh⁷* mutant with *esg>gfp* and human WDR4 overexpression (N = 16) flies. (F) Overexpression of human METTL1, but not the catalytic-dead mutant (hMETTL1^KD^) in ISCs (labeled by *Dl^ts^>gfp*, green) of *dmettl1^RNAi^* knockdown PMGs suppresses the increased ISC expansion and proliferation. Yellow arrows in E indicate the representative mitotic ISCs. Fractions of gut sections with the ISC/EB pattern shown are indicated. (F') Quantification of PHH3⁺ ISC number per midgut in the *Dl^ts^>mcherry^RNAi(P)^* (N = 27), *Dl^ts^>dwdr4^RNAi(P)^* (N = 26), *Dl^ts^>dmettl1^RNAi^* (N = 21), with human METTL1 (N = 21), and hMETTL1^KD^ (N = 23) flies. DAPI, blue. Insets in (C) show enlarged views of the white square in each panel; (A, B) show enlarged views of the square regions in the insets. Each dot in (A', B', C') represents a single ISC or EB, and each dot in (E', F') represents one midgut. The low magnification and enlarged views of (A, B, C) are shown at 50 and 25 μm; (D, F) are 50 μm. White dashed lines in (C, D, F) outline the edge of the gut. Bars represent the mean; error bars indicate standard deviation (SD). Statistically significant differences in (A', B') were identified by unpaired Student's *t* test, in (C', E', F') were identified by non-parametric Mann–Whitney *U* test, and in (D') were analyzed by Chi-square. (A', B') m7G intensity decreased in *esg^ts^>dwdr4^RNAi(P)^* and *esg^ts^>dmettl1^RNAi^* compared to *esg^ts^>mcherry^RNAi^* (***P = < 0.0001). (C') The ***P = < 0.0001. (D') The ***P = 0.001. (E) The PHH3⁺ cells showed differences in *wh⁷* compared to WT without *hWDR4* (*P = 0.017) and with *hWDR4* (*P = 0.013); PHH3⁺ cells differed between *esg^ts^>dwdr4^RNAi(P)^* with and without *hWDR4* (***P = < 0.0001). (F') The PHH3⁺ cells showed differences in *Dl^ts^>dwdr4^RNAi(P)^* (**P = 0.002), *Dl^ts^>dmettl1^RNAi^* (**P = 0.004), *Dl^ts^>dmettl1^RNAi^* with *hMETTL1^CD^* (***P = < 0.0001) compared to *Dl^ts^>mcherry^RNAi(P)^*; but no difference in *Dl^ts^>dmettl1^RNAi^* with *hMETTL1* (ns, P = 0.16). PHH3⁺ cells differed in *Dl^ts^>dmettl1^RNAi^* with and without *hMETTL1* (*P = 0.04), but not with *hMETTL1^CD^* (*P = 0.0006). N, number of midguts; n, number of ISCs or ISC/EBs analyzed, unless otherwise stated. Source data are available online for this figure.

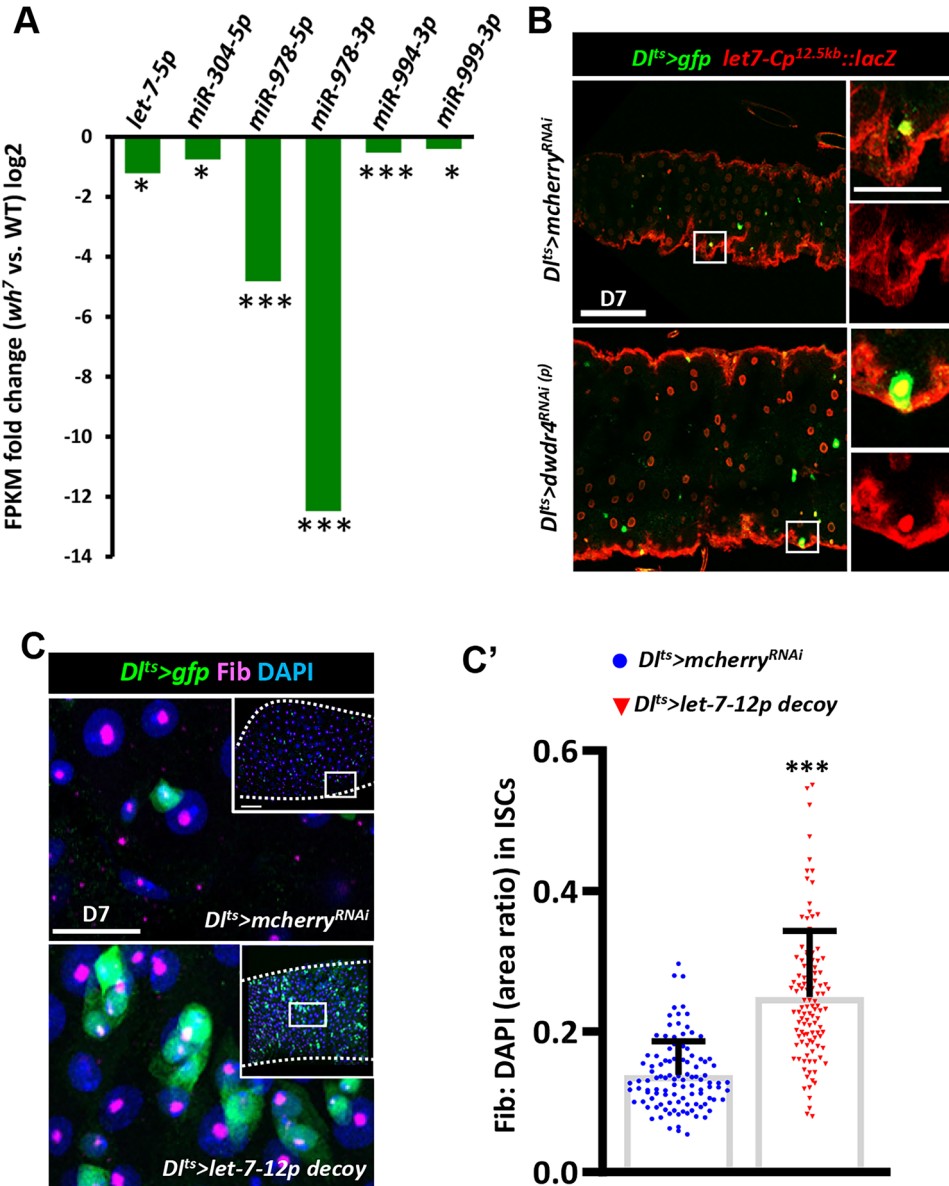

**Figure EV6. dWdr4 does not control *let-7* transcription and *let-7* disruption phenocopies *dwdr4* mutant PMGs.**

(A) Fold changes in gene expression according to RNA-seq (fragments per kilobase of transcript per million mapped reads [FPKM]) in wdr4 mutant (*wh⁷*) vs. wild-type (WT) ovaries; log₂ values are shown. Statistical analysis was performed on two biological replicates. *$P < 0.05$; ***$P < 0.001$. (B) Posterior midgut (PMG) from *Dl^ts^>dwdr4^RNAi(P)^* (N = 7), females bearing *Dl^ts^>gfp* (ISCs, green) show no obvious change in the expression of *let7-Cp^12.5kb^::lacZ* (red, transcriptional reporter of *let-7*) as compared to control (*Dl^ts^>mcherry^RNAi^*) (N = 6). (C) Disruption of *let-7* function (*Dl^ts^>let-7-12p decoy*) (N = 11) increases the size of Fibrillarin (Fib) (magenta, nucleoli) in ISCs (labeled by *Dl^ts^>gfp*, green) as compared to control (*Dl^ts^>mcherry^RNAi^*) (N = 9) (n = 110 ISC cells). DAPI (blue) marks nuclei. (C') Quantification of nucleolar (Fib) to nuclear (DAPI) area ratio in ISCs in the indicated genotypes. Insets in (B) show enlarged views of the boxed region in each panel; (C) shows enlarged views of the boxed region in the insets. Each dot in (C') represents a single ISC. The low magnification and enlarged views of (B, C) are shown at 50 and 25 µm. White dashed lines in (C) outline the edge of the gut. Bars represent the mean; error bars indicate standard deviation (SD). Statistically significant differences in (A) were identified by unpaired Student's *t* test, and in (C') were by non-parametric Mann–Whitney *U* test. (C') the nucleolar size is increased in *Dl^ts^>let-7-12p decoy* compared to *Dl^ts^>mcherry^RNAi^* (***$P = < 0.0001$). N, number of midguts; n, number of ISCs or ISC/EBs analyzed, unless otherwise stated. Source data are available online for this figure.

