## [Peer Review File · EMBO Reports]

Wdr4 Regulates Ribosome Biogenesis and Intestinal Homeostasis via let-7

Kreeti Kajal, Elham Rastegari, Wen-Der Wang, Jian-Chiuan Li, Chun-Hong Chen, Wan Hsuan Chou, Wen Chang, Tzu-Yang Lin, Kevin Tsai, Tsai Lu, Kartik Venkatachalam, and Hwei-Jan Hsu

Corresponding author(s): Hwei-Jan Hsu (cohsu@gate.sinica.edu.tw)

Review Timeline:

Submission Date:	15th Feb 24
Editorial Decision:	28th Mar 24
Appeal Received:	19th Sep 25
Editorial Decision:	21st Oct 25
Revision Received:	19th Nov 25
Editorial Decision:	12th Dec 25
Revision Received:	22nd Dec 25
Accepted:	21st Jan 26

Editor: Achim Breiling

Transaction Report:

Dear Prof. Hsu,

Thank you for the transfer of your research manuscript to EMBO reports. I have now received the full set of referee reports that are copied below.

As you will see, all three referees have several major concerns, indicating that the conclusions drawn are not sufficiently supported by the data (or even seem to contradict the data). Moreover, all referees also note several technical and experimental shortcomings. As the reports are below, I will not further detail them here.

Given the comments of the referees, the amount of work required to address them, the fact that EMBO reports can only invite revision of papers that receive overall positive support from all the referees upon initial assessment, I cannot offer to publish your manuscript.

I am sorry to have to disappoint you this time. I nevertheless hope that the referee comments will be helpful in your continued work in this area, and I thank you once more for your interest in our journal.

Yours sincerely

Referee #1:

I would like to start by apologising to the authors for my lateness in sending the review.

In their manuscript, Kajal et al. tackle the role of ribosome biogenesis in regulating stem cell activity by studying *Drosophila* Wdr4 effects in the adult intestine. They find a dysplastic phenotype in the zygotic loss of function of wuho, the gene encoding dWdr4, which is recapitulated with the RNAi-mediated knock-down of *dwdr4* as well as mosaics of the mutation. Then they trace this phenotype as an excess of proliferation, concomitant with an increased nucleolar size, which they can correct manipulating the TOR and JNK pathways as well as the transcription factor dMyc.

They summarise their interpretation of the results in Fig7H, which proposes that, in the intestinal stem cell (ISC), there is a pathway like this:

dWdr4+Mettl1 -> 7mG:let7 --| TOR -> ribosome biogenesis and ROS...

...ROS -> JNK -> dMyc -> ribosome biogenesis and ISC proliferation

Whereas in neighbouring cells, the ROS produced by ISCs leads to:

ROS -> JNK -> dMyc

While this tenet is reasonable, and the authors provide good evidence for it, I find that it is at odds with some of the observations as well as some of the wording in the paper; I also think some fundamental support for its core proposal is missing. I would like the authors to comment on the following issues I think I have found:

In the first place, the 'pathway' delineated in Fig 7H is at odds with the title and the abstract, which suggest that somehow it is the lack of proper ribosome biogenesis what induces the loss of homeostasis. For instance, the title goes: "Wdr4 ... Maintain[s] Steady-State Ribosome Biogenesis for Intestinal Homeostasis". And in the abstract: "... TOR signaling and dMyc enhance ribosome biogenesis, facilitating ISC proliferation" and "... our findings unravel ... modulation in ... ribosome biogenesis for ISC homeostasis". However, I could not find any experiment that shows that the increased ribosome biogenesis induced by loss of dWdr4 leads to excessive proliferation and dysplasia (i.e. interfering with ribosome biogenesis experimentally, for instance with a Minute mutant). Another problem with making ribosomal biogenesis so central in the proposal of the paper is that the only observation that is made about it is the size of the domain of fluorescence of anti-Fibrillar in ISCs/EBs labelled with fluorescence (which has a problem of perdurance that can make the identification of cell populations dubious, see below). I would expect more than one technique to prove this point (e.g. ethynyl uridine incorporation), even if it is only in a subset of experiments.

The second element (peripheral though it is) is the production of excessive ROS. I only find one experiment where the production of ROS is addressed -- the increase in dihydroethidium dye fluorescence in Fig4A. However, while the micrographs indeed show higher levels of DHE fluorescence in the *dwdr4* mutants, this is not the appropriate readout of the dye. Dihydroethidium acts as a ROS sensor because its oxidation makes it intercalate into DNA, whereas when reduced it stays in the cytoplasm. All the fluorescence shown is cytoplasmic, so regardless of its intensity, it must be reduced and therefore it is not indicating an increase in superoxide radical. Apart from this, the authors do not show any epistatic relationship between the

production of ROS and the activation of JNK; the authors describe how in intestines harbouring mitotic recombination clones of *dwdr4*, heterozygous cells outside the clones as well as the mutant cells display high levels of phosphorylated JNK (Fig4C), and then say "the increased pJNK levels were not observed in *dwdr4* heterozygous guts without clones... suggesting that the increase of pJNK was due to the spread of ROS between cells and not to loss of a single copy of *dwdr4*." However, I do not see how the authors have established that the JNK activation in *dwdr4* cells or tissues is due to the presence ROS. Lastly, ROS are unlikely to be able to travel from one cell to the next as they are extremely short-lived molecules - so I cannot see how the authors can explain the non-cell autonomy of some (most) of the phenotypes.

A more substantial issue is that other epistasis experiments to prove the pathway are only partially convincing. For instance, all TOR epistasis is based on the use of rapamycin, when inhibition could be achieved orthogonally with RNAi. The same is true for dMyc - using zygotic heterozygosis only, instead of combining this with RNAi in at least some choice experiments that validate that approach for the rest of the paper. Moreover, the readouts are only partially convincing: the stainings of pJNK and dMyc in Figs6B,E Fig7F are very noisy with either a very high gain or surprisingly high contrast, with some images having clearly higher background levels than those we have to compare them with. And with dMyc in Fig6H, I really do not see a difference outside the small area indicated by the authors.

A technical point, that is however important for the interpretation, is that most of the determination of cell types is based on expression of Gal4 lines driving expression of GFP in a context of dysplasia and accelerated proliferation. In these situations, cells often differentiate before the GFP that was expressed until that point has had enough time to degrade, leading to seemingly misdifferentiation or excessive abundance of "progenitor" cells. This is important, because it can lead to misidentification of early ECs (say, 4n or 8n) as ISC/EBs, specifically in the mutant situation, and to logically find larger nuclei there. Some of these experiments, therefore, should be performed with antibody stainings for ISC/EB cells (e.g. Headcase).

Minor issues

- Throughout: the genetic nomenclature for *dWdr4* is confusing. Please use the official gene symbol and gene name, *wuho* (not *wh*, which is the official gene symbol for whisker) throughout the manuscript to refer to the locus. Use *wuho*-GFP or *wuho*:GFP to refer to the GFP-tagged rescue fragment.
- Results p6: "Since maintenance of a functional gut plays a key role in determining lifespan (Hodge et al., 2022)" This paper is a review focused on *C. elegans*; there are more directly relevant papers in the *Drosophila* literature, for instance from the work of Heinrich Jasper or Nicolas Buchon (among others).
- Results p6: "Discs large (Dlg) [...] was greatly reduced in the EC-EC junction of the *dwdr4* mutant female intestine, as compared to the control (Fig. 1B [...])." Methods section does not establish how the microscopy was performed to allow quantitative interpretation. Please clarify this from image acquisition through to quantification approach including normalisation. Provide a measure of the difference in protein amounts.
- Results p6: "actin filaments in the visceral muscle layer surrounding the *dwdr4* mutant female gut were generally disorganized and discontinuous". The only evidence here is a micrograph. Can the authors please provide a bit more data/quantification. I am not asking for something too sophisticated or going overboard - an approach similar to Fig2A with phenotype classifications would be enough.
- Results p7: "According to our RNA-sequencing analysis..." At this stage, some details are missing - one should not need to jump to Methods and back to this point to understand what comes after. It would be enough with simply stating the organs from where RNA was extracted, at which life stage, and the genotypes/sex/other (which by the way, is not explained in Methods.)
- Results p7: I think that section "*dWdr4* is expressed in the ISC lineage and controls gut morphogenesis" and its corresponding Figure panels (Fig 1DEF) needs improvement. First of all, what does it mean that *dWdr4* is expressed "in the ISC lineage"? It seems to be expressed in at least a fraction of all cell types in the intestinal epithelium. Lineage has nothing to do with this. Second, the authors use as cell-type markers Delta-Gal4 and the size of the nuclei. This is just insufficient to distinguish ISCs from pEEs nor EBs from EEs. At least Prospero should be added as a marker -- if the authors do not have access to a 4-laser confocal, they can use Prospero+Delta in the same channel. Or Prospero+Headcase. If the anti-*dWdr4* antibody is raised in mouse, they can validate the *dwdr4*:GFP rescue fragment expression with that antibody, and then use that with Delta-Gal4 UAS-lacZ or any other marker protein the authors can use. In any case, the anti-*dWdr4*/*dwdr4*:GFP staining should be shown at this stage, in the main figure or a supplementary. Finally, at least an estimate of which fraction of cells of each type express *dWdr4* (half, 90%, occasionally?) should be described.
- Results p8: In the section "The loss of *Wdr4* causes gut dysplasia and disrupts ISC lineage homeostasis", I think some of the interpretations the authors make of their observations may be incorrect. A typical phenotype that leads to 'dysplasia' is due to the acceleration of tissue turnover, which is clearly happening here (Fig2DE). This often leads to EBs quickly differentiating and displaying polyploid nuclei before the GFP expressed while they were EBs has had a chance to degrade. However, the authors are interpreting these cells as a sign of mis-differentiation, which is a different phenotype. Using a destabilised GFP would help clarify this, or the use of markers that are expressed in ISCs/EBs (e.g. Headcase) at the same time as EC-specific ones (*myoIA*, Crinkled). A minor, similar misinterpretation is the apparent observation that ISCs and 'progenitors' (not clearly defined) are found in clusters - well, that is because if you accelerate turnover, ISCs will proliferate, and EBs and pEEs will differentiate, and they will all carry GFP expressed by influence of the *esg*-Gal4 driver for a while, until it is degraded. The amount of cells is just indicative of how much quicker the turnover is than the degradation rate of the GFP - or how long has been since the accelerated turnover started. Again, using a UAS-destabilisedGFP would be helpful, if the authors really want to claim that misdifferentiation and unusual clustering is happening. Moreover, the title of the section is misleading, as there seems to be very little "disruption of ISC lineage homeostasis" (it is not clear to me what this means).

- Results p9-10: the authors claim that "In controls, asymmetric division of ISCs generated ISC-EB pairs, marked by *esg>mCD8gfp*, with only ISCs expressing DI [note - it is not clear whether this latter phrase is describing the literature or their own observation]. In contrast, cells expressing *esg>mCD8gfp* in the *dwdr4* mutant PMGs often formed clusters and were positive for DI, suggesting a defect in ISC-EB differentiation that give rise to misdifferentiated ECs. These results demonstrate the role of *Wdr4* in controlling the balance between ISC self-renewal and differentiation." However, it is simply not possible to access the consequences of ISC division without using lineage tracing approaches that the authors do not use in the manuscript (i.e. twin-spot approaches or single-colour clones coupled to mathematical modelling). The observation of multiple *esg-Gal4* UAS-mCD8GFP cells making clusters where several cells express Delta is not fundamentally different from what was shown before (*esg-Gal4*, UAS-GFP cells displaying polyploid nuclei). Therefore I see here no demonstration that the balance between self-renewal and differentiation is altered (though the overall turnover kinetics might be). A bit later (p12), the authors conclude the section "*dWdr4* cell-autonomously controls ISC division and differentiation..." by saying "Our results therefore suggest that loss of *dWdr4* in ISCs increases ROS levels and activates JNK signaling, which promotes ISC proliferation toward symmetric division to increase ISCs", there is again nothing in their experiments that allows to draw conclusions about the fate symmetry of ISC divisions. An accumulation of ISCs could be due simply to accelerated divisions without a parallel increase in differentiation. Moreover, the fact that there are more ISCs in *dwdr4* mutant cells or organs is not well substantiated, as it is only addressed directly in Fig3C, and using the *Gal4-UAS* system, which can create a lag of degradation that overestimates cell numbers when turnover is accelerated.

- Results p10: In section "Loss of *dWdr4* increases ISC proliferation and reduces the capacity for proper differentiation", the authors conclude that "*dWdr4* controls ISC division and differentiation capacity in a cell-autonomous manner." I agree with the first part (ISC division) but not the second (differentiation). Indeed the clones have a large amount of *esg+Pros-* cells (Fig3A'), but they are also larger (Fig3A'), which suggests that the amount of differentiated offspring has not changed significantly. To conclude that there is a lack of differentiation, the amount of differentiated offspring in the clones should be significantly less than in control clones. It is difficult to deduce from the figures as they are designed, but by eye it looks like the amount of average total cells per clone changes 15-30%, depending on the time point (Fig3A') -- the increase of *esg+Pros-* cells seems smaller (again by eye, in Fig3A').

- Results p11: The authors say "the *dwdr4*-knockdown ISC/EB clusters also showed ectopic Notch activation [marked by *Su(H)GBE-lacZ*], in comparison to the ISC-EB pair in the control intestine where only the EB was positive for *lacZ* (SFig. 5B). This ectopic activation of Notch may contribute to the observed defect in ISC differentiation toward ECs upon depletion of *dWdr4*." Expanded *Su(H)GBE* expression is typical of dysplasia and/or accelerated turnover - EBs turn into ECs that show beta-Galactosidase protein because this has not had time to degrade since it was synthesised when the cell was still an EB. It is fine for the authors to connect it to their phenotype, but the word "ectopic" is not appropriate - this is not a developmental context. Also, it makes little sense to say "in the control intestine ... only EBs are positive for [*Su(H)GBE*]-*lacZ*", as the EBs are defined in practice by their expression of *Su(H)GBE-lacZ* (or other markers, like *klumpfuss*), or by their capacity to differentiate only into ECs (which cannot be assessed here). Suggesting that those cells are activating Notch signalling anomalously is as parsimonious as saying that all those cells are EBs, only that *dwdr4* mutation causes EBs to become large and polyploid. Finally, if those were indeed cells perceiving Notch signalling anomalously, one would expect that to enforce their differentiation into ECs, and therefore to contribute less, not more, to the observed phenotype.

- Results p11: The authors say "Furthermore, *esg>dwdr4RNAi* flies died earlier than control flies (SFig. 5G), indicating that *dWdr4* function in ISCs is required for maintaining healthy lifespan." If this were true, a similar phenotype would be found with in *DI>dwdr4RNAi* animals. Was that the case? If not, that's fine - it is still an interesting observation, but it should be interpreted in terms of ISCs+EBs, not just ISCs.

- Methods p22: The authors say the flies used to generate mosaic guts with the *wuho[7]* mutation are "*hs-flp1 act-gfp FRT19A/FRT19A* and *hs-flp1 act-gfp FRT19A/wh7 FRT19A*". These genotypes cannot produce mutant clones positively marked with GFP expression. Can you please clarify?

- Figure 6: the order of panels is a bit difficult to follow - eyes tend to go from left to right and then top to bottom, so the natural order of reading that figure would be AB EFC DGH C'D'G'. So many insets and sub-panels without much different in the width of gaps between them, make it even harder to follow.

Language suggestions

- Please send the revised ms with line and page numbers (this is something that could have been picked up by the editor)

- Abstract: "mechanically" x mechanistically

- Intro p3: "Proper gut function depends on the constant growth and regeneration of gastrointestinal tissue" - I think I understand what the authors mean here and I agree with them, but the literality of the sentence means something else, which is not correct.

- Intro p4: "The [*Drosophila*] adult intestine houses well-characterized ISCs ..."

- Intro p4: "The midgut is the counterpart of the mammalian small intestine..." Also shares some functions with the stomach.

- Intro p4: "EBs differentiate into ECs which undergo extensive growth and DNA [endo]replication..."

- Results p6: "intestinal leakage was rescued by adding back one copy of the wild-type *dwdr4* allele (n=8) (Fig. 1C)." It should be Fig. 1A.

- Results p12: "Hemipteruous" should be Hemipterous.

Referee #2:

In the proposed manuscript, Kajal et al characterize a previously unknown function of the WD repeat protein 4 (*Wdr4*) in

Drosophila melanogaster intestinal stem cell (ISC) function and gut homeostasis. The authors show that dWdr4 controls basal levels of ISC proliferation and its absence or down-regulation lead to increased ROS levels, increased TOR and JNK-dMyc signaling and tissue dysplasia. Mechanistically, the authors propose that the above phenotypes are mediated by let-7 mirRNA levels via its m7G modification.

The study provides a careful analysis of intestinal dysplastic phenotypes in dWdr4 mutant flies as well as in tissue-specific knock-down and MARCM experiments. For the most part, these experiments are well conducted and the involvement of dWdr4 in the control of ISC proliferation is well supported by the experimental evidence provided.

Nevertheless, I have the following concerns regarding the current version of the manuscript:

1 - Concerning lifespan analysis (Fig S2):

- I did not find enough information about the genetic background of compared flies. Were the wh7 mutant lines backcrossed to the same wild-type strain that served as a control? Although, based on the demonstrated gut phenotypes, I do believe that loss of dWdr4 likely significantly affects lifespan, standardizing genetic backgrounds is a necessary good practice approach in order to reliably compare lifespans. Otherwise, authors should better opt for GeneSwitch>RNAi experiments, where flies with identical genotypes can be compared upon RU486 feeding.

- 10 females per replicate, with a total of 40 females (4 replicates) is a rather small sample size for lifespan analysis

2 - In MARCM analysis (Fig 3A, 4C, 5C), wild-type clones appear very small (one or two cells only on most images), which is problematic in assessing lineage phenotypes reliably. Tissues should be dissected at later time points or heat-shock conditions modified to ensure sufficient clone growth in control conditions.

3 - Some observations lack quantifications, which should be performed throughout multiple guts and added to the figures. The fly gut is, in general, a highly variable tissue in immunofluorescence assays and quantitative statements based on one microscopy image are difficult. Please provide quantifications for Fig 4A, B and C, with a sufficient wild-type clone size for the latter.

4 - There is no statement on data availability. RNA-seq data should be deposited in an appropriate database. Also, basic statistics of sequenced RNA-seq libraries should be provided. It is not clear if the analysis was performed in replicates.

While, apart from the few experimental shortcomings mentioned above, the dWdr4 involvement in the control of ISC proliferation is overall well supported by the data presented, unfortunately, I have stronger concerns regarding the mechanistic insights provided further in the manuscript.

5 - The conclusion that dWdr4 regulates ribosome biogenesis (Fig 5) appears not convincingly supported with the experimental evidence.

- The in silico analysis of genes "predictive of dwdr4 expression" (Fig 5A) by itself is not convincing. Are the identified gene categories (involved in protein translation, ribosome biogenesis and regulation) also affected in the dWdr4-deficient RNA-seq data? Also, RNA-seq data from which tissue type were used for this analysis?

- Moreover, conclusions on ribosome biogenesis drawn solely on the provided measure of nucleolar (fibrillar) size (Fig 5B and C, the latter not quantified and with small wt clones), appear rather weak. Primarily because authors reported that esg positive dWdr4 RNAi cells show increased nuclear size ("mis-differentiated" ECs), fibrillar size should rather be reported as a ratio of nucleolar to DNA (nucleus) volume (as previously done, e.g. <https://doi-org.insb.bib.cnrs.fr/10.1038/ncb1223>).

- Finally, some other measures of ribosome function should be used to further support the authors conclusions, e.g. rRNA and ribosomal genes transcript levels.

Thus, in the current version of the manuscript, the assumptions on increased ribosomal biogenesis upon dWdr4 depletion (present throughout the text) are not well supported with experimental evidence. As a consequence, conclusions such as "our study shows that dWdr4 is essential regulator of ribosome biogenesis" (Discussion and elsewhere) are overstated.

6 - Regarding the dWdr4-Mettl1-let7 m7G axis, the following concerns can be noted:

- On multiple occasions authors imply that dWdr4 and Mettl1 act in complex in the fly gut. In absence of any experimental evidence of the interaction between the two proteins in this system, such statements should be avoided. Also, please provide quantification on multiple guts for Fig 7A and S8A.

- Does dWdr4 KD lead to decreased Mettl1 levels (RNA or protein)? Can dwdr4 RNAi phenotypes be rescued by Mettl1 over-expression?

- Finally, let7 over-expression rescue experiments convincingly support the notion that dWdr4 phenotypes are, at least in part, mediated by diminished levels of let-7. However, since let7 m7G modification (and its loss upon dWdr4 knock-down), is not demonstrated, the involvement of this modification and Mettl1 protein in let7 stability remains only a hypothesis based on

previous studies.

7 - Finally, the experiments of human WDR4 overexpression rescue are barely discussed and are not convincing based only on Fig 7G. Please provide quantification for this figure panel (as in Fig 2A' and the mitotic index). The text states "OE of human WDR4 in ISCs/EBs also rescued the dWdr4-deficiency phenotypes" - please specify which phenotypes (all?) were rescued and show the data.

Additional minor points:

Page 9:

- Please spell out PHH3 when used for the first time
- Change to "ISCs are the major cell type known to undergo division" in light of the fact that EEp divides to create 2 EEs

For clarity, authors should report all results in the Results section, rather than reporting some to the Discussion (Fig S10).

Materials and Methods:

- Please provide the composition of the Drosophila medium used

Referee #3:

In this manuscript, Kajal and co-workers examine the role of Wdr4 in the proliferation of stem cells in the fruit fly intestine. The authors show evidence that Wdr4 is required for wild-type survival and intestinal homeostasis. Wdr4 is expressed in intestinal stem cells (ISC) and plays a role in their proliferation and gut morphogenesis. They show that Wdr4 impacts let-7 expression and subsequently TOR activation, ROS levels and Myc activity. They present evidence that Wdr4 regulates ribosome biogenesis in ISCs via Myc and TOR. Overall the study is nicely executed but I believe there are a few issues that would need to be addressed before I can recommend publication:

Major issues:

1) There are some key points that I am not persuaded are well supported by the data:

1a) The authors indicate that Wdr4 may impact let-7 stability through its m7G modification but the only data presented indicate transcriptional regulation of let-7 is impacted by loss of Wdr4.

1b) The authors propose that ribosome biogenesis is regulated by Wdr4 but there are no direct measures of ribosome biogenesis, only a convenient but indirect measure of nucleolar size.

1c) The authors propose that ISC proliferation is regulated by Wdr4 via ribosome biogenesis but this is not directly tested (i.e. is ribosome biogenesis necessary and/or sufficient for Wdr4 effects on ISC proliferation)

I think the authors will have to provide new data or substantially change their conclusions.

2) The fly lines used for lifespan assays do not appear to have been backcrossed. Backcrossing is essential for fly lifespan experiments. I suggest the authors either backcross the lines and re-do the experiments or exclude lifespan analysis from the manuscript.

3) RNA-Seq raw data will need to be deposited in a suitable repository before publication. False discovery rates will need to be stated for all analyses.

4) There are two experiments that are missing a genotype required for full interpretation of the results:

4a) Figure 5 - in addition to *dmyc4/+ Df>dwdr4* I think the authors need to include *dmyc4/+* alone. If this results in a reduction in pH3+ cells and nucleolar size in the absence of RNAi equivalent to what is observed in the presence of RNAi, the authors' conclusions may be different.

4b) Similarly, in Figure 7 *let-7OE* alone is missing.

5) Some important literature, from the area of ageing, ribosome biogenesis and ISCs, does not appear to have been cited: For example:

<https://doi.org/10.1038/ncomms16083>

<https://doi.org/10.1016/j.celrep.2020.01.017>

Minor:

6) In the abstract "which suppress TOR signaling to preserve steady-state ribosome biogenesis in ISCs" is awkward as TOR is expected to promote ribosome biogenesis.

7) For most of the introduction, it is unclear what is referring specifically to the fly gut rather than mammalian (or other) gut.

** As a service to authors, EMBO Press provides authors with the ability to transfer a manuscript that one journal cannot offer to publish to another journal, without the author having to upload the manuscript data again. To transfer your manuscript to another EMBO Press journal using this service, please click on
Link Not Available

We sincerely thank the reviewers and editor for their time, thoughtful feedback, and constructive suggestions, which have greatly improved the clarity and quality of our manuscript. All changes made in response to the reviewers' comments are marked in red for easy reference.

Referee #1:

I would like to start by apologizing to the authors for my lateness in sending the review. In their manuscript, Kajal et al. tackle the role of ribosome biogenesis in regulating stem cell activity by studying *Drosophila* Wdr4 effects in the adult intestine. They find a dysplastic phenotype in the zygotic loss of function of wuho, the gene encoding dWdr4, which is recapitulated with the RNAi-mediated knock-down of dwdr4 as well as mosaics of the mutation. Then they trace this phenotype as an excess of proliferation, concomitant with an increased nucleolar size, which they can correct manipulating the TOR and JNK pathways as well as the transcription factor dMyc.

They summarise their interpretation of the results in Fig7H, which proposes that, in the intestinal stem cell (ISC), there is a pathway like this:

dWdr4+Mettl1 -> 7m7G:let7 --| TOR -> ribosome biogenesis and ROS...

...ROS -> JNK -> dMyc -> ribosome biogenesis and ISC proliferation

Whereas in neighbouring cells, the ROS produced by ISCs leads to:

ROS -> JNK -> dMyc

While this **tenet is reasonable**, and the authors provide good evidence for it, I find that it is at odds with some of the observations as well as some of the wording in the paper; I also think some fundamental support for its core proposal is missing. I would like the authors to comment on the following issues I think I have found:

In the first place, the 'pathway' delineated in Fig 7H is at odds with the title and the abstract, which suggest that somehow **it is the lack of proper ribosome biogenesis what induces the loss of homeostasis**. For instance, the title goes: Wdr4 ... Maintain[s] Steady-State Ribosome Biogenesis for Intestinal Homeostasis". And in the abstract: "... TOR signaling and dMyc enhance ribosome biogenesis, facilitating ISC proliferation" and "... our findings unravel ... modulation in ... ribosome biogenesis for ISC homeostasis". However, I could not find any experiment that shows that the increased ribosome biogenesis induced by loss of dWdr4 leads to excessive proliferation and dysplasia (i.e. interfering with ribosome biogenesis experimentally, **for instance with a Minute mutant**). Another problem with making ribosomal biogenesis so central in the proposal of the paper is that the only observation that is made about it is the size of the domain of fluorescence of anti-Fibrillarin in ISCs/EBs labelled with fluorescence (which has a problem of perdurance that can make the identification of cell populations dubious, see below). I would expect **more than one technique to prove this point** (e.g. ethynyl uridine incorporation), even if it is only in a subset of experiments.

<Response to the reviewer>

We thank the reviewer for this critical comment. As suggested, we performed 5-ethynyl uridine (EU) incorporation in *dwdr4* mutant guts to detect nascent RNA synthesis, of which ribosomal RNA (rRNA) is the predominant form. Consistent with our Fibrillarin-marked nucleoli results, the intensity of EU signals was markedly increased in ISCs (marked by Delta) and in ECs (Delta-negative cells with large nuclei) of *dwdr4* mutant posterior midguts (PMGs) compared with controls. Moreover, the ratio of 5'-EU (nucleolar) to DAPI (nuclear) area in ISCs was significantly elevated. These findings indicate that *dWdr4* depletion enhances ribosome biogenesis. We have incorporated these results in Fig. 4C and C' and added the corresponding description in the main text (pg. 11, lines 256–259).

To determine whether increased ribosome biogenesis accounts for the gut phenotypes caused by *dWdr4* loss, we treated flies with BMH-21, a small molecule that suppresses RNA polymerase I-mediated rDNA transcription and is widely used to inhibit ribosome biogenesis in cultured cells, flies, and mice (Cho et al., *Cellular and Molecular Gastroenterology and Hepatology*, 2025; Gui et al., *Cell Report*, 2021). Four-day-old wild-type and *wh7* mutant flies were treated with 5 μ M or 25 μ M BMH-21 for 3 days. We found that 5 μ M BMH-21 was sufficient to reduce the Fibrillarin/DAPI area ratio in ISCs/EBs of both wild-type and *wh7* mutant PMGs compared with untreated controls. While PHH3⁺ ISC numbers were not reduced in wild-type guts, they were significantly decreased in *dwdr4* mutant guts, along with suppression of the ISC/EB expansion phenotype. These results identify *dWdr4* as a key regulator of ribosome biogenesis in ISCs, with its loss triggering elevated ribosome biogenesis, ISC overproliferation, and ISC/EB expansion. We have incorporated these results into Fig. 4D–D'' and described in the main text (p. 11, lines 260–270).

As suggested by the reviewer, we removed one copy of the Minute gene (*RpS17* or *RpL15[8-1]*) in the *wh7* mutant background. However, these flies were very unhealthy; therefore, we chose to use BMH-21 treatment rather than Minute gene reduction.

The second element (peripheral though it is) is the production of excessive ROS. I only find one experiment where the production of ROS is addressed -- the increase in dihydroethidium dye fluorescence in Fig. 4A. However, while the micrographs indeed show higher levels of DHE fluorescence in the *dwdr4* mutants, this is not the appropriate readout of the dye. Dihydroethidium acts as a ROS sensor because its oxidation makes it intercalate into DNA, whereas when reduced, it stays in the cytoplasm. All the fluorescence shown is cytoplasmic, so regardless of its intensity, it must be reduced, and therefore, it does not indicate an increase in superoxide radical

<Response to the reviewer>

We appreciate the reviewer's concern. DHE is oxidized by superoxide and other ROS to form 2-hydroxyethidium and ethidium, both of which emit red fluorescence (Nazarewicz et al., *Journal of Biomolecular Screening*, 2012). These oxidized products can intercalate into DNA upon nuclear entry, resulting in enhanced nuclear fluorescence (Ortega-Villasante et al., *Free Radical Biology and Medicine*, 2018). However, it is important to note that oxidized DHE derivatives, particularly 2-

hydroxyethidium, can accumulate and fluoresce in the cytoplasm before translocating to the nucleus, as reported in previous studies [Grefte and Koopman, *Methods Mol Biol.*, 2021 (see Fig. 2); Dikalov et al., *Antioxidants & Redox Signaling*, 2014 (see Fig. 1)].

In our previous study (Huang et al., *Development*, 2024), we treated dissected fly guts with paraquat (PQ), a known superoxide generator (Castello et al., *Journal of Biological Chemistry*, 2007), and used DHE staining under the same conditions described in the current manuscript. We observed DHE-derived fluorescence in both the cytoplasm and nucleus, supporting that oxidized DHE products can localize to both compartments.

In the present study, fly guts were incubated with 30 μ M DHE for 7 minutes, followed by fixation. Under these conditions, we observed DHE signals in both the cytoplasm and nucleus in some gut sections, and predominantly cytoplasmic signals in others. Similar cytoplasmic DHE signals in *Drosophila* intestinal cells have been reported (Zhai et al., *Nature Communications*, 2015). In contrast, other studies using different staining conditions (e.g., 50 μ M DHE for 20 minutes) observed stronger nuclear fluorescence (Deng et al., *Cell Death Discovery*, 2018).

These findings suggest that cytoplasmic localization of oxidized DHE can indeed reflect increased cytoplasmic ROS, particularly when fixation is performed shortly after staining. Moreover, as the reviewer suggested (see below), we tested antioxidant intervention by overexpressing Catalase in wh7 mutant guts, which significantly suppressed both ISC overproliferation and ISC/EB expansion phenotypes (Fig EV4E and E' in the revised manuscript). Thus, the observed cytoplasmic fluorescence is consistent with ROS elevation and supports our conclusion that dWdr4 loss increases oxidative stress in intestinal cells.

However, I do not see how the authors have established that the JNK activation in dwdr4 cells or tissues is due to the presence of ROS.

<Response to the reviewer>

We thank the reviewer for raising this important point. While JNK activation by ROS is well established in the *Drosophila* gut, we acknowledge that our previous version did not provide direct evidence linking ROS to JNK activation in the context of dWdr4 loss. To address this, we overexpressed Catalase—a key ROS scavenger that converts H₂O₂ into water and oxygen—in ISCs/EBs of dwdr4 mutant guts. Catalase overexpression significantly suppressed elevated JNK signaling, as well as ISC expansion and overproliferation. These results demonstrate that ROS functionally contributes to the observed phenotypes and strongly suggest that ROS acts upstream of JNK activation. The new data are now included in Fig EV4D-E', with the corresponding description added to the main text (p. 13, lines 309–311).

Lastly, ROS are unlikely to be able to travel from one cell to the next as they are extremely short-lived molecules - so I cannot see how the authors can explain the non-cell autonomy of some (most) of the phenotypes.

<Response to the reviewer>

We thank the reviewer for raising this important point. We agree that ROS are highly

reactive and unlikely to directly diffuse across multiple cells to account for the non-cell-autonomous phenotypes. However, although ROS are short-lived, prior studies show that JNK signaling can amplify stress responses and induce secondary ROS production in neighboring cells, thereby explaining the non-cell-autonomous phenotypes we observe (Zhang et al., Scientific Reports, 2024; Santabárbara-Ruiz et al., PLOS Genetics, 2015). Furthermore, *dWdr4* depletion in ECs promotes EC endoreplication, which may also induce ROS-JNK-dMyc signaling cascades, as ROS, pJNK, and dMyc expression are elevated in the entire PMGs upon *dWdr4* loss.

We now include a discussion of how *dWdr4* loss may promote non-cell-autonomous phenotypes through such mechanisms (pg 17, line 410-422).

A more substantial issue is that other epistasis experiments to prove the pathway are only partially convincing. For instance, all TOR epistasis is based on the use of rapamycin, when inhibition could be achieved orthogonally with RNAi. The same is true for dMyc - using zygotic heterozygosity only, instead of combining this with RNAi in at least some choice experiments that validate that approach for the rest of the paper.

<Response to the reviewer>

We thank the reviewer for this constructive suggestion. To strengthen our epistasis analysis, we have now included new experiments in which TOR was knocked down via RNAi in ISCs/EBs of *dwdr4*-depleted midguts. Genetic inhibition of TOR substantially suppressed the ISC overproliferation and ISC/EB expansion phenotypes induced by *dWdr4* loss, while TOR knockdown alone did not affect ISC behavior. These findings are consistent with our pharmacological inhibition results using rapamycin and are now presented in Fig. 5B and B' and described in the main text (pg 12, line 277-278).

With respect to dMyc, RNAi-mediated knockdown produced strong autonomous defects, including loss of ISC maintenance and a pronounced reduction in nucleolar size (labeled by Fibrillarin), which precluded clear interpretation of interactions with *dwdr4*. To circumvent this, we reduced dMyc dosage using a heterozygous *dmyc⁴* mutant background. While *dmyc⁴/+* alone did not alter ISC mitosis, combining *dmyc⁴/+* with *dwdr4* depletion suppressed the increased nucleolar size, ISC overproliferation, and ISC/EB expansion phenotypes. These new data are now included in Fig. 5F and F' and Fig. EV3C and C', with a description in pg. 12, lines 288–291.

Moreover, the readouts are only partially convincing: the stainings of pJNK and dMyc in Figs 6 B, E, and Fig. 7F are very noisy with either a very high gain or surprisingly high contrast, with some images having clearly higher background levels than those we have to compare them with. And with dMyc in Fig6H, I really do not see a difference outside the small area indicated by the authors.

<Response to the reviewer>

We thank the reviewer for the helpful observation. In response, we have replaced the images in Figs. 6B, 6E (now are Fig 6C, 6D) and 7F (now is Fig 7D) with higher-quality versions that were acquired under consistent imaging conditions and optimized to reduce background noise and contrast artifacts.

Regarding Fig. 6H (now Fig. 6D and D'), Hep is depleted specifically in ISCs/EBs of the

wh7 mutant guts; therefore, dMyc expression is retained in non-ISC/EB populations, which explains the apparent difference being confined to the indicated region.

A technical point, that is however important for the interpretation, is that most of the determination of cell types is based on expression of Gal4 lines driving expression of GFP in a context of dysplasia and accelerated proliferation. **In these situations, cells often differentiate before the GFP that was expressed until that point has had enough time to degrade, leading to seemingly misdifferentiation or excessive abundance of "progenitor" cells.** This is important because it can lead to misidentification of early ECs (say, 4n or 8n) as ISCs/EBs, specifically in the mutant situation, and to logically find larger nucleoli there. Some of these experiments, therefore, should be performed with antibody stainings for ISC/EB cells (e.g. Headcase).

<Response to the reviewer>

We thank the reviewer for raising this important technical concern. To distinguish whether the apparent ISC–EC misdifferentiation observed in *wh7* mutants could be an artifact of GFP perdurance (leading to misidentification of early ECs as ISCs/EBs), we performed additional marker analyses. Staining with Delta (DI), a well-established ISC marker, revealed DI expression not only in ISCs with small nuclei but also in cells with enlarged EC-like nuclei.

As suggested by the reviewer, we also stained with anti-Headcase (Hdc). Consistent with the DI results, Hdc was detected not only in small ISCs/EBs but also in cells with large EC-like nuclei in *wh7* mutant intestine. These findings indicate that such large cells retain ISC/progenitor identity, supporting true misdifferentiation rather than misidentification due to GFP perdurance. The new DI and Hdc staining results are now included in Fig. 2D and Fig. EV2A, with a corresponding description in the main text (pg. 7, lines 171–174).

Minor issues

- Throughout, the genetic nomenclature for *dWdr4* is confusing. Please use the official gene symbol and gene name, *wuho* (not *wh*, which is the official gene symbol for whisker) throughout the manuscript to refer to the locus. Use *wuho*-GFP or *wuho*: GFP to refer to the GFP-tagged rescue fragment.

<Response to the reviewer>

We thank the reviewer for this comment. The *wuho* gene was originally identified and the allele *wh7* named in Wu et al. (Developmental Biology, 2006), and we retain *wh7* exclusively to refer to this established mutant allele.

To ensure clarity and consistency, we now uniformly refer to the gene as *dwdr4* throughout the manuscript, and to the GFP-tagged rescue construct as *dwdr4*-GFP. This nomenclature avoids confusion with unrelated loci and aligns with our use of *hWDR4* for the human ortholog. The manuscript has been revised accordingly.

- Results p6: "Since maintenance of a functional gut plays a key role in determining lifespan (Hodge et al., 2022)" This paper is a review focused on *C. elegans*; there are more directly relevant papers in the *Drosophila* literature, for instance from the work

of Heinrich Jasper or Nicolas Buchon (among others).

<Response to the reviewer>

We thank the reviewer for the suggestion. In addition to the *C. elegans*-based study previously cited, we have now included a directly relevant *Drosophila* study (Biteau et al., PLOS Genetics, 2010) demonstrating that gut homeostasis contributes to lifespan regulation. This reference has been added to the revised manuscript (pg. 5, line 108-109).

- Results p6: "Discs large (Dlg) [...] was greatly reduced in the EC-EC junction of the *dwdr4* mutant female intestine, as compared to the control (Fig. 1B [...])." The methods section does not establish how the microscopy was performed to allow quantitative interpretation. Please clarify this from image acquisition through to the quantification approach, including normalisation. Provide a measure of the difference in protein amounts.

<Response to the reviewer>

We thank the reviewer for this concern. In the revised manuscript, we co-stained control and *wh7* mutant posterior midguts with antibodies against Dlg (epithelial junction marker) and LamC (nuclear envelope marker). For quantification, Dlg fluorescence intensity was measured at EC-EC junctions within the largest ECs and normalized to LamC fluorescence intensity from the corresponding nuclei to account for sample-to-sample variation in staining and imaging. This analysis revealed a significant reduction of Dlg expression at EC-EC junctions in *dwdr4* mutants compared to controls. The quantification is now included in Fig 1B and B', the results are described in the main text (pg. 5, lines 115-117), and methodological details have been added to the Materials and Methods section (pg. 30, lines 736-744, and pg. 31, lines 759-762).

- Results p6: "actin filaments in the visceral muscle layer surrounding the *dwdr4* mutant female gut were generally disorganized and discontinuous". The only evidence here is a micrograph. Can the authors please provide a bit more data/quantification? I am not asking for something too sophisticated or going overboard - an approach similar to Fig. 2A with phenotype classifications would be enough.

<Response to the reviewer>

We thank the reviewer for this helpful suggestion. We classified visceral muscle F-actin as "normal" (long, continuous filaments with minimal lateral projections) or "branching" (discontinuous filaments with multiple lateral projections). Using this approach, 34% of sections from *dwdr4* mutant PMGs (n=29) showed the branching phenotype, compared to only 0.6% in controls (n=29; P<0.001). This quantification has been added to the text (pg. 5, lines 117-122), and scoring criteria are described in the Materials and Methods (pg. 31, lines 763-766).

- Results p7: "According to our RNA-sequencing analysis..." At this stage, some details are missing - one should not need to jump to Methods and back to this point to understand what comes after. It would be enough with simply stating the organs from where RNA was extracted, at which life stage, and the genotypes/sex/other (which by the way, is not explained in Methods.)

<Response to the reviewer>

We thank the reviewer for this important point. We now state in the Results that RNA-seq was performed on dissected intestines from 1-week-old control (*yw*) and *dwdr4* mutant (*wh7*) female flies (pg. 6, line 128-131), and we have also clarified these details in the Materials and Methods (pg. 27, line 657).

- Results p7: I think that section "dWdr4 is expressed in the ISC lineage and controls gut morphogenesis" and its corresponding Figure panels (Fig 1DEF) needs improvement. First of all, what does it mean that dWdr4 is expressed "in the ISC lineage"? It seems to be expressed in at least a fraction of all cell types in the intestinal epithelium. Lineage has nothing to do with this. Second, the authors use as cell-type markers Delta-Gal4 and the size of the nuclei. This is just insufficient to distinguish ISCs from pEEs nor EBs from EEs. At least Prospero should be added as a marker -- if the authors do not have access to a 4-laser confocal, they can use Prospero+Delta in the same channel. Or Prospero+Headcase. If the anti-dWdr4 antibody is raised in mouse, they can validate the *dwdr4*:GFP rescue fragment expression with that antibody, and then use that with Delta-Gal4 UAS-lacZ or any other marker protein the authors can use. In any case, the anti-dWdr4/*dwdr4*:GFP staining should be shown at this stage, in the main figure or a supplementary. Finally, at least an estimate of which fraction of cells of each type express dWdr4 (half, 90%, occasionally?) should be described.

<Response to the reviewer>

We thank the reviewer for this comment. To more accurately reflect our observations, we have revised the subtitle of this section to "dWdr4 is expressed in all intestinal cells and controls gut morphogenesis."

The genomic *dwdr4*-GFP transgene, which contains the *dwdr4* endogenous promoter and coding region fused to GFP, was previously used to examine dWdr4 expression in the testis and ovary (Wu et al., *Developmental Biology*, 2006). In this revised manuscript, we analyzed intestines carrying this transgene and co-stained with GFP (green, dWdr4), Hdc (red, ISC/EB/pEE marker), Pros (red, pEE/EE marker), and DAPI (nuclei). Based on these markers, ISCs/EBs were defined as Hdc⁺/Pros⁻ with small nuclei, pEEs as Hdc⁺/Pros⁺ with small nuclei, EEs as Hdc⁻/Pros⁺ with small nuclei, and ECs as Hdc⁻/Pros⁻ with large nuclei. We found that dWdr4 is expressed in all intestinal cell types, as shown in Fig. 1E and described in the text (pg. 6, lines 135–140).

Regarding antibody validation, the anti-dWdr4 antibody used previously is no longer available, preventing direct comparison of endogenous staining to *dwdr4*-GFP. Nevertheless, the GFP-tagged genomic construct fully rescues the *wh7* mutant phenotype, supporting that it faithfully reports dWdr4 expression.

- Results p8: In the section "The loss of Wdr4 causes gut dysplasia and disrupts ISC lineage homeostasis", I think some of the interpretations the authors make of their observations may be incorrect. A typical phenotype that leads to 'dysplasia' is due to the acceleration of tissue turnover, which is clearly happening here (Fig2DE). This often leads to EBs quickly differentiating and displaying polyploid nuclei before the GFP expressed while they were EBs has had a chance to degrade.

However, the authors are interpreting these cells as a sign of mis-differentiation,

which is a different phenotype. Using a destabilised GFP would help clarify this, or the use of markers that are expressed in ISCs/EBs (e.g. Headcase) at the same time as EC-specific ones (myoIA, Crinkled). A minor, similar misinterpretation is the apparent observation that ISCs and 'progenitors' (not clearly defined) are found in clusters - well, that is because if you accelerate turnover, ISCs will proliferate, and EBs and pEEs will differentiate, and they will all carry GFP expressed by influence of the *esg-Gal4* driver for a while, until it is degraded. The number of cells is just indicative of how much quicker the turnover is than the degradation rate of the GFP - or how long has been since the accelerated turnover started. Again, using a UAS-destabilised GFP would be helpful if the authors want to claim that misdifferentiation and unusual clustering is happening. Moreover, the title of the section is misleading, as there seems to be very little "disruption of ISC lineage homeostasis" (it is not clear to me what this means).

<Response to the reviewer>

We thank the reviewer for this critical comment, which we have also addressed in response to the major point raised.

To distinguish whether the apparent ISC–EC misdifferentiation observed in *wh7* mutants could be an artifact of GFP perdurance (leading to misidentification of early ECs as ISCs/EBs), we performed additional marker analyses. Staining with Delta (DI), a well-established ISC marker, revealed DI expression not only in ISCs with small nuclei but also in cells with enlarged EC-like nuclei.

We also stained with anti-Headcase (Hdc). Consistent with the DI results, Hdc was detected not only in small ISCs/EBs but also in cells with large EC-like nuclei in *wh7* mutant intestine. These findings indicate that such large cells retain ISC/progenitor identity, supporting true misdifferentiation rather than misidentification due to GFP perdurance. The new DI and Hdc staining results are now included in Fig. 2D and Fig. EV2A, with a corresponding description in the main text (pg. 7, lines 171–174).

In addition, we have revised the section title to “*dWdr4* regulates ISC proliferation and differentiation to preserve intestinal tissue integrity” to more accurately reflect our findings.

- Results p9-10: the authors claim that "In controls, asymmetric division of ISCs generated ISC-EB pairs, marked by *esg>mCD8gfp*, with only ISCs expressing DI [note - it is not clear whether this latter phrase is describing the literature or their observation].

<Response to the reviewer>

We thank the reviewer for this comment. The statement refers to our own observation, where in ISC–EB pairs marked by *esg>mCD8GFP*, only the ISC expressed DI (Fig. 2D, previous version). We have replaced these data with higher-quality DI staining in both wild-type and *dwdr4* mutant PMGs (Fig. 2D, revised manuscript) and revised the text to make it explicit that this conclusion is based on our own observation.

In contrast, cells expressing *esg>mCD8gfp* in the *dwdr4* mutant PMGs often formed clusters and were positive for DI, suggesting a defect in ISC-EB differentiation that give

rise to misdifferentiated ECs. These results demonstrate the role of Wdr4 in controlling the balance between ISC self-renewal and differentiation."

However, it is simply not possible to access the consequences of ISC division without using lineage tracing approaches that the authors do not use in the manuscript (i.e. twin-spot approaches or single-colour clones coupled to mathematical modelling).

The observation of multiple *esg-Gal4 UAS-mCD8GFP* cells making clusters where several cells express Delta is not fundamentally different from what was shown before (*esg-Gal4, UAS-GFP* cells displaying polyploid nuclei). Therefore, I see here no demonstration that the balance between self-renewal and differentiation is altered (though the overall turnover kinetics might be).

A bit later (p12), the authors conclude the section "dWdr4 cell-autonomously controls ISC division and differentiation..." by saying "Our results therefore suggest that loss of dWdr4 in ISCs increases ROS levels and activates JNK signaling, which promotes ISC proliferation toward symmetric division to increase ISCs", there is again nothing in their experiments that allows to draw conclusions about the fate symmetry of ISC divisions.

An accumulation of ISCs could be due simply to accelerated divisions without a parallel increase in differentiation. Moreover, the fact that there are more ISCs in *dwdr4* mutant cells or organs is not well substantiated, as it is only addressed directly in Fig3C, and using the Gal4-UAS system, which can create a lag of degradation that overestimates cell numbers when turnover is accelerated.

<Response to the reviewer>

We thank the reviewer for the comment. Under a broad definition of cell fate, a cell is considered to have adopted an incorrect fate if it begins to express proteins characteristic of another lineage, regardless of whether this results from an inappropriate transcriptional program or from delayed degradation of lineage-specific markers.

In the revised manuscript, we show that clustered cells with small nuclei, as well as some EC-sized cells in *wh7* mutant PMGs, express DI (an ISC marker) and Hdc (ISC/EB markers) (Fig. 2D and Fig. EV2A), indicating misexpression of lineage-specific markers in ECs and reflecting their mis-differentiation.

In addition, our MARCM clonal analysis revealed that *wh7* mutant clones were larger than control clones (Fig. 3A') and contained more ISCs/EBs (Fig. 3A''), indicating an increase in ISC proliferation.

Furthermore, ISC-specific knockdown of dWdr4 using DI-GAL4 phenocopied the *wh7* mutant PMG phenotypes (Fig. 3C, C'), including ISC overproliferation and expansion and EC misdifferentiation.

Altogether, these findings suggest that Wdr4 cell-autonomously regulates the balance between ISC self-renewal and differentiation. However, to avoid any potential confusion, we have softened our description in the revised manuscript to: "Wdr4 cell-autonomously controls ISC proliferation and proper differentiation ... (pg. 8, line 188)"

and “Our results therefore suggest that dWdr4 loss elevates ROS levels and activates JNK signaling, which in turn drives ISC hyperproliferation and expansion..... (pg. 13, line 316-318)”.

- Results p10: In section "Loss of dWdr4 increases ISC proliferation and reduces the capacity for proper differentiation", the authors conclude that "dWdr4 controls ISC division and differentiation capacity in a cell-autonomous manner." I agree with the first part (ISC division) but not the second (differentiation). Indeed, the clones have a large amount of esg+Pros- cells (Fig. 3A'), but they are also larger (Fig. 3A'), which suggests that the amount of differentiated offspring has not changed significantly. To conclude that there is a lack of differentiation, the number of differentiated offspring in the clones should be significantly less than in control clones. It is difficult to deduce from the figures as they are designed, but by eye it looks like the amount of average total cells per clone changes 15-30%, depending on the time point (Fig3A') -- the increase of esg+Pros- cells seems smaller (again by eye, in Fig3A').

<Response to the reviewer>

We thank the reviewer for the critical comment. Indeed, we did not observe a reduction of ECs, rather it is slightly increased, in wh7 mutant intestinal clones. To be accurate, we modified our conclusions regarding clonal analysis to “Wdr4 controls ISC proliferation in a cell-autonomous manner (pg9, line 209)”

- Results p11: The authors say "the dwdr4-knockdown ISC/EB clusters also showed ectopic Notch activation [marked by Su(H)GBE-lacZ], in comparison to the ISC-EB pair in the control intestine where only the EB was positive for lacZ (SFig. 5B). This ectopic activation of Notch may contribute to the observed defect in ISC differentiation toward ECs upon depletion of dWdr4." Expanded Su(H)GBE expression is typical of dysplasia and/or accelerated turnover - EBs turn into ECs that show beta-Galactosidase protein because this has not had time to degrade since it was synthesised when the cell was still an EB. It is fine for the authors to connect it to their phenotype, but the word "ectopic" is not appropriate - this is not a developmental context.

<Response to the reviewer>

We thank the reviewer for the critical comment. We have modified the sentence accordingly (pg. 9, line 217-218).

Also, it makes little sense to say "in the control intestine ... only EBs are positive for [Su(H)GBE]-lacZ", as the EBs are defined in practice by their expression of Su(H)GBE-lacZ (or other markers, like klumpfuss), or by their capacity to differentiate only into ECs (which cannot be assessed here).

Suggesting that those cells are activating Notch signalling anomalously is as parsimonious as saying that all those cells are EBs, only that the dwdr4 mutation causes EBs to become large and polyploid. Finally, if those were indeed cells perceiving Notch signalling anomalously, one would expect that to enforce their differentiation into ECs, and therefore to contribute less, not more, to the observed phenotype.

<Response to the reviewer>

We thank the reviewer for the comment. We have modified the sentence accordingly (pg. 9, line 217-218).

- Results p11: The authors say "Furthermore, *esg>dwdr4RNAi* flies died earlier than control flies (SFig. 5G), indicating that *dWdr4* function in ISCs is required for maintaining healthy lifespan." If this were true, a similar phenotype would be found with in *DI>dwdr4RNAi* animals. Was that the case? If not, that's fine - it is still an interesting observation, but it should be interpreted in terms of ISCs+EBs, not just ISCs.

<Response to the reviewer>

We thank the reviewer for recognizing that this is an interesting observation. We did not perform the lifespan assay with *DI>dwdr4RNAi*; we modified the original description to "indicating that *dWdr4* function in ISCs/EBs is essential for intestinal homeostasis and organismal health." (pg. 10, line 227-229).

- Methods p22: The authors say the flies used to generate mosaic guts with the *wuho*[7] mutation are " *act-gfp; FRT19A hsflp tub Gal80/FRT19A* and *act-gfp; FRT19A hsflp tub Gal80/wh7 FRT19A*". These genotypes cannot produce mutant clones positively marked with GFP expression. Can you please clarify?

<Response to the reviewer>

We thank the reviewer for pointing out this mistake. We have corrected the genotypes used for clonal analysis in the current version.

- Figure 6: the order of panels is a bit difficult to follow - eyes tend to go from left to right and then top to bottom, so the natural order of reading that figure would be ABEFCDGHC'D'G'. So many insets and sub-panels without much different in the width of gaps between them, make it even harder to follow.

<Respond to the reviewer>

We thank the reviewer, and we have remade the figure to make it easy to follow.

- Please send the revised ms with line and page numbers (this is something that could have been picked up by the editor)

<Response to the reviewer>

We thank the reviewer for the suggestion. We have added the line and page numbers in the current manuscript.

- Abstract: "mechanically" x mechanistically

<Response to the reviewer>

We thank the reviewer for pointing out this mistake. We have corrected it.

- Intro p3: "Proper gut function depends on the constant growth and regeneration of gastrointestinal tissue" - I think I understand what the authors mean here and I agree with them, but the literality of the sentence means something else, which is not correct.

<Response to the reviewer>

We thank the reviewer for pointing out this inappropriate sentence. We have modified the original sentence to "Gut homeostasis is essential for the health and longevity of all metazoans. This homeostasis relies on both the maintenance and regeneration of gastrointestinal tissues, processes driven by the self-renewal and differentiation of intestinal stem cells (ISCs). pg. 3, line 51-53).

- Intro p4: "The [Drosophila] adult intestine houses well-characterized ISCs ..."

<Response to the reviewer>

We thank the reviewer for pointing out this missing information. We have added "Drosophila" as a suggestion (pg. 3, line 58).

- Intro p4: "The midgut is the counterpart of the mammalian small intestine..." Also shares some functions with the stomach.

<Response to the reviewer>

We thank the reviewer for pointing this out. We have added this information to the text as "the midgut functions like the small intestine and partly like the stomach, performing digestion and absorption" (pg. 3, line 61-62)

- Intro p4: "EBs differentiate into ECs which undergo extensive growth and DNA [endo]replication..."

<Response to the reviewer>

We thank the reviewer for pointing out this mistake. We have modified our description.

- Results p6: "intestinal leakage was rescued by adding back one copy of the wild-type *dWdr4* allele (n=8) (Fig. 1C)." It should be Fig. 1A.

<Response to the reviewer>

We thank the reviewer for pointing out this mistake. We have corrected it.

- Results p12: "Hemipteruous" should be Hemipterous.

<Response to the reviewer>

We thank the reviewer for pointing out this mistake. We have corrected it.

Referee #2:

In the proposed manuscript, Kajal et al characterize a previously unknown function of the WD repeat protein 4 (*Wdr4*) in *Drosophila melanogaster* intestinal stem cell (ISC) function and gut homeostasis. The authors show that *dWdr4* controls basal levels of ISC proliferation and its absence or down-regulation lead to increased ROS levels, increased TOR and JNK-dMyc signaling and tissue dysplasia. Mechanistically, the authors propose that the above phenotypes are mediated by *let-7* mirRNA levels via its m7G modification.

The study provides a careful analysis of intestinal dysplastic phenotypes in *dWdr4* mutant flies as well as in tissue-specific knock-down and MARCM experiments. For the most part, these experiments are well conducted and the involvement of *dWdr4* in the control of ISC proliferation is well supported by the experimental evidence provided.

<Response to the reviewer>

We thank the reviewer for their positive evaluation of our study.

Nerveless, I have the following concerns regarding the current version of the manuscript:

1 - Concerning lifespan analysis (Fig S2):

- I did not find enough information about the genetic background of compared flies. Were the wh7 mutant lines backcrossed to the same wild-type strain that served as a control? Although, based on the demonstrated gut phenotypes, I do believe that loss of *dWdr4* likely significantly affects lifespan, standardizing genetic backgrounds is a necessary good practice approach in order to reliably compare lifespans. Otherwise, authors should better opt for GeneSwitch>RNAi experiments, where flies with identical genotypes can be compared upon RU486 feeding.

<Response to the reviewer>

We thank the reviewer for this important comment. In the current manuscript, the lifespan assays were performed with wh7 flies that had been backcrossed to a yw strain for 10 generations (Fig. EV1A, B). Our results showed that *dWdr4* regulates lifespan in both males and females, consistent with data obtained from non-backcrossed *dWdr4* mutants. We have now included this genetic background information in the Materials and Methods section (pg. 25, lines 613–617).

2 - In MARCM analysis (Fig 3A, 4C, 5C), wild-type clones appear very small (one or two cells only on most images), which is problematic in assessing lineage phenotypes reliably. Tissues should be dissected at later time points or heat-shock conditions modified to ensure sufficient clone growth in control conditions.

<Response to the reviewer>

We thank the reviewer for this suggestion. We induced clones by heat-shocking 3–5-day-old flies once at 37 °C for 45 min, a condition commonly used in intestinal MARCM studies (e.g., Tauc et al., *Development*, 2017). Clones were analyzed at 3, 7, and 14 days after induction (*aci*). Approximately 54–60% of clones in both control and *dwdr4* heterozygous guts consisted of a single cell, consistent with a previous report (Neophytou & Pitsouli, 2022), where 2–4-day-old flies were heat-shocked for 1 h. This likely reflects GFP labeling of an EB or pEE, the immediate daughter of a dividing ISC, after recombination.

To avoid potential confounding effects of aging, we did not analyze clones at 21 days *aci*. Importantly, despite the small size of many clones, our results consistently showed that *dwdr4* mutant ISCs divide faster than control ISCs. We further validated this finding by ISC-specific knockdown of *dWdr4* using ISC-specific GAL4, which produced similar outcomes. Therefore, while we did not test longer heat-shock durations or later times, multiple independent approaches confirmed the robustness of our conclusions.

3 - Some observations lack quantifications, which should be performed throughout multiple guts and added to the figures. The fly gut is, in general, a highly variable tissue in immunofluorescence assays and quantitative statements based on one microscopy image are difficult. Please provide quantifications for Fig 4A, B, and C, with a sufficient wild-type clone size for the latter.

<Response to the reviewer>

We thank the reviewer for this critical comment. In the revised manuscript, we have

added quantification for all relevant experiments, based on multiple guts to account for tissue variability. These quantifications are now included in the figures and figure legends.

4 - There is no statement on data availability. RNA-seq data should be deposited in an appropriate database. Also, basic statistics of sequenced RNA-seq libraries should be provided. It is not clear if the analysis was performed in replicates.

<Response to the reviewer>

We thank the reviewer for pointing out this. Our RNA-seq data has been deposited in Gene Expression Omnibus (GEO) under accession number GSE298087. We have included this information in the current manuscript (pg. 32 line 782-783).

While, apart from the few experimental shortcomings mentioned above, the dWdr4 involvement in the control of ISC proliferation is overall well supported by the data presented, unfortunately, I have stronger concerns regarding the mechanistic insights provided further in the manuscript.

5 - The conclusion that dWdr4 regulates ribosome biogenesis (Fig 5) appears not convincingly supported with the experimental evidence.

- The in silico analysis of genes "predictive of dwdr4 expression" (Fig 5A) by itself is not convincing. Are the identified gene categories (involved in protein translation, ribosome biogenesis and regulation) also affected in the dWdr4-deficient RNA-seq data? Also, RNA-seq data from which tissue type were used for this analysis?

<Response to the reviewer>

We thank the reviewer for raising this question. We performed our analyses using RNA-seq data available from FlyBase (RNA-Seq RPKM values, <https://flybase.org/downloads/bulkdata>). These data comprise normalized RNA-seq counts from 166 samples of wild-type *Drosophila* spanning various developmental phases, tissue types, and organs. This large dataset was used to identify genes whose expression correlated significantly with that of dWdr4. Our rationale was that genes associated with a particular pathway are often coregulated/coexpressed due to the activity of shared upstream transcription factors. Thus, our analysis is suited for making testable predictions about the pathways in which dWdr4 may participate, but not whether the loss of dWdr4 (just one component of a large regulatory network) would result in widespread transcriptional changes in other members of the pathway. Nevertheless, as several ribosomal protein gene transcripts were significantly altered in dWdr4 mutant guts compared with controls, we conclude that some genes whose expression is positively correlated with dWdr4 are differentially expressed in response to the loss of dWdr4. This information has been included in Appendix Table S1B.

- Moreover, conclusions on ribosome biogenesis drawn solely on the provided measure of nucleolar (fibrillar) size (Fig 5B and C, the latter not quantified and with **small** wt clones), appear rather weak. Primarily because authors reported that esg positive dWdr4 RNAi cells show increased nuclear size ("mis-differentiated" ECs), fibrillar size should rather be reported as a ratio of nucleolar to DNA (nucleus)

volume (as previously done, e.g. <https://doi-org.insb.bib.cnrs.fr/10.1038/ncb1223>).

<Response to the reviewer>

We thank the reviewer for this valuable and constructive comment. In the revised manuscript, we have reanalyzed all Fibrillarin data and now present it as the ratio of nucleolar (Fibrillarin) to nuclear (DAPI) area, following the approach used in previous studies. This quantification has been incorporated into the figures and figure legends, as well as the Materials and Methods section (pg. 31 line 756-758)..

- Finally, some other measures of ribosome function should be used to further support the authors conclusions, e.g. rRNA and ribosomal genes transcript levels.

<Response to the reviewer>

We thank the reviewer for this critical comment. As suggested, we performed 5-ethynyl uridine (EU) incorporation in *dwdr4* mutant guts to detect nascent RNA synthesis, of which ribosomal RNA (rRNA) is the predominant species. Consistent with our Fibrillarin-marked nucleoli results, the intensity of EU signals was markedly increased in ISCs (marked by Delta) and in ECs (Delta-negative cells with large nuclei) of *dwdr4* mutant posterior midguts (PMGs) compared with controls. Moreover, the ratio of 5'-EU (nucleolar) to DAPI (nuclear) area in ISCs was significantly elevated. These findings indicate that *dWdr4* depletion enhances ribosome biogenesis. We have incorporated these results in Fig. 4C and C' and added the corresponding description in the main text (p. 11, lines 256–259).

As suggested by reviewer 1, we also asked whether increased ribosome biogenesis accounts for the gut phenotypes caused by *dWdr4* loss. To do so, we treated flies with BMH-21, a small molecule that suppresses RNA polymerase I-mediated rDNA transcription and is widely used to inhibit ribosome biogenesis in cultured cells, flies, and mice (Cho et al., Cellular and Molecular Gastroenterology and Hepatology, 2025; Gui et al., Cell Reports, 2021). Four-day-old wild-type and *wh7* mutant flies were treated with 5 μ M or 25 μ M BMH-21 for 3 days. We found that 5 μ M BMH-21 was sufficient to reduce the Fibrillarin/DAPI area ratio in ISCs/EBs of both wild-type and *wh7* mutant PMGs compared with untreated controls. While PHH3⁺ ISC numbers were not reduced in wild-type guts, they were significantly decreased in *dwdr4* mutant guts, along with suppression of the ISC/EB expansion phenotype. These results identify *dWdr4* as a key regulator of ribosome biogenesis in ISCs, with its loss triggering elevated ribosome biogenesis, ISC overproliferation, and ISC/EB expansion. We have incorporated these results into Fig. 4D–D'' and described in the main text (p. 11, lines 260–270).

We attempted genetic manipulation by removing one copy of the Minute gene (ribosomal protein small subunit RpS17 or large subunit RpL15[8-1]) in the *wh7* mutant background. However, these flies were extremely unhealthy; therefore, we opted to use BMH-21 treatment instead of Minute gene reduction.

Thus, in the current version of the manuscript, the assumptions on increased ribosomal biogenesis upon *dWdr4* depletion (present throughout the text) are not well supported with experimental evidence. As a consequence, conclusions such as "our study shows that *dWdr4* is an essential regulator of ribosome biogenesis"

(Discussion and elsewhere) are overstated.

<Response to the reviewer>

We thank the reviewer for this critical comment. In the revised manuscript, we have strengthened our evidence by including analyses of the nucleolar-to-nuclear area ratio, EU incorporation assays, and BMH-21 treatment. With these additional results, we are confident in concluding that *dWdr4* is an essential regulator of ribosome biogenesis.

6 - Regarding the *dWdr4*-*Mettl1*-*let7* m7G axis, the following concerns can be noted:
- On multiple occasions authors imply that *dWdr4* and *Mettl1* act in complex in the fly gut. In absence of any experimental evidence of the interaction between the two proteins in this system, such statements should be avoided. Also, please provide quantification on multiple guts for Fig 7A and S8A.

<Response to the reviewer>

We thank the reviewer for this insightful comment. To test whether *dWdr4* associates with *dMettl1* in the intestine, we performed proximity ligation assays (PLA) in posterior midguts of flies carrying a *dWdr4*-GFP transgene and expressing *dMettl1*-V5 in ISCs/EBs under *esg*-GAL4. PLA puncta were detected when both anti-GFP and anti-V5 antibodies were applied, whereas no signal was observed in the control stained with anti-V5 alone. These data indicate that *dWdr4*-GFP and *dMettl1*-V5 are in proximity in ISCs/EBs, consistent with the formation of a *dWdr4*-*dMettl1* complex. This result is now included in Fig EV6C, C', and described in the text (p. 14, lines 345–349).

We have added quantification data for Fig. 7A and S8A from multiple guts, which are now presented as graphs in Fig. EV6A' and B' of the revised manuscript.

- Does *dWdr4* KD lead to decreased *Mettl1* levels (RNA or protein)? Can *dwdr4* RNAi phenotypes be rescued by *Mettl1* overexpression?

<Response to the reviewer>

We thank the reviewer for this question. In our RNA-seq data, *Mettl1* transcript levels are not reduced upon *dWdr4* loss, and a validated anti-*Drosophila* *Mettl1* antibody is not currently available to assess protein abundance. Nevertheless, based on prior studies and our own evidence that *Wdr4* and *Mettl1* act together as a complex, we consider it unlikely that *Mettl1* overexpression alone would rescue the phenotypes caused by *dWdr4* depletion.

- Finally, *let7* over-expression rescue experiments convincingly support the notion that *dWdr4* phenotypes are, at least in part, mediated by diminished levels of *let-7*. However, since *let7* m7G modification (and its loss upon *dWdr4* knock-down) is not demonstrated, the involvement of this modification and *Mettl1* protein in *let7* stability remains only a hypothesis based on previous studies.

<Response to the reviewer>

We appreciate the reviewer's concern. In the revised manuscript, we immunoprecipitated m7G-modified RNAs from control female guts using an anti-m7G antibody, followed by RT-PCR to detect *let-7* levels. A sno-lncRNA was used as a negative control. Our results showed that m7G-methylated *let-7* represented $1.7 \pm$

0.3% of the input, compared to $0.04 \pm 0.05\%$ for the sno-lncRNA ($P < 0.001$), demonstrating that *let-7* in the gut can be m7G-methylated. These results are presented in Section 7B and described on pg. 15, lines 371–372.

We did not perform this experiment in *dwdr4* mutant females due to technical challenges. *dwdr4* is X-linked, and homozygous females are sterile. To obtain mutant females, it is first necessary to generate *dwdr4/Y; dwdr4-gfp/CyO* males and then cross them with *dwdr4/FM7c* flies, which produces *dwdr4* mutant females at ~25% frequency (see Appendix Information, pg. 2). In addition, each RNA immunoprecipitation requires 200–300 guts, further increasing the experimental difficulty. Nevertheless, we showed that *let-7* levels were reduced in guts with ISC/EB-specific *dWdr4* depletion (Fig. 7C, pg. 15, lines 372–374), consistent with a requirement for m7G methylation in *let-7* processing (Pandolfini et al., Molecular Cell, 2019).

Although we did not perform the m7G pull-down in *dMettl1*-depleted intestines, multiple lines of evidence support a role for *dMettl1* in mediating m7G modification of *let-7*. Specifically, *dMettl1* knockdown reduced m7G levels in the PMG (Fig. EV6B), similar to *dWdr4* knockdown (Fig. EV6A), and phenocopied the gut phenotypes induced by *dWdr4* depletion (Fig. 7A, A'). Furthermore, human wild-type METTL1, but not its catalytically inactive mutant (METTL1^{DK}), rescued the phenotypes caused by *dMettl1* depletion (Fig. EV6E, E'). In addition, *dWdr4* physically interacts with *dMettl1* (Fig. EV6C). Together, these findings strongly support our conclusion that the m7G modification of *let-7* is mediated by the *dWdr4*-*dMettl1* complex, which is essential for maintaining intestinal homeostasis.

7 - Finally, the experiments of human WDR4 overexpression rescue are barely discussed and are not convincing based only on Fig 7G. **Please provide quantification** for this figure panel (as in Fig 2A' and the mitotic index). The text states "OE of human WDR4 in ISCs/EBs also rescued the *dWdr4*-deficiency phenotypes" - please specify which phenotypes (all?) were rescued and show the data.

<Response to the reviewer>

We thank the reviewer for this suggestion. We overexpressed human WDR4 in ISCs/EBs of *dwdr4* mutant intestines using *esg>GAL4* and examined ISC/EB expansion (assessed by *esg>mCD8gfp*-positive cells) and ISC overproliferation (assessed by PHH3-positive cells). Our results show that ISC/EB expansion is dramatically reduced (*dwdr4* mutant: 15%, 28%, and 42% of gut sections displayed strong, intermediate, and individual phenotypes, respectively; vs. *dwdr4* mutant with hWDR4: 4%, 9%, and 12% of gut sections, respectively). In addition, the number of mitotic ISCs was also significantly reduced (*dwdr4* mutant: 24.3 ± 17.6 PHH3⁺ ISCs, $n = 13$; vs. *dwdr4* mutant with hWDR4: 6.8 ± 5.5 PHH3⁺ ISCs, $n = 16$; $P < 0.001$). In this revised manuscript, we have added these quantifications to Fig. EV6D' and E, and the corresponding figure legends.

Additional minor points:

Page 9:

- Please spell out PHH3 when used for the first time.

<Response to the reviewer>

We thank the reviewer for the helpful suggestion. We have added the full name of PHH3 (phosphorylated histone H3) [pg. 8, line 176-177] when it first appears in the text to improve clarity.

Change to "ISCs are the major cell type known to undergo division" in light of the fact that EEp divides to create 2 EEs

<Response to the reviewer>

We thank the reviewer for this correction. We have revised the sentence to "*ISCs are the primary cell type known to undergo division* (pg. 8, line 180)", to more accurately reflect that EEPs can also divide.

For clarity, authors should report all results in the Results section, rather than reporting some to the Discussion (Fig S10).

<Response to Reviewer>

We thank the reviewer for this suggestion. To ensure a smoother flow of the manuscript, we have moved the data previously shown in Fig. S10 as Appendix Fig. S1, and we now reference it in the Introduction.

Materials and Methods:

- Please provide the composition of the Drosophila medium used

<Response to Reviewer>

We thank the reviewer for this suggestion. We used standard cornmeal–agar–yeast food, and this information has now been included in the Materials and Methods section (pg. 19, line 474).

Referee #3:

In this manuscript, Kajal and co-workers examine the role of Wdr4 in the proliferation of stem cells in the fruit fly intestine. The authors show evidence that Wdr4 is required for wild-type survival and intestinal homeostasis. Wdr4 is expressed in intestinal stem cells (ISC) and plays a role in their proliferation and gut morphogenesis. They show that Wdr4 impacts let-7 expression and subsequently TOR activation, ROS levels and Myc activity. They present evidence that Wdr4 regulates ribosome biogenesis in ISCs via Myc and TOR. Overall the study is **nicely executed** but I believe there are a few issues that would need to be addressed before I can recommend publication:

<Response to the reviewer>

We thank the reviewer for their positive evaluation of our study.

Major issues:

1) There are some key points that I am not persuaded are well supported by the data: I think the authors will have to provide new data or substantially change their conclusions.

1a) The authors indicate that Wdr4 may impact let-7 stability though its m7G modification but the only data presented indicate transcriptional regulation of let-7 is impacted by loss of Wdr4.

<Response to the reviewer>

We appreciate the reviewer's concern. In the revised manuscript, we immunoprecipitated m7G-modified RNAs from control female guts using an anti-m7G antibody, followed by RT-PCR to detect let-7 levels. A sno-lncRNA was used as a negative control. Our results showed that m7G-methylated let-7 represented $1.7 \pm 0.3\%$ of the input, compared to $0.04 \pm 0.05\%$ for the sno-lncRNA ($P < 0.001$), demonstrating that let-7 in the gut can be m7G-methylated. This result is included in Fig. 7B and described on pg. 15, lines 371–372.

We did not perform this experiment in *dWdr4* mutant females due to technical challenges. *dWdr4* is X-linked, and homozygous females are sterile. To obtain mutant females, it is first necessary to generate *dWdr4/Y; dWdr4-gfp/CyO* males and then cross them with *dWdr4/FM7c* flies, which produces *dWdr4* mutant females at ~25% frequency (see Appendix Information, pg. 2). In addition, each RNA immunoprecipitation requires 200–300 guts, further increasing the experimental difficulty. Nevertheless, we showed that let-7 levels were reduced in guts with ISC/EB-specific *dWdr4* depletion (Fig. 7C, pg. 15, lines 372–374), consistent with a requirement for m7G methylation in let-7 processing (Pandolfini et al., *Molecular Cell*, 2019).

By combining our results that *dMettl1* depletion reduces m7G levels in the PMG (Fig. EV6B), *dMettl1* knockdown in ISCs/EBs phenocopies the phenotypes of *dWdr4*-depleted intestines (Fig. 7A, A'), human wild-type METTL1 but not its catalytically inactive mutant rescues *dMettl1* depletion phenotypes (Fig. EV6E, E'), and *dWdr4* physically interacts with *dMettl1* (Fig. EV6C), we conclude that m7G modification of let-7 is mediated by the *dWdr4*-*dMettl1* complex.

1b) The authors propose that ribosome biogenesis is regulated by *Wdr4*, but there are no direct measures of ribosome biogenesis, only a convenient but indirect measure of nucleolar size.

<Response to the reviewer>

We thank the reviewer for this critical comment. As suggested, we performed 5-ethynyl uridine (EU) incorporation in *dWdr4* mutant guts to detect nascent RNA synthesis, of which ribosomal RNA (rRNA) is the predominant species. Consistent with our Fibrillarin-marked nucleoli results, the intensity of EU signals was markedly increased in ISCs (marked by Delta) and in ECs (Delta-negative cells with large nuclei) of *dWdr4* mutant posterior midguts (PMGs) compared with controls. Moreover, the ratio of 5'-EU (nucleolar) to DAPI (nuclear) area in ISCs was significantly elevated. These findings indicate that *dWdr4* depletion enhances ribosome biogenesis. We have incorporated these results in Fig. 4C and C' and added the corresponding description in the main text (p. 11, lines 256–259).

1c) The authors propose that ISC proliferation is regulated by *Wdr4* via ribosome biogenesis but this is not directly tested (i.e. is ribosome biogenesis necessary and/or sufficient for *Wdr4* effects on ISC proliferation)

<Response to the reviewer>

As suggested by reviewer 1, we also asked whether increased ribosome biogenesis accounts for the gut phenotypes caused by *dWdr4* loss. To do so, we treated flies with

BMH-21, a small molecule that suppresses RNA polymerase I-mediated rDNA transcription and is widely used to inhibit ribosome biogenesis in cultured cells, flies, and mice (Cho et al., Cellular and Molecular Gastroenterology and Hepatology, 2025; Gui et al., Cell Reports, 2021). Four-day-old wild-type and *wh7* mutant flies were treated with 5 μ M or 25 μ M BMH-21 for 3 days. We found that 5 μ M BMH-21 was sufficient to reduce the Fibrillarin/DAPI area ratio in ISCs/EBs of both wild-type and *wh7* mutant PMGs compared with untreated controls. While PHH3⁺ ISC numbers were not reduced in wild-type guts, they were significantly decreased in *dWdr4* mutant guts, along with suppression of the ISC/EB expansion phenotype. These results identify *dWdr4* as a key regulator of ribosome biogenesis in ISCs, with its loss triggering elevated ribosome biogenesis, ISC overproliferation, and ISC/EB expansion. We have incorporated these results into Fig. 4D–D'' and described in the main text (p. 11, lines 260–270).

We attempted genetic manipulation by removing one copy of the Minute gene (ribosomal protein small subunit RpS17 or large subunit RpL15[8-1]) in the *wh7* mutant background. However, these flies were extremely unhealthy; therefore, we opted to use BMH-21 treatment instead of Minute gene reduction.

2) The fly lines used for lifespan assays do not appear to have been backcrossed. Backcrossing is essential for fly lifespan experiments. I suggest the authors either backcross the lines and redo the experiments or exclude lifespan analysis from the manuscript.

<Response to the reviewer>

We thank the reviewer for this important comment. In the current manuscript, the lifespan assays were performed with *wh7* flies that had been backcrossed to a *yw* strain for 10 generations (Fig. EV1A, B). Our results showed that *dWdr4* regulates lifespan in both males and females, consistent with data obtained from non-backcrossed *dWdr4* mutants. We have now included this genetic background information in the Materials and Methods section (pg. 25, lines 612–617).

For assessing the lifespan of flies bearing *dWdr4*-depleted ISCs, we used *esgts*-GAL4 to knock down *dWdr4* (TRIP RNAi line) and a TRIR RNAi control generated from the BDSC, both of which share the same genetic background. Consistent with our previous results in which a *esgts*>*mCherry*RNAi was used as the control genotypes, flies with *dWdr4*-depleted ISCs showed a slightly but significantly reduced lifespan. We have now included this data in Fig EV2H, and this genetic background information in the Materials and Methods section (pg. 25, lines 618–623).

3) RNA-Seq raw data will need to be deposited in a suitable repository before publication. False discovery rates will need to be stated for all analyses.

<Response to the reviewer>

We thank the reviewer for this important comment. Our RNA-seq data have been deposited in Gene Expression Omnibus (GEO) under accession number GSE298087. We have included this information in the current manuscript (pg. 32, line 781-782).

In the figures presenting RNA-seq results, we originally showed fold changes for genes

with $p < 0.05$. We now additionally report p -values with corresponding false discovery rates (FDR) for all genes listed in Appendix Table S1.”

4) There are two experiments that are missing a genotype required for full interpretation of the results:

4a) Figure 5 - in addition to *dmyc4/+* *DI>dwdr4* I think the authors need to include *dmyc4/+* alone. If this results in a reduction in pH3+ cells and nucleolar size in the absence of RNAi equivalent to what is observed in the presence of RNAi, the authors conclusions may be different.

<Response to the reviewer>

We thank the reviewer for this critical comment. In the revised manuscript, we have analyzed ISC proliferation and ribosome biogenesis in *dmyc4/+* guts. Our results show that *dmyc4/+* guts display numbers of proliferative ISCs and nucleolar-to-nuclear area ratios comparable to controls. These results have now been incorporated in Fig. 5F, F', and Fig. EV3C', and described shortly in the main text, pg. 12, line 288-291.

4b) Similarly, in Figure 7, *let-7* OE alone is missing.

<Response to the reviewer>

We thank the reviewer for the comment. In the current manuscript, we have now included data on mitotic ISC numbers in *esgts > let-7-OE* PMGs, alongside the other relevant genotypes. Our results show that overexpression of *let-7* significantly suppressed ISC overproliferation, ISC/EB expansion, and elevated ribosome biogenesis induced by *dWdr4* depletion, whereas overexpression of *let-7* alone did not produce obvious differences compared with controls. We have now included the *let-7* overexpression—alone data in Fig. 7C' and briefly described it in the main text (pg. 16, lines 377–378).

Since *let-7* overexpression alone did not affect ISC proliferation or ribosome biogenesis, we did not further examine whether this manipulation influences TOR or JNK signaling; these pathways were specifically analyzed in the context of *dwdr4* depletion combined with *let-7* overexpression.

5) Some important literature, from the area of ageing, ribosome biogenesis and ISCs, does not appear to have been cited: For example:

<https://doi.org/10.1038/ncomms16083>

<https://doi.org/10.1016/j.celrep.2020.01.017>

<Response to the reviewer>

We thank the reviewer for bringing these important references to our attention. We have now included both citations in the revised manuscript (pg.3, lines 56-57) to appropriately acknowledge relevant work in the areas of aging, ribosome biogenesis, and ISC regulation.

Minor:

6) In the abstract, "which suppresses TOR signaling to preserve steady-state ribosome biogenesis in ISCs" is awkward, as TOR is expected to promote ribosome

biogenesis.

<Response to the reviewer>

We thank the reviewer for pointing this out. We agree that the original phrasing was misleading, as TOR signaling typically promotes ribosome biogenesis. We have revised the sentence in the abstract to more accurately reflect our findings and the role of TOR signaling in this context.

7) For most of the introduction, it is unclear what is referring specifically to the fly gut rather than the mammalian (or their) gut.

<Response to the reviewer>

We thank the reviewer for this helpful comment. In the revised Introduction, we have clarified throughout which statements refer specifically to the *Drosophila* gut versus the Mammalian gut.

Dear Prof. Hsu,

Thank you for the submission of your revised manuscript to our editorial offices. I have now received the reports from the three referees that I asked to re-evaluate the study, you will find below. As you will see, all three referees now support publication of your study in EMBO reports. However, all three referees have remaining concerns or suggestions to improve the manuscript, I ask you to address in a final revised manuscript. Please change the manuscript accordingly and discuss the indicated limitations in the text. Please provide a final p-b-p-response to the remaining referee points and my editorial requests.

Editorial requests:

- Please reduce the number of keywords to five and order the manuscript sections like this, using only these names: Title page - Abstract - Keywords - Introduction - Results - Discussion - Methods - Data availability section - Acknowledgements - Disclosure and Competing Interests Statement - References - Figure legends

- Please make sure that all the funding information is also entered into the online submission system and that it is complete and similar to the one in the acknowledgement section of the manuscript text file.

- Please provide final individual production quality figure files as .eps, .tif, .jpg (one file per figure), of main figures and EV figures. Please upload these as separate, individual files upon re-submission. Please combine some panels of the EV figures to have not more than 6 final EV figures. Then, please update all the affected callouts.

- Please name the pdf file 'Appendix information' file 'Appendix'. The Appendix should have page numbers and needs to include a table of content (TOC) on the first page (with page numbers). Please state on the first page 'Appendix for ...' followed by title, with the TOC below. Please follow the nomenclature 'Appendix Figure Sx' and 'Appendix Table Sx' throughout the text (for the callouts) and also label the figures and tables in the Appendix file according to this nomenclature. For the figure, please move the respective legend below the Appendix figure. Moreover, we do not allow methods information in the Appendix. Please move the part "Fly genetics for generating wh7 mutants with and without other genetic elements" and the related reference to the main manuscript text.

- There are two EV Tables mentioned at the end of the main manuscript text file but not provided. I would suggest adding Table EV1 to the Appendix. In contrast, the primer information provided in table EV2 needs to be added directly to the Reagents & Tools Table. Please do that and add callouts to the methods section where appropriate. Finally, please remove the mention of EV tables from the main manuscript text.

- Please check again that the number "n" for how many independent experiments were performed, their nature (biological versus technical replicates), the bars and error bars (e.g. SEM, SD) and the test used to calculate p-values is indicated in the respective figure legends (main, EV and Appendix figures). Please also check that all the p-values are explained in the legend, and that these fit to those shown in the figure. Please provide statistical testing where applicable. Please avoid the phrase 'independent experiment' but clearly state if these were biological or technical replicates. Please also indicate (e.g. with n.s.) if testing was performed, but the differences are not significant. In case n=2, please show the data as separate datapoints without error bars and statistics. See also:

<http://www.embopress.org/page/journal/14693178/authorguide#statisticalanalysis>

- Please add to each legend (main, EV and Appendix figures, where applicable) a 'Data Information' section explaining the statistics used or providing information regarding replicates and scales. See:

- Please check again that our reference format is used:

- Please provide a complete author checklist, which you can download from our author guidelines (<https://www.embopress.org/page/journal/14693178/authorguide>). Please insert page numbers in the checklist to indicate where the requested information can be found in the manuscript. The completed author checklist will also be part of the RPF.

- We now request the publication of original source data with the aim of making primary data more accessible and transparent to the reader. You will receive a separate email with instructions for providing source data with your revised manuscript, including information how to upload and organize the files. Please upload this as one folder per main figure, grouping together all the files for this figure (and ZIPed together), and as one folder for the EV figures containing separate folders for each EV figure and one folder for the Appendix figures.

- Please add scale bars of similar style and thickness to all microscopic images, using clearly visible black or white bars (depending on the background). Please place these in the lower right corner of the images themselves. Please do not write on or near the bars in the image but define the size in the respective figure legend. Presently many inserts or magnification boxes are missing scale bars. Please also provide scale bars for panel 2C.

In addition, I would need from you uploaded separately (please remove this from the manuscript text file):

Please let us know whether (1) you agree with the publication of the RPF and (2), if you want to remove or not any figures from it prior to publication. Please note that retaining unpublished data in the Peer Review File means that these count as published and that the Peer Review File would need to be referenced in future publications.

I look forward to seeing a revised form of your manuscript when it is ready.

Yours sincerely,

Referee #1:

I find that in this re-submitted manuscript, Kajal and colleagues have made a very substantial improvement of the paper. It is clearer and thoroughly supported by the data (which has increased considerably), with more direct observations at both the molecular and cellular level. I think that all the concerns I raised have been generously addressed -- I want in particular to thank the authors for their patient explanation of how to interpret the DHE fluorescence, about which I had a narrow view.

The paper shows now clearly and convincingly the role of *dWdr4/dMettl1/let-7* in intestinal homeostasis and, more importantly, the effects of impaired ribosomal biogenesis on tissue homeostasis, which will be helpful for the field in general.

I only want to point out that the 'Drosophila' way to refer to the *_wuho_* mutant specimens should be *_wuho*^[7] ([] for superscript, *_* for italics), and that any journal that would follow FlyBase conventions (i.e. Development or Journal of Cell Science) would expect this to be followed. It is not my role to enforce this, but I'd like to point it out to the authors that I believe this should be the way to refer to the mutant. The other choices the authors took seem fine to me.

This is the only reason I suggest minor revision. I think I have spotted a few very minor language errors that can be dealt with at production.

Referee #2:

The revised version of the manuscript by Kajal et al is greatly improved in many aspects. For most part, my concerns regarding

quantifications of phenotypes and lack of experimental support for the proposed mechanisms have been addressed, often with more than one approach. I would like to congratulate the authors for this extensive revision that significantly improved the manuscript.

I do have a few final comments, regarding the text (conclusions and data interpretation) that I strongly believe should still be addressed. From my point of view, these do not require any additional experiments, but rather rewriting some statements in the text that are not fully in accordance with the data provided.

In my opinion the most important are the following two points:

1- Authors should avoid immediate indirect conclusions when discussing their results.

E.g. line 289: It should be: "treatment with rapamycin... suppressed ISC proliferation and increased nucleolar size...", rather than "elevated ribosome biogenesis". What is actually measured is nucleolar size and it should be stated as such. Later, one may state that this implies rescue of the elevated ribosome biogenesis.

The same is true in other places of the manuscript, e.g. line 378 and 384.

2- Even more importantly, although I do appreciate all the new results that provide more evidence behind the proposed Wdr4-Mettl1-let7 axis, the manuscript does not demonstrate that in *dwdr4* or *mettl1* mutant/RNAi conditions let-7 m7G modification is abolished/decreased. Thus, I do not think that statements such as "Wdr4-Mettl1-mediated m7G modification of let-7..." (page 16, lines 359 and 392, paragraph titles; and line 1205, title of Fig 7) are appropriate. Authors need to tone down such phrasing throughout the text and the abstract. Otherwise, they are overinterpreting their results.

Other minor comment:

Line 93-94: Data from the current Appendix Fig S1 should not be discussed in the Introduction. In the previous version of the manuscript the data was included only in the discussion. After my comment, the authors have decided to move it to the introduction. It actually makes it very confusing if these data are new or have previously been published in the referenced paper. If these are new result, please discuss them in the results section.

Line 101: "These findings (...) link (*Wdr4*) expression to the outcomes of gastrointestinal cancer"

The results presented in this manuscript do not provide such a link. Thus, this is a misleading statement. Please remove, or rephrase that such a possibility could be further tested.

Line 107 and throughout: Although I have not noticed this before, for which I apologize, *wh* is not the official Flybase gene name for *Drosophila* *dwdr4*. This is very confusing. As I understand, the gene and the mutant allele were originally called *wh* and renamed later to *wuho*. By insisting on using the old nomenclature the authors make it difficult to find the relevant information on FlyBase.

Line 428: Rephrase the sentence.

Materials and Methods: As I suggested earlier, please provide the exact composition of the fly food. "Standard" is not very meaningful as cornmeal-yeast-agar recipes differ between laboratories, e.g. the amount of yeast (m/v %) varies, which is known to influence gut function.

Referee #3:

The authors have done an adequate job at addressing my original comments.

Regarding my original points 1a, 1b and 1c, I think that the additional data presented give better support to the conclusions. Responses to points 3,4,5,6 and 7 are adequate.

Regarding my original point 2. It's good that the mutant was backcrossed (even though this may have been done a long time ago). However, it cannot be assumed that RNAi lines that were generated as part of the TRiP project are in the same background - flies can accumulate spontaneous mutations over time. This, the very small effect observed in Figure EV2H (compared to the effect of the mutation) and the lack of GAL4-alone and UAS-alone controls, all makes me think that this result provides quite weak support for the authors' conclusion.

On balance, I am recommending publication in EMBO reports.

Referee #1:

I find that in this re-submitted manuscript, Kajal and colleagues have made a very substantial improvement of the paper. It is clearer and thoroughly supported by the data (which has increased considerably), with more direct observations at both the molecular and cellular level. I think that all the concerns I raised have been generously addressed -- I want in particular to thank the authors for their patient explanation of how to interpret the DHE fluorescence, about which I had a narrow view.

The paper shows now clearly and convincingly the role of dWdr4/dMettl1/let-7 in intestinal homeostasis and, more importantly, the effects of impaired ribosomal biogenesis on tissue homeostasis, which will be helpful for the field in general.

I only want to point out that the 'Drosophila' way to refer to the *wuho* mutant specimens should be *wuho*⁷ ([⁷] for superscript, *wuho* for italics), and that any journal that would follow FlyBase conventions (i.e. Development or Journal of Cell Science) would expect this to be followed. It is not my role to enforce this, but I'd like to point it out to the authors that I believe this should be the way to refer to the mutant. The other choices the authors took seem fine to me.

This is the only reason I suggest minor revision. I think I have spotted a few very minor language errors that can be dealt with at production.

<Response to the reviewer>

We thank the reviewer for raising this point. *Drosophila wdr4* (dWdr4) was originally identified in Wu, Hou, and Hsieh (Dev Biol, 2006), where the gene was named *wuho*, derived from the first authors' surnames (Wu and Hou) and also referring to the sterility phenotype ("no progeny" in Chinese). In that study, the *wuho*⁷ (*wh*⁷) mutant allele was generated by P-element excision, and this allele designation (*wh*⁷) has since been recorded in FlyBase.

In the manuscript, we introduce the gene as dWdr4 (historically named *Wuho*, *Wh*) at its first mention (line 87), and use *wh* strictly when referring to the original mutant allele.

We would also like to acknowledge that this work builds on foundational studies led by Dr. Tao-Shih Hsieh, who made major contributions to *wuho* gene research and to the development of *Drosophila* genetics in Taiwan. Dr. Hsieh passed away in 2016, and retaining a brief mention of the historical gene name respects this scientific lineage while still aligning with current FlyBase nomenclature.

We hope the reviewer finds this clarification appropriate.

Referee #2:

The revised version of the manuscript by Kajal et al is greatly improved in many aspects. For most part, my concerns regarding quantifications of phenotypes and lack of experimental support for the proposed mechanisms have been addressed, often with more than one approach. I would like to congratulate the authors for this extensive revision that significantly improved the manuscript.

I do have a few final comments, regarding the text (conclusions and data interpretation) that I strongly believe should still be addressed. From my point of view, these do not require any

additional experiments, but rather rewriting some statements in the text that are not fully in accordance with the data provided.

In my opinion the most important are the following two point:

1- Authors should avoid immediate indirect conclusions when discussing their results.

E.g. line 289: It should be: "treatment with rapamycin... suppressed ISC proliferation and increased nucleolar size...", rather than "elevated ribosome biogenesis". What is actually measured is nucleolar size and it should be stated as such. Later, one may state that this implies rescue of the elevated ribosome biogenesis.

The same is true in other places of the manuscript, e.g. line 378 and 384.

<Response to the reviewer>

We thank the reviewer for this helpful comment. We agree that the statements should directly reflect the measurements performed. Accordingly, we have revised the relevant sentences to explicitly describe nucleolar size rather than ribosome biogenesis, and we now refer to increased ribosome biogenesis only as an inference in the interpretation.

[Line 283] Consistently, treating *dwdr4* mutant flies or flies bearing *dwdr4* knockdown ISCs with rapamycin, a TOR inhibitor (Lamming, 2016), suppressed ISC proliferation, reduced ISC/EB clustering, and decreased nucleolar enlargement in ISCs/EBs.

[Line 289] Furthermore, decreasing dMyc expression in *esgts>dwdr4RNAi* flies by removing a copy of *dmyc* suppressed both ISC overexpression and enlarged nucleoli in ISCs/EBs, indicating that elevated ribosome biogenesis upon *dWdr4* depletion requires dMyc.

[Line 377] Moreover, overexpression of *let-7* in ISCs/EBs significantly suppressed ISC overproliferation, ISC/EB expansion, and nucleolar enlargement, demonstrating that elevated ribosome biogenesis in *dWdr4*-depleted intestines is mediated by *let-7* loss.

2- Even more importantly, although I do appreciate all the new results that provide more evidence behind the proposed *Wdr4-Mettl1-let7* axis, the manuscript does not demonstrate that in *dwdr4* or *mettl1* mutant/RNAi conditions *let-7* m7G modification is abolished/decreased. Thus, I do not think that statements such as "*Wdr4-Mettl1*-mediated m7G modification of *let-7*..." (page 16, lines 359 and 392, paragraph titles; and line 1205, title of Fig 7) are appropriate. Authors need to tone down such phrasing throughout the text and the abstract. Otherwise, they are overinterpreting their results.

<Response to the reviewer>

We agree with the reviewer's point of view. We have softened the description as "*dWdr4-dMettl1* maintains *let-7* levels, likely through m7G modification, to control TOR signaling (p.g. 15, line 354)" and "*dWdr4* maintains *let-7*, which can be m7G modified, to regulate the TOR-JNK-dMyc regulatory axis and maintain ISC homeostasis (line 1214, title of Fig 7).

Other minor comment:

Line 93-94: Data from the current Appendix Fig S1 should not be discussed in the Introduction. In the previous version of the manuscript the data was included only in the discussion. After my comment, the authors have decided to move it to the introduction. It actually makes it very confusing if these data are new or have previously been published in the referenced paper. If these are new result, please discuss them in the results section.

<Response to the reviewer>

We understand the reviewer's concern. In the revised manuscript, we have removed the description of the Appendix Fig S1 data from the Introduction and moved it to the Results section (page 5, line 109).

Line 101: "These findings (...) link (Wdr4) expression to the outcomes of gastrointestinal cancer" The results presented in this manuscript do not provide such a link. Thus, this is a misleading statement. Please remove, or rephrase that such a possibility could be further tested.

<Response to the reviewer>

We thank the reviewer for this critical point. We have rephrased the sentence as "These findings reveal a novel mechanism by which Wdr4 safeguards ISC homeostasis and suggest a possible association with gastrointestinal cancer outcomes (p.g. 4, line 98).

Line 107 and throughout: Although I have not noticed this before, for which I apologize, wh is not the official Flybase gene name for drosophila dwdr4. This is very confusing. As I understand, the gene and the mutant allele were originally called wh and renamed later to wuho. By insisting on using the old nomenclature the authors make it difficult to find the relevant information on FlyBase.

<Response to the reviewer>

We thank the reviewer for raising this point. Drosophila wdr4 (dWdr4) was originally identified in Wu, Hou, and Hsieh (Dev Biol, 2006), where the gene was named wuho, derived from the first authors' surnames (Wu and Hou) and also referring to the sterility phenotype ("no progeny" in Chinese). In that study, the wuho⁷ (wh⁷) mutant allele was generated by P-element excision, and this allele designation (wh⁷) has since been recorded in FlyBase.

In the manuscript, we introduce the gene as dWdr4 (historically named Wuho, Wh) at its first mention (line 87), and use wh strictly when referring to the original mutant allele. We have checked the text to ensure consistent and unambiguous nomenclature throughout.

We would also like to acknowledge that this work builds on foundational studies led by Dr. Tao-Shih Hsieh, who made major contributions to wuho gene research and to the development of Drosophila genetics in Taiwan. Dr. Hsieh passed away in 2016, and retaining a brief mention of the historical gene name respects this scientific lineage while still aligning with current FlyBase nomenclature.

We hope the reviewer finds this clarification appropriate.

Line 428: Rephrase the sentence.

<Response to the reviewer>

We have rephrased the sentence.

Materials and Methods: As I suggested earlier, please provide the exact composition of the fly food. "Standard" is not very meaningful as cornmeal-yeast-agar recipes differ between laboratories, e.g. the amount of yeast (m/v %) varies, which is known to influence gut function.

<Response to the reviewer>

We thank the reviewer for this suggestion. We maintain fly stocks on standard cornmeal-agar-yeast food (7.1%, 0.8%, and 2.4% w/v, respectively), and we have now included this information in the Materials and Methods section (p.g. 20, line 477).

Referee #3:

The authors have done an adequate job at addressing my original comments. Regarding my original points 1a, 1b and 1c, I think that the additional data presented give better support to the conclusions. Responses to points 3,4,5,6 and 7 are adequate.

Regarding my original point 2. It's good that the mutant was backcrossed (even though this may have been done a long time ago). However, it cannot be assumed that RNAi lines that were generated as part of the TRiP project are in the same background - flies can accumulate spontaneous mutations over time. This, the very small effect observed in Figure EV2H (compared to the effect of the mutation) and the lack of GAL4-alone and UAS-alone controls, all makes me think that this result provides quite weak support for the authors' conclusion.

On balance, I am recommending publication in EMBO reports.

<Response to the reviewer>

We agree with the reviewer's point. We are aware that TRiP RNAi lines, including their corresponding controls, may accumulate background mutations over time. To address this concern, we previously used an independent RNAi control line (UAS-mCherry[^]RNAi) with the same GAL4 driver and observed similar lifespan results (Materials and Methods, p. 26, line 649). These data support our conclusion that dWdr4 function in ISCs contributes to fly lifespan.

Additionally, because dWdr4 is normally expressed across multiple tissues, it is not surprised that whole-animal dWdr4 mutants show a more severe lifespan reduction compared to ISC-specific knockdown animals. Nevertheless, we have now softened the wording in the revised manuscript to reflect this conclusion more cautiously (p. 10, line 228).

Dear Prof. Hsu,

Thank you for the submission of your further revised manuscript to our editorial offices. I went through your further p-b-p-response and consider the remaining concerns and suggestions of the referees as adequately addressed. However, I have some remaining editorial requests that I ask you to address in a final revised manuscript. Please also provide a final p-b-p-response regarding the editorial requests below.

Editorial requests:

- There is an author name discrepancy. It is Wei-Chiao Chang on the manuscript title page versus Wen Chiao Chang in the submission system. Please check and make sure the same and correct name is used in the manuscript and the submission system.
- Please provide a complete author checklist, which you can download from our author guidelines (<https://link.springer.com/journal/44319/submission-guidelines#cms-Revised-submissions>). Please insert page numbers in the checklist to indicate where the requested information can be found in the manuscript. The completed author checklist will also be part of the RPF.
- Please provide the final individual production quality figure files as .eps, .tif, .jpg (one file per figure), of main figures and EV figures. We do not accept pptx files.
- Please check again that the number "n" for how many independent experiments were performed, their nature (biological versus technical replicates), the bars and error bars (e.g. SEM, SD) and the test used to calculate p-values is indicated in the respective figure legends (main, EV and Appendix figures). Please also check that all the p-values are explained in the legend, and that these fit to those shown in the figure. Please provide statistical testing where applicable. Please avoid the phrase 'independent experiment' but clearly state if these were biological or technical replicates. Please also indicate (e.g. with n.s.) if testing was performed, but the differences are not significant. In case n=2, please show the data as separate datapoints without error bars and statistics. See also:

<https://link.springer.com/journal/44319/submission-guidelines#cms-Figure-and-data-presentation>

If n<5, please show single datapoints for diagrams. Moreover:

- Please note that the exact p values are not provided in the legends of figures 1B', F'; 2A', C', E'; 3A', A', B', C'; 4B', C', D'; 5A', B', C', D', E', F'; 6A', B', C', D'; 7A', B, C, D', E', F'; EV1 A-D; EV2 D', F', H; EV3A', B', C'; EV4 B', D', E', F', G'; A', B', C', D', E, F'; EV5 C, D, F.
- Please note that the measure of center for the error bars needs to be defined in the legends of figures 1B, F; 2C', E'; 3A', B', C'; 4B', C', D'; 5A', B', C', D', E', F'; 6A', B', C', D'; 7A', B, C, D', E', F'; EV1 C, D; EV2 B, D', F'; EV3 A', B', C'; EV4 B', D', E', F', G'; EV5 A', B', C', E, F'; EV6 A, C'
- Please note that the white arrows are not defined in the legend of figure 4C. This needs to be rectified.
- Please note that the dotted borders are not defined in the legends of figures 2A, B, D, E; 3A-C; 4D, 5A, B, C, D, E, F; 6A-D; 7A, D, E, F; EV2C, D, E, F; EV3 B, C; EV4C, D, E, F, G; EV5 C, D, F; EV6 C. This needs to be rectified.
- Please make sure that each figure panel is called out separately and that the panels are called out sequentially. Presently, Fig. 3A is called out before Fig. 2D and a callout for Fig. 3C seems missing. Please check.
- A Table EV1 is called out in the text and a legend is provided. However, no Table EV1 was uploaded. Please provide Table EV1 and put its legend on the first TAB of the excel file. Then, please remove the legend from the manuscript text file.
- Please simplify the nomenclature of the Appendix Tables. Please just call these Appendix Table S1, S2, S3 and so one (not S1A, S1B, S1C and S2A). Please then update their legends and the callouts.
- Thanks for providing the source data (SD). Please upload this as one folder per main figure, grouping together all the files for this figure (and ZIPed together). Moreover, please provide the source data checklist and upload it separately. There seems to be a discrepancy in the list and the files for Figure 4 and the SD following panels appear to be missing: 1ACF, 2BD. Please check. Moreover, for any data deposited at a repository, we need an entry (with a specific URL) in the DAS (data availability section).
- During our figure integrity analysis, we noted overlaps between the images shown in Fig. 2D (upper panels) and in Fig. 4C (upper panel). Please check. If this re-use is intentional, please clearly indicate this in the respective figure legends.
- During our figure integrity analysis, we noted also overlaps between one image shown in Fig. 2A (panel 'intermediate') and one in Fig. EV5D (upper panel). Please check. If this re-use is intentional, please clearly indicate this in the respective figure legends.

- Moreover, we noticed that the microscopy images across the figure set appear pixelated under analysis. This is a common result of converting original 16-bit TIFF images to RGB format for publication, and while not a cause for concern, it can sometimes give the impression of image alteration to critical readers. To resolve this, please upload high-resolution 16-bit source data for all microscopy images.

I look forward to seeing the further revised version of your manuscript when it is ready. Please let me know if you have questions regarding the revision.

Best,

All minor editorial requests have been addressed by the authors.

Prof. Hwei-Jan Hsu
Academia Sinica
Institute of Cellular and Developmental Biology
NO. 128, Academia R.d., Sec.2, Academia Sinica
R337, Institute of Cellular and Organismic Biology
Nankang, Taipei 11529
Taiwan

Dear Prof. Hsu,

I am very pleased to accept your manuscript for publication in the next available issue of EMBO reports. Thank you for your contribution to our journal.

You may qualify for financial assistance for your publication charges - either via a Springer Nature fully open access agreement or an EMBO initiative. Check your eligibility: <https://link.springer.com/journal/44319/how-to-publish-with-us>

Yours sincerely,

>>> Please note that it is EMBO Reports policy for the transcript of the editorial process (containing referee reports and your response letter) to be published as an online supplement to each paper. If you do NOT want this, you will need to inform the Editorial Office via email immediately. More information is available here: <https://link.springer.com/partners/embo-press/editorial-policies#Peer%20review>